# The ribosome lowers the entropic penalty of protein folding

Julian O. Streit[1,3], Ivana V. Bukvin[1,3], Sammy H. S. Chan[1,3 ✉], Shahzad Bashir[1], Lauren F. Woodburn[1], Tomasz Włodarski[1], Angelo Miguel Figueiredo[1], Gabija Jurkeviciute[1], Haneesh K. Sidhu[1], Charity R. Hornby[1], Christopher A. Waudby[1], Lisa D. Cabrita[1], Anaïs M. E. Cassaignau[1 ✉] & John Christodoulou[1,2 ✉]

Most proteins fold during biosynthesis on the ribosome[1], and co-translational folding energetics, pathways and outcomes of many proteins have been found to differ considerably from those in refolding studies[2–10]. The origin of this folding modulation by the ribosome has remained unknown. Here we have determined atomistic structures of the unfolded state of a model protein on and off the ribosome, which reveal that the ribosome structurally expands the unfolded nascent chain and increases its solvation, resulting in its entropic destabilization relative to the peptide chain in isolation. Quantitative [19]F NMR experiments confirm that this destabilization reduces the entropic penalty of folding by up to 30 kcal mol[−1] and promotes formation of partially folded intermediates on the ribosome, an observation that extends to other protein domains and is obligate for some proteins to acquire their active conformation. The thermodynamic effects also contribute to the ribosome protecting the nascent chain from mutation-induced unfolding, which suggests a crucial role of the ribosome in supporting protein evolution. By correlating nascent chain structure and dynamics to their folding energetics and post-translational outcomes, our findings establish the physical basis of the distinct thermodynamics of co-translational protein folding.

Most proteins fold co-translationally during biosynthesis on the ribosome[1]. There is increasing evidence of a direct role for the ribosome in regulating folding of the nascent chain[2–10], with increasing clarity on how it interacts with the elongating nascent chain[7,11–13], which is thought to contribute to alterations to nascent chain thermodynamic stability[6–10] and folding and unfolding rates[5,8]. Consequently, co-translational folding (coTF) differs from in vitro refolding studies of analogous, isolated counterparts[3,5,6,14–16], with unique intermediate conformations in coTF[3,5,6,14,15,17], folding in the absence of the complete protein sequence[4,14], and the ability of the ribosome to mitigate misfolding-prone destabilizing mutations[4] among the many discriminating observations whose origins remain poorly understood. This is a crucial gap in our understanding of proteostasis as many proteins reach an active conformation following coTF, whereas post-translational unfolding–refolding in the cell is generally avoided owing to high kinetic stabilities, and when proteins are unfolded (in vitro), they often do not refold spontaneously, but instead misfold and aggregate[1,18,19].

In contrast to refolding studies, the unfolded state on the ribosome exists under native conditions[7], and is adopted by all proteins during early biosynthesis. The ribosome-bound unfolded state has not been characterized in structural detail owing to technical challenges, yet is likely to be crucial to understanding folding thermodynamics and pathways[20–22]. Here, using paramagnetic relaxation enhancement (PRE) NMR spectroscopy (PRE-NMR) combined with atomistic molecular dynamics simulations, we have determined structural ensembles of

the unfolded state and found that the ribosome structurally expands the conformational ensemble. We infer an entropically driven destabilization of the unfolded state on the ribosome relative to in isolation arising primarily from the increased solvation of the more expanded ensemble. Experiments show that this results in the ribosome reducing the entropic penalty of protein folding by up to around 30 kcal mol[−1]. Despite previous suggestions that interactions between nascent chains and the ribosome surface influence folding kinetics and thermodynamics[5–7], we show here that these interactions account for a minor fraction of the energetic changes observed between protein folding on and off the ribosome. Instead, we establish that the entropic destabilization of the unfolded state provides the fundamental basis for why protein folding on the ribosome is distinct to refolding in vitro.

## Structures of the unfolded state

We investigated the unfolded state of a model immunoglobulin-like domain, FLN5[2,6,7,11,23–25], and determined a set of structural ensembles on and off the ribosome. FLN5 folds reversibly in isolation, thus also facilitating detailed, quantitative comparisons of post-translational folding versus coTF thermodynamics[6,7,11,23,24]. The variant FLN5 $A_3A_3$ (Extended Data Fig. 1a) enables the characterization by NMR of conformational and dynamic preferences of unfolded FLN5 without the complication of folding[7,24]. For the ribosome–nascent chain complex (RNC), FLN5 $A_3A_3$ is tethered to the ribosome peptidyl transferase

[1]Institute of Structural and Molecular Biology, Department of Structural and Molecular Biology, University College London, London, UK. [2]Department of Biological Sciences, Birkbeck College, London, UK. [3]These authors contributed equally: Julian O. Streit, Ivana V. Bukvin, Sammy H. S. Chan. ✉e-mail: s.chan.12@ucl.ac.uk; anais.cassaignau.09@alumni.ucl.ac.uk; j.christodoulou@ucl.ac.uk

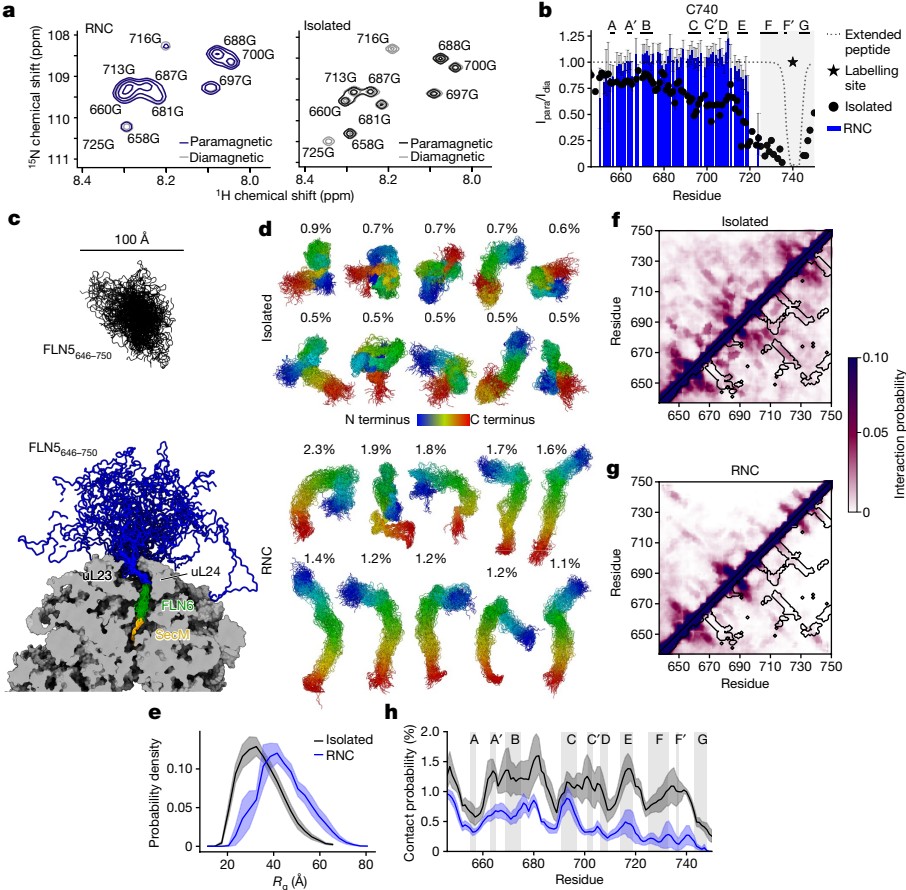

**Fig. 1 | The ribosome modulates the conformational ensemble of the unfolded state. a**, Exemplar regions of a $^1H-^{15}N$ HMQC NMR spectrum of isolated FLN5 $A_3A_3$ spin-labelled at C740 (right) and FLN5+31 $A_3A_3$ C740 (left) with the paramagnetic and diamagnetic spectrum overlaid. NMR data were recorded at 800 MHz, 283 K. **b**, PRE-NMR intensity ratio profiles (intensity in the paragmagnetic, $I_{para}$, and diamagnetic, $I_{dia}$, spectrum; fitted mean ± root mean square error (RMSE) propagated from spectral noise) for isolated FLN5 $A_3A_3$ spin-labelled at C740 (black) and FLN5+31 $A_3A_3$ C740 (RNC; blue). Theoretical reference profiles expected for a fully extended polypeptide are shown as dashed lines (Methods). The secondary structure elements (β-strands) of native FLN5 are indicated at the top. The shaded region at the C terminus represents the region of FLN5 that is broadened beyond detection owing to ribosome interactions[7] (N730–K746, in the RNC). **c**, Representative ensembles

of unfolded FLN5 on and off the ribosome. The structures shown represent the top 50 cluster centroids (to scale; clustering details in Methods) after reweighting with the PRE-NMR data. **d**, Top 10 individual structural clusters on and off the ribosome labelled with their respective population (not to scale). **e**, Distributions of the radius of gyration ($R_g$) for both ensembles for the residues belonging to the FLN5 domain (M637–G750). The shaded area represents the s.e.m. estimated from block averaging. **f,g**, Average inter-residue contact maps of isolated (**f**) and ribosome-bound (**g**) FLN5 (zoomed to a probability of 0.1 for ease of visualization). The black contours represent contacts formed in the native state of FLN5. **h**, Average inter-residue long-range (defined here as a separation of at least ten residues) contact probability along the protein sequence (shaded regions represent s.e.m. from block averaging).

centre (PTC) via a 31-amino acid linker (FLN5+31 $A_3A_3$), comprising the subsequent FLN6 domain and SecM stalling sequence (Extended Data Fig. 1a). This construct has the entire FLN5 sequence emerged from the ribosomal exit tunnel and is the earliest linker length at which some folding is observed in wild-type FLN5[6,11]. PRE-NMR experiments of FLN5 $A_3A_3$ showed less broadening for the RNC compared with the isolated proteins (Fig. 1a,b, Extended Data Figs. 1 and 2 and Methods), suggesting that the conformational ensemble is less compact on the ribosome (Supplementary Notes 1–4). Restraints obtained from these experiments were used to reweight all-atom molecular dynamics simulations with explicit solvent of the unfolded states (Methods and Supplementary Notes 5–7). Molecular dynamics simulations of the isolated protein were initially used to identify a suitable force field for this protein (Extended Data Fig. 3 and Supplementary Note 5) and were subsequently validated against the radius of hydration ($R_h$), NMR chemical shifts, residual dipolar couplings (RDCs) and small-angle X-ray scattering (SAXS) data (Extended Data Fig. 4 and Supplementary Note 6). The simulations exhibited good convergence with respect to the overall compactness, secondary structure and long-range contacts

in the ensembles (Supplementary Note 7). The reweighted structural ensembles are in good agreement with both the PRE-NMR and validation data on and off the ribosome (Extended Data Figs. 4 and 5).

Both structural ensembles of the unfolded state on and off the ribosome display heterogeneity (Fig. 1c,d). An analysis of the main structural clusters reveals that the isolated ensemble samples more compact and spherical states (Fig. 1d and Extended Data Fig. 6i) with the radius of gyration of the nascent chain increasing by approximately 26% on the ribosome from 34.9 ± 1.0 Å to 44.1 ± 1.8 Å (Fig. 1e). This structural expansion (throughout this Article, 'expansion' refers to structural expansion) of the ensemble is partly caused by steric exclusion from and tethering to the ribosome, but additional factors also contribute (Supplementary Note 8). Owing to the expansion, the amount of β-strand secondary structure in the RNC ensemble decreases along the entire sequence from 3.2 ± 0.5% to 1.1 ± 0.3% in total (Extended Data Fig. 5f) and fewer contacts are observed compared with the isolated protein (0.4 ± 0.1% and 1.0 ± 0.2% on average, respectively; Fig. 1h). Most of these transient contacts are non-native (Fig. 1f,g) and only 1.4 ± 0.2% and 1.0 ± 0.1% of native contacts are formed off and on the ribosome,

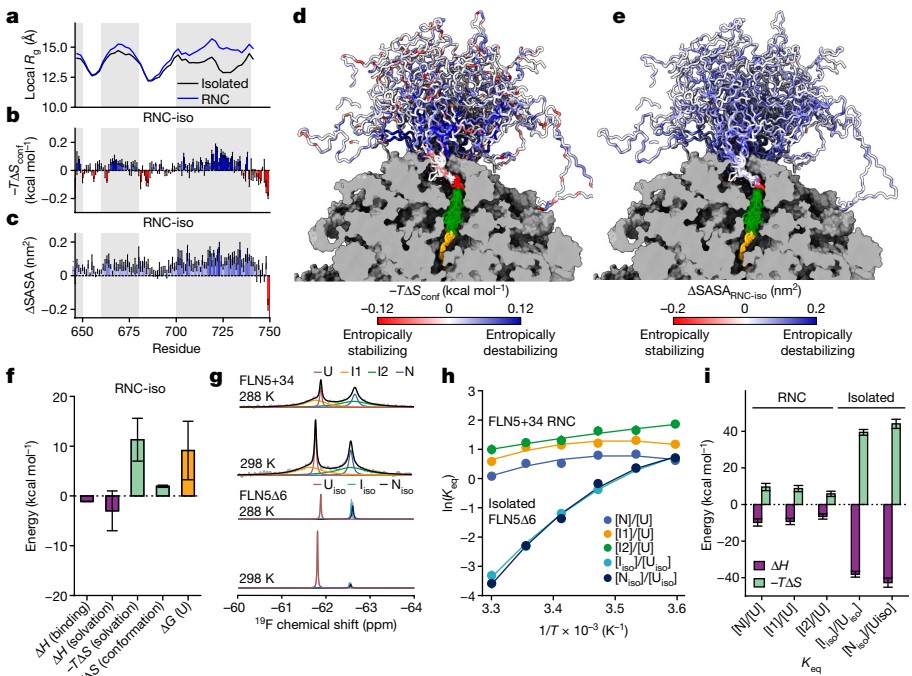

**Fig. 2 | The unfolded state is entropically destabilized on the ribosome.**
**a**, The local radius of gyration along the sequence (21-residue moving average). The shaded regions correspond to residues K646–S650 (left), G660–K680 (middle) and G700–N740 (right). **b**, Difference (RNC-iso) in the total conformational entropy using 50 bins (Methods). The s.e.m. was estimated from block averaging (using a 7.5 µs sampling block size). Bars are coloured according to the gradient in **d**. **c**, Difference in the maximum theoretical SASA of the unfolded state (RNC-iso; Methods). The s.e.m. was estimated from block averaging. The bars are coloured according to the gradient in **e**. **d**, Entropy difference between the RNC and isolated ensembles mapped onto representative ensembles of FLN5+31 $A_3A_3$. **e**, Changes in SASA between the RNC and isolated ensembles is mapped onto the representative ensembles of FLN5+31 $A_3A_3$. **f**, Bar chart (mean ± s.e.m.) summarizing the

energetic changes between the unfolded state on and off the ribosome (RNC-iso) at 298 K. All quantities are estimated based on the molecular dynamics ensemble averages, except the 'ribosome binding' contribution, which was experimentally determined[7]. Errors were combined from block averaging and empirical parameter uncertainties (Methods). **g**, [19]F NMR spectra of the folding equilibrium of FLN5 labelled with 4-trifluoromethyl-L-phenylalanine (tfmF) at position 655 on and off the ribosome at 288 K and 298 K. Native (N), unfolded (U) and intermediate states on (I1, I2) and off ($I_{iso}$) the ribosome are indicated. **h**, Temperature dependence of the folding equilibrium constant ($K_{eq}$) of FLN5 on and off the ribosome measured by [19]F NMR (mean ± s.e.m.). Data were fit to a modified Gibbs–Helmholtz equation (Methods). **i**, Thermodynamic parameters (mean ± s.d. from fits, $T$ = 298 K) calculated from the nonlinear fit in **h**.

respectively. Long-range contacts are particularly reduced at the C terminus (residues N730–G750) of FLN5 $A_3A_3$ (Fig. 1h), which in turn is bound to the ribosome surface around 80% of the time[7] (Extended Data Fig. 5g,h). These nascent chain–ribosome interactions are driven predominantly by electrostatic effects and mediated via ribosomal RNA and RNA-bound $Mg^{2+}$ ions (Extended Data Fig. 5i,j), whereas contacts within the unfolded protein itself occur more frequently between hydrophobic amino acids (Extended Data Fig. 5k). This structural analysis demonstrates that the ribosome significantly affects the global structural properties of the unfolded state.

## Entropic destabilization on the ribosome

We utilized our structures of the unfolded state on and off the ribosome to estimate their effect on folding energetics from an enthalpic ($\Delta H$) and entropic ($\Delta S$) point of view, both of which determine the folding free energy ($\Delta G = \Delta H - T\Delta S$). Ribosome interactions have been shown to modulate folding thermodynamics and these interactions with the unfolded FLN5+31 $A_3A_3$ nascent chain result in a destabilization of the folding free energy[7] ($\Delta\Delta G_{N-U,RNC-iso}$; where N, U, and iso are native state, unfolded state and isolated protein, respectively) by +1 kcal mol[−1].

We explored whether the overall entropy of the unfolded state changes on the ribosome compared to off the ribosome ($\Delta S_{RNC-iso}$). Using our molecular dynamics ensembles, we analysed the protein conformational entropy ($\Delta S_{conf}$) and solvation entropy ($\Delta S_{solv}$) (Methods), which together comprise the total entropy change ($\Delta S = \Delta S_{conf} + \Delta S_{solv}$). A residue-specific analysis of the unfolded state shows distinct

expanded regions on the ribosome (having an increased local radius of gyration; Fig. 2a). The same nascent chain regions (for example, residues G700–N740) also show a significant reduction in the conformational entropy on the ribosome (Fig. 2b) and more restricted sampling of the Ramachandran map (Extended Data Fig. 6e), which is consistent with a more elongated shape of the nascent chain ensemble[26] (Fig. 1b and Extended Data Fig. 6h). We observed this decrease in conformational entropy orthogonally through a cluster analysis, revealing fewer accessible conformational states on the ribosome relative to off the ribosome (Extended Data Fig. 6a,b). Notably, the entropic destabilization is observed even for residues distal to the ribosome (for example, V664–I674), showing that the ribosome exerts a long-range entropic effect that arises from more than ribosome interactions alone (Fig. 2b). The conformational restriction imposed by the ribosome is estimated to globally destabilize the unfolded state relative to the isolated unfolded protein ($-T\Delta S_{RNC-iso,conf}$) by at least +2 kcal mol[−1] at 298 K (Methods and Extended Data Fig. 6g).

An increase in solvation entropy has long been described as the major driving force of the hydrophobic collapse in protein folding[27]. The solvation entropy of the unfolded state was thus explored by analysing the solvent-accessible surface area (SASA) of FLN5. The SASA was significantly increased on the ribosome compared to off the ribosome (+6 ± 1 nm[2] in total; Extended Data Fig. 6i), particularly in regions where the nascent chain is locally expanded (Fig. 2c). On the basis of the changes in SASA, we estimated the resulting solvent entropy changes (Methods and Supplementary Note 9). These calculations show a reduced solvation entropy which further destabilizes the

unfolded state on the ribosome ($-T\Delta S_{\text{RNC-iso,solv}}$) by $+11 \pm 4$ kcal mol$^{-1}$ at 298 K. Both the conformational and solvation entropy are thus globally reduced throughout the RNC ensemble (Fig. 2d,e). This results in a combined entropic destabilization of $13 \pm 4$ kcal mol$^{-1}$, outcompeting both enthalpic gains in stability due to ribosome interactions and increased solvation ($\Delta H_{\text{RNC-iso,solv}} = -3 \pm 4$ kcal mol$^{-1}$). The net increase in free energy of the unfolded state on compared to off the ribosome is therefore expected to be $+9 \pm 6$ kcal mol$^{-1}$ (Fig. 2f) at this nascent chain length. The strong contribution of solvation entropy effects was also verified using direct entropy calculations, resulting in an estimate of approximately $30 \pm 10$ kcal mol$^{-1}$ at 298 K (Supplementary Note 10).

## Measurements of folding thermodynamics

To experimentally consider these entropic effects, we sought to determine $\Delta S$ and $\Delta H$ of folding by investigating the temperature dependence of folding for wild-type FLN5. The FLN5+34 RNC and the C-terminal truncation FLN5$\Delta$6 variant[24] as the analogous isolated protein were selected, both of which enable the simultaneous observation of the unfolded and native states by $^{19}$F NMR spectroscopy[6] within the same temperature range (278 K–303 K). Under these conditions, we also observe two folding intermediates in the FLN5+34 RNC (I1 and I2) and one intermediate in the isolated FLN5$\Delta$6 variant[6,24] (I$_{\text{iso}}$) (Fig. 2g and Extended Data Fig. 7a,b).

The 1D $^{19}$F NMR spectra were fitted to determine the population of each species, enabling quantification of thermodynamic parameters from a nonlinear fit of the equilibrium constant as a function of temperature (Fig. 2h and Methods). Both on and off the ribosome, the apparent enthalpy of folding ($\Delta H_{\text{N-U}}$) is negative, whereas the apparent entropy of folding ($-T\Delta S_{\text{N-U}}$) is positive—that is, the folding reaction is enthalpy-driven to compensate for an unfavourable entropic penalty. The heat capacity of folding ($\Delta C_{\text{p,N-U}}$) obtained for the isolated protein ($-1.7 \pm 0.3$ kcal mol$^{-1}$ K$^{-1}$) is as expected on the basis of protein size[28] ($-1.5 \pm 0.2$ kcal mol$^{-1}$ K$^{-1}$), but increases on the ribosome ($\Delta\Delta C_{\text{p,N-U,RNC-iso}} = +0.9 \pm 0.4$ kcal mol$^{-1}$ K$^{-1}$), presumably owing to the increased water ordering and local ion concentration near the ribosome surface[29]. These experiments also show the temperature dependence of folding of the RNC to be significantly attenuated compared to the corresponding isolated protein (Fig. 2g,h) with the magnitudes of $\Delta H$ and $-T\Delta S$ being strongly reduced on the ribosome ($\Delta\Delta H_{\text{N-U,RNC-iso}} = +32.9 \pm 3.2$ kcal mol$^{-1}$, $-T\Delta\Delta S_{\text{N-U,RNC-iso}} = -34.5 \pm 3.2$ kcal mol$^{-1}$ at 298 K; Fig. 2i). Folding on the ribosome is consequently less enthalpically driven but also exhibits a lower entropic penalty (Fig. 2i). The reduction in $-T\Delta S$ on the ribosome experimentally confirms the predicted entropic destabilization of the unfolded state and is within the range of the estimated solvation entropy change based on a solvation analysis of our molecular dynamics ensembles ($30 \pm 10$ kcal mol$^{-1}$; Supplementary Note 10). Of note, folding from the intermediate state(s) to the native state is only marginally sensitive to temperature, both on and off the ribosome (Extended Data Fig. 7c,d), corroborating that the entropic differences originate predominantly from modulation of the unfolded state. The less negative $\Delta H$ on the ribosome must therefore predominantly result from the destabilization of the native state on the ribosome relative to off the ribosome[6] (Supplementary Note 11). Our thermodynamic experiments thus clearly show that the expansion of the unfolded state results in the lowering of the entropic penalty of folding relative to the isolated protein.

## Entropy effects are sequence-independent

Given the strong interactions of the unfolded FLN5 nascent chain with the negatively charged ribosome surface as observed in our structures, we next examined its effect on the large folding enthalpy and entropy differences on and off the ribosome. We performed $^{19}$F NMR experiments of a polyglutamate mutant (E6) (Extended Data Fig. 7e,f),

which has reduced ribosome interactions[7] (from $85 \pm 5\%$ to $10 \pm 2\%$). Large changes in folding enthalpy and entropy (relative to an analogous isolated protein) are still observed and only marginally reduced relative to wild-type ($\Delta\Delta H_{\text{N-U,RNC-iso}} = +22.6 \pm 5.5$ kcal mol$^{-1}$, $-T\Delta\Delta S_{\text{N-U,RNC-iso}} = -20.6 \pm 5.5$ kcal mol$^{-1}$ at 298 K, Extended Data Fig. 7g,h). These results show that ribosome interactions only partially contribute to the large change in coTF energetics. This is consistent with the entropic effects originating predominantly from the increased hydration of the expanded nascent chain (Fig. 2f), suggesting that this phenomenon may be sequence-independent.

## Persistence during biosynthesis

We reasoned that the structural expansion of the unfolded state and re-balanced enthalpy–entropy of coTF should decrease in magnitude as the nascent chain elongates and the distance between FLN5 and the ribosome surface increases. To test this, we performed PRE-NMR experiments on the unfolded nascent chain of two longer FLN5 RNCs (FLN5+47 A$_3$A$_3$ and FLN5+67 A$_3$A$_3$; Extended Data Fig. 8a–c). The measured PRE intensity ratios decrease with increasing nascent chain length (Fig. 3a), showing that the expansion decreases as expected (see Supplementary Note 2). However, the intensity ratios remain higher than those of the isolated protein, indicating that the unfolded nascent chain remains more expanded on the ribosome at all RNC lengths tested, highlighting the long-range effect that the ribosome exerts on nascent chain structure. We next measured the enthalpy and entropy of folding of FLN5+67 using our $^{19}$F NMR approach (Fig. 3b and Extended Data Fig. 8d–i). Correlating with the decreased structural expansion of the unfolded state at FLN5+67 (relative to FLN5+31), we observed that the change in folding entropy on the ribosome persists but is reduced from $-34.5 \pm 3.2$ kcal mol$^{-1}$ at FLN5+34 to $-10.1 \pm 2.8$ kcal mol$^{-1}$. Likewise, the enthalpy of folding becomes more favourable as the nascent chain elongates from FLN5+34 to FLN5+67 and the native state becomes less destabilized further away from the ribosome surface[6,9,30,31]. We conclude that the thermodynamic effects persist during biosynthesis but progressively decrease in magnitude. These experiments also establish a direct relationship between the structure of the unfolded nascent chain and coTF thermodynamics.

We then explored whether the entropic destabilization of the unfolded nascent chain during biosynthesis could rationalize the observed differences in the folding of FLN5 on and off the ribosome, common to other proteins[3,5,8–10,17,30–33]. Whereas the native state is destabilized on the ribosome relative to the native state in isolation[6,7,25] (Extended Data Fig. 7i–k), FLN5 paradoxically populates two coTF intermediates that are significantly more stable than the single intermediate found in isolation (I$_{\text{iso}}$ of FLN5$\Delta$6; Fig. 2g) and which are completely undetectable in full-length isolated FLN5[6,24]. The stabilities of the coTF intermediates are modulated by the nascent chain length, such that at FLN5+47, their stabilities are more than 4 kcal mol$^{-1}$ greater than that of I$_{\text{iso}}$ (relative to their respective unfolded states[6]) (Fig. 3c). We quantified the contribution of ribosome binding to stabilizing the coTF intermediates by estimating the population of intermediates bound to the ribosome surface based on their measured rotational correlation times—that is, how fast the domain tumbles in solution (Extended Data Fig. 8j–p and Methods). These experiments indicate that such binding can only account for less than 0.1 kcal mol$^{-1}$ of stabilization on the ribosome at FLN5+47 (Fig. 3c). Therefore, ribosome interactions contribute only weakly to stabilizing coTF intermediates. These measurements are also consistent with the observed persistence of the intermediates within a broad folding transition[6] (that is, from approximately FLN5+31 to FLN5+67) and in a range of conditions that disrupt or reduce ribosome–nascent chain interactions, including changes in the distance from the ribosome (Fig. 3c), high concentrations of salt and urea, nascent chain and ribosome surface mutations[2,6] (Extended Data Fig. 7e–h), and temperature (Fig. 2g,h).

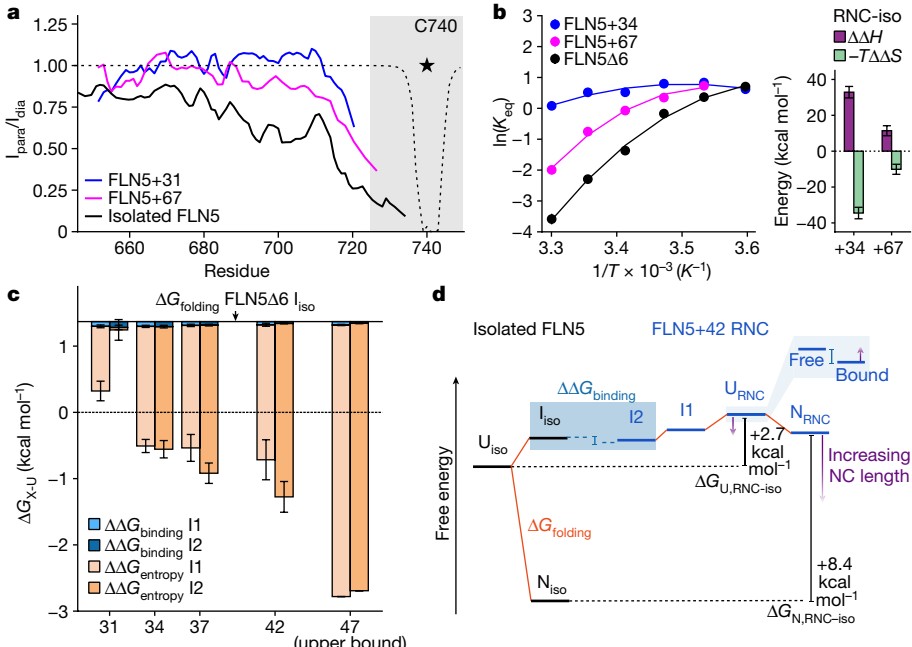

**Fig. 3 | Entropic destabilization persists at long linker lengths and leads to stable coTF intermediates. a**, PRE-NMR intensity ratio profiles are shown for the C740 labelling site (black star) window averaged over three residues for isolated FLN5 $A_3A_3$, FLN5+31 $A_3A_3$ and FLN5+67 $A_3A_3$ as in Fig. 1b. **b**, Left, temperature dependence of the folding equilibrium constants of isolated FLN5Δ6, FLN5+34 (wild-type) and FLN5+67 measured by $^{19}$F NMR (mean ± s.e.m.) fit to a modified Gibbs–Helmholtz equation (Methods). The error bars of individual datapoints are similar in magnitude to the size of the circles. Right, Thermodynamic parameters (mean ± s.d., $T$ = 298 K) calculated from the nonlinear fits and shown as the difference relative to the isolated protein. **c**, Folding free energies (mean ± s.e.m. propagated from NMR lineshape fits) of the coTF intermediates I1 and I2 at different linker lengths ($x$ axis) (ref. 6 and Extended Data Fig. 8o). The folding free energy of the isolated FLN5Δ6

intermediate is shown as a horizontal line. The contributions to the stabilization of the intermediates on the ribosome due to ribosome binding ($\Delta\Delta G_{binding} = RT\ln(1 - p_B)$, where $p_B$ is fraction bound) and entropy ($\Delta\Delta G_{entropy} = \Delta\Delta G_{I\text{-}U,RNC\text{-}iso} - \Delta\Delta G_{binding}$) are shown as vertical bars. The ribosome-bound population was estimated using a $\tau_{R,bound}$ (rotational correlation time of the bound state) of 3,003 ns ($S^2_{bound}$ = 1.0; order parameter of the bound state). **d**, Model of the free energy landscape of folding on and off the ribosome. The unfolded (U) state is destabilized on the ribosome relative to in isolation, outcompeted by stabilizing ribosome interactions. $I_2$ is stabilized by less than 0.1 kcal mol$^{-1}$ on the ribosome owing to interactions (see **c**) from its folding free energy of at least $\Delta G_{I(iso)\text{-}U(iso)}$ (that is, the stability of $I_{iso}$, owing to the structural similarity[6] between $I_{iso}$ and I2; lower bound estimate). $^{19}$F NMR has shown that the native state is destabilized relative to U[6].

We next built a model of the free energies of coTF by comparison to the isolated protein. As the most stable intermediate off the ribosome, $I_{iso}$, is structurally similar to I2 (ref. 6), we used our measurements of binding energies (Fig. 3c) to link the relative free energies of FLN5 on and off the ribosome (Fig. 3d). From this thermodynamic analysis, we can infer that the unfolded state in the FLN5+42 RNC (the longest linker length at which an unfolded population is observed[6]) is destabilized by at least 2.7 kcal mol$^{-1}$ relative to the isolated unfolded protein ($\Delta\Delta G_{U,RNC\text{-}iso}$; Fig. 3d). Together, we conclude that the ribosome persistently destabilizes unfolded and folded FLN5 during biosynthesis to promote the formation of partially folded intermediates.

## Thermodynamic effects across proteins

As the entropic effects are at least partly sequence-independent (Extended Data Figs. 5l,m and 7), we examined whether the reduction of the folding entropy penalty on the ribosome and its implications for coTF are also observed for other proteins. We investigated the folding of a structurally homologous domain, titin I27 (the 27th immunoglobulin-like domain of titin; Fig. 4a) and the common oncoprotein HRAS[34], a GTPase protein with an α/β-fold[35] (Fig. 4b). In isolation, I27 has been shown to fold reversibly[36], which, as for FLN5, enables thermodynamic comparisons of folding on and off the ribosome. I27 exhibits two-state folding behaviour in urea (Fig. 4c) but populates one high-energy intermediate in a destabilized mutant ($I_{iso}$; Extended Data Fig. 9a). Although a previous study suggested that the ribosome does not affect folding of I27[37], our results show two folding intermediates being stabilized on

the ribosome (Fig. 4c, I1 and I2). Similarly, $^{19}$F NMR spectra of HRAS also show the population of stable coTF intermediates, even before complete translation, that are not populated in isolation (Fig. 4d). The coTF intermediates of both proteins are partially folded, since they are completely destabilized by mutations that disrupt the native hydrophobic core[37] (Extended Data Fig. 9c,e,f). Furthermore, as observed for FLN5, the temperature dependence of folding is reduced for I27 on the ribosome (Fig. 4e,f and Extended Data Fig. 9a,b), with a reduced enthalpy of folding ($\Delta\Delta H_{N\text{-}U,RNC\text{-}iso}$ = +28.8 ± 10.1 kcal mol$^{-1}$ at 298 K) and a lower entropic penalty of at least 18 kcal mol$^{-1}$ on the ribosome ($-T\Delta\Delta S_{N\text{-}U,RNC\text{-}iso}$ = −28.5 ± 10.1 kcal mol$^{-1}$ at 298 K). Folding from the unfolded state to the first HRAS intermediate (I1, HRAS$_{1\text{-}81}$ RNC) is similarly temperature-insensitive (Fig. 4g and Extended Data Fig. 9d) and exhibits an entropic penalty ($-T\Delta S_{I1\text{-}U}$) of only +5.0 ± 2.0 kcal mol$^{-1}$ (Fig. 4h). The thermodynamic effects reported in this work and the resulting population of stable coTF intermediates thus appear to be a general phenomenon.

Given the differences in folding thermodynamics and pathways on and off the ribosome, we sought to examine how coTF events may determine the post-translational fate of nascent proteins. Whereas our model systems FLN5 and I27 have been shown to fold reversibly to their native state in isolation[36,38], many proteins are not able refold off the ribosome[1,18,19]. Indeed, the proteolytic stability of the KRAS isoform has been found to be modulated by codon usage[39], and so we examined whether the acquisition of native HRAS structure is also dependent on its coTF pathway. Consistent with prior observations on KRAS, refolded isolated HRAS showed reduced proteolytic stability compared with control or native HRAS (Extended Data Fig. 9h), which

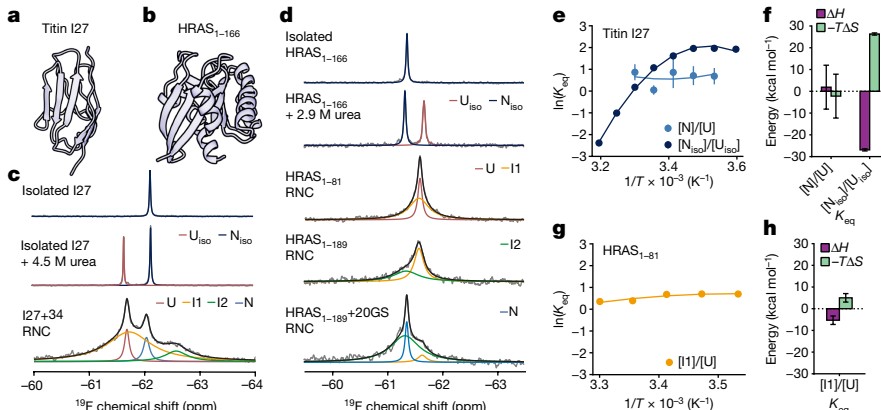

**Fig. 4 | Co-translational folding intermediates of titin I27 and HRAS are stabilized on the ribosome. a,b**, Crystal structures of titin I27 (Protein Data Bank (PDB): 1TIT) (**a**) and HRAS (PDB: 4Q21) (**b**). **c**, [19]F NMR spectra of titin I27, tfmF-labelled at position 14, off the ribosome (isolated), isolated I27 in urea, and I27 tethered with a 34-residue linker to the ribosome (I27+34 RNC) recorded at 298 K. **d**, [19]F NMR spectra of the HRAS G-domain (residues 1–166), tfmF-labelled at position 32, off the ribosome (isolated), in urea and HRAS on the ribosome arrested at 3 different lengths and recorded at 298 K. HRAS$_{1–81}$, HRAS$_{1–189}$ and HRAS$_{1–189}$+20GS correspond to residues 1–81 of HRAS, full-length HRAS and full-length HRAS with an additional 20-residue poly(glycine-serine) linker, respectively, each tethered to the ribosome by the arrest-enhanced SecM motif. **e,g**, Nonlinear fits to a modified Gibbs–Helmholtz equation (Methods) of the equilibrium constants (mean ± s.e.m. propagated from NMR lineshape fits) of titin I27 folding off and on the ribosome (**e**) and the HRAS$_{1-81}$ RNC (**g**) measured by [19]F NMR. **f**, Thermodynamic parameters determined from the nonlinear fit (mean ± s.d., $T$ = 298 K) of the temperature dependence of I27 folding. The heat capacity of folding ($\Delta C_{p,N-U}$) obtained for the isolated and ribosome-bound protein are −2.0 ± 0.1 and +0.6 ± 1.7 kcal mol⁻¹ K⁻¹, respectively. The heat capacity of the isolated protein is similar to the literature value reported for the wild-type variant[28] (−1.4 kcal mol⁻¹). **h**, Thermodynamic parameters determined from the nonlinear fit (mean ± s.d., $T$ = 298 K) of folding for HRAS$_{1–81}$. The heat capacity of folding obtained from the fit is −0.4 ± 0.3 kcal mol⁻¹ K⁻¹.

also persisted when refolded in eukaryotic cell lysate (Extended Data Fig. 9i). A residue-specific analysis by [1]H,[15]N NMR shows that refolded HRAS forms a native-like, GDP-bound conformation, consistent with prior biophysical experiments[40], but distinct structural regions, including the switch 2 region that is involved in nucleotide exchange[41], show increased NMR signal intensities—that is, probably altered backbone dynamics (Extended Data Fig. 9j). Indeed, when assessing HRAS function with a GDP/GTP nucleotide exchange assay, we found that refolded HRAS is completely inactive, whereas HRAS purified from cells and the HRAS+20GS RNC are both active (Extended Data Fig. 9g). Subtle differences in structure and dynamics thus alter the fate of refolded HRAS, which appears to be kinetically trapped in an inactive state. These results show that the thermodynamic modulation by the ribosome and resulting coTF pathway appear to be obligate to the formation of functionally active HRAS.

## Mutations are buffered on the ribosome

We hypothesized that the ribosome additionally modulates the effect of destabilizing mutations[4]. To test this, we designed nine variants of FLN5 that include disruptions to the hydrophobic core, proline isomerization[24] and electrostatic charge[7]. For all mutants, we measured the folding free energy on and off the ribosome using [19]F NMR (Fig. 5a,b and Extended Data Fig. 10a,b,h,i). The mutants exhibited a wide range of stabilities ($\Delta G_{N-U}$) from −0.7 to −5.4 kcal mol⁻¹ off the ribosome (equal to a 4.7 ± 0.3 kcal mol⁻¹ range in stabilities). However, on the ribosome, the stabilities of the mutants exhibited a narrower range of 1.5 ± 0.1 kcal mol⁻¹ (Fig. 5a) and were all less destabilizing ($\Delta\Delta G_{N-U}$) by 0.3–3.7 kcal mol⁻¹ (Fig. 5b).

We speculated that the re-balanced enthalpy–entropy compensation contributes to this buffering effect. Given the large contribution from increased nascent chain solvation (Fig. 2f), we also measured the destabilization of four hydrophobic mutants in the presence of 2.5 M urea (Extended Data Fig. 10c); urea weakens the hydrophobic effect by displacing several water molecules from the protein solvation shell[42,43]. By effectively reducing the gains in solvation of the unfolded RNC, we find that the mutations are less strongly buffered in urea (Fig. 5c).

In agreement with this, the differences in entropy and enthalpy of folding on the ribosome (relative to the isolated protein) are reduced in urea compared with in pure water (Fig. 5d and Extended Data Fig. 10e–g). Thus, the magnitude and extent of mutation buffering correlates with the reduced temperature dependence of protein folding on the ribosome. We conclude that an additional consequence of the destabilized unfolded and folded states is to buffer and therefore mitigate the effect of destabilizing mutations during coTF folding.

## Discussion

Here we have determined an atomistic structural ensemble of a nascent, unfolded protein tethered to the ribosome, analysed its differences to the protein in isolation, and studied its implications for coTF. Our structures reveal that the unfolded state on the ribosome is more structurally expanded and samples fewer long-range contacts than off the ribosome. The ribosome thus has a key role in shaping the conformational space of the emerging nascent chain. The expansion and increased solvation of the unfolded state on the ribosome (Fig. 6) result in reduced conformational and water entropies, a finding that constrasts with previous theoretical studies[44,45]. The entropic destabilization observed in the unfolded nascent chain relative to the isolated protein outcompetes the enthalpic stabilization provided by electrostatic ribosome interactions[7] and increased solvation (Fig. 2f). Meanwhile, the native state structure or environment is perturbed (Extended Data Fig. 7i,j) and enthalpically destabilized on the ribosome relative to its isolated form[6] (Figs. 2i and 3d), probably owing to the space constraints near the exit vestibule[30] and long-range electrostatic effects from the negatively charged ribosome surface[9,31]. Together, these effects result in a marked re-balancing of the enthalpy–entropy compensation for protein folding that occurs on the ribosome (Fig. 6).

The re-balanced enthalpy–entropy compensation of coTF folding provides a physical rationale for understanding differences in the folding on and off the ribosome. It enables nascent proteins to fold via distinct partially folded intermediates on the ribosome that are absent or significantly less stable in isolation[6,24], as observed for multiple proteins[3,6,14,15,17,32,33] including all three model systems in this work

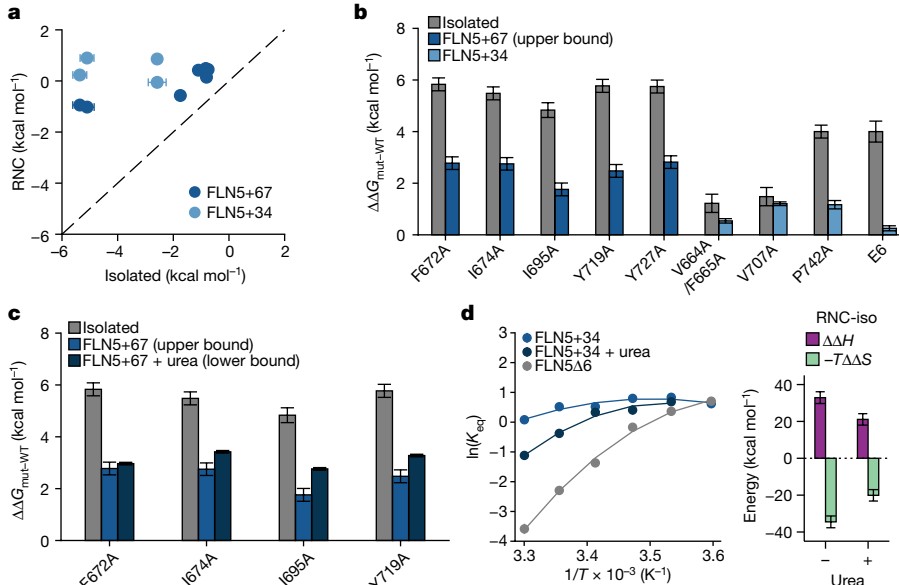

**Fig. 5 | Destabilizing mutations are buffered by the ribosome. a**, Folding free energies of destabilizing mutants off and on the ribosome (FLN5+34 and FLN5+67 depending on the stability of the mutant) determined by [19]F NMR from U and N state populations. **b**, Destabilization ($\Delta\Delta G_{N\text{-}U,\text{mut-WT}}$) of all mutants in isolation compared to the RNC. WT, wild type. **c**, Destabilization of 4 mutants in isolation, on the ribosome (FLN5+67) and on the ribosome in the presence of 2.5 M urea. **d**, Left, temperature dependence of the folding equilibrium constant involving the N and U state of isolated FLN5Δ6, FLN5+34 and

FLN5+34 in 1.5 M urea measured by [19]F NMR fit to a modified Gibbs−Helmholtz equation (Methods). The error bars of individual datapoints (propagated from bootstrapped errors of NMR lineshape analyses) are similar in magnitude to the size of the circles. Right, thermodynamic parameters ($T$ = 298 K) from the nonlinear fits (mean ± s.d.) shown as the difference relative to the isolated protein. Unless stated otherwise, all values in the figure represent the mean ± s.e.m. propagated from NMR lineshape fits.

(Figs. 3 and 4). Although the biological effect could be reduced for small proteins that unfold and refold post-translationally, the substantial physical, thermodynamic changes on the ribosome are likely to affect the entire folded proteome during biosynthesis. Indeed, our analyses reveal that the expansion and destabilization of the unfolded state is partially caused by the physical effects of steric exclusion and tethering (Extended Data Fig. 5l,m) and is not dependent on sequence-specific ribosome–nascent chain interactions (Extended Data Fig. 7e–h). The ribosome can therefore act as a universal foldase, that in contrast to others is ATP-independent, and can promote the formation of functionally active proteins, many of which do not spontaneously unfold and refold off the ribosome[1,18,19], including HRAS (Extended Data Fig. 9g–j).

The distinct thermodynamics of nascent chains may benefit other co-translational processes that are also entropically disfavoured, such as chaperone binding[46,47], translocation[47] or protein assembly[47–49]. The high stabilities of coTF intermediates across a wide folding transition, as observed for FLN5[6,24] and HRAS (Fig. 4d), may additionally provide a longer time frame for such processes to occur. Conversely, partially folded intermediates may result in the formation of non-productive states, such as off-pathway or misfolded species[4,50–52], highlighting that in the cellular environment there is indeed a fine line between folding and misfolding on the ribosome.

Finally, we present quantitative evidence of mutation buffering by the ribosome as an additional consequence of the thermodynamic effects occurring co-translationally. Throughout evolution proteins diversify through mutations, most of which are destabilizing and impose limits on evolvability while maintaining a fold and function[53,54]. Many destabilizing mutants studied here would be expected to be fully unfolded co-translationally in the absence of buffering, despite being completely natively folded in isolation (Extended Data Fig. 10h), because folded structure is less stable on the ribosome (Fig. 3d). The buffering effect thus minimizes the increased population of unfolded nascent chain resulting from a destabilizing mutation, effectively promoting coTF over post-translational folding and averting potentially harmful consequences for mutant proteins. For example, accumulation of

unfolded populations on the ribosome and failure of coTF have been linked to co-translational ubiquitination and degradation of nascent chains[55]. Additionally, lack of coTF could be detrimental for nascent chains that rely on co-translational complex assembly (more than 20% of the proteome[48]), chaperone engagement[46], or cannot fold into an active conformation post-translationally[1,18,19] (Extended Data Fig. 9g–j). Notably, cellular chaperones have also been implicated in mutation buffering and their availability has been linked to the rate at which proteins evolve[53,56–60]. CoTF may therefore also have a universal role in mutation buffering and evolution during the initial stages of protein folding before transferring nascent chains to chaperones[47].

In conclusion, we have demonstrated that the ribosome entropically destabilizes the unfolded state. This provides a general, physical

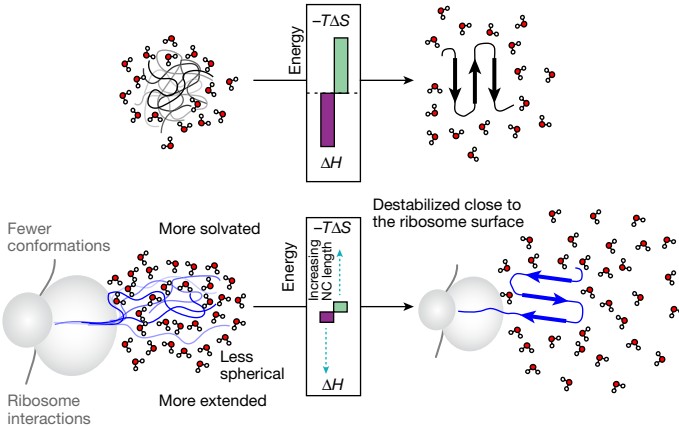

**Fig. 6 | The ribosome re-balances the enthalpy–entropy compensation of protein folding.** The unfolded state is entropically destabilized owing to a lower conformational entropy and increased solvation, whereas the native state is enthalpically destabilized. This results in a reduction of the entropic penalty for folding and a less favourable folding enthalpy. The labels in grey and black indicate weak and strong energetic contributions, respectively.

explanation for the fundamental differences in protein folding pathways and energetics observed in vitro versus on the ribosome. Beyond the effects of steric exclusion and tethering, other factors that contribute to the destabilization of the unfolded and native states on the ribosome remain unexplored. Deeper insights may decipher additional physical principles behind what we propose to be a universal phenomenon during de novo protein folding.

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

## Methods

### Protein expression and purification

DNA constructs of FLN5 were previously described[7,11]. Coding sequences for titin I27 and HRAS were introduced into the pLDC-17 vector using standard procedures. Further mutations were introduced using site-directed mutagenesis; for [19]F labelling, amber stop codons were introduced[6] in position 32 in HRAS, and residue 14 with an additional K87H point mutation in I27. FLN5 variants were expressed as His-tagged proteins and isotopically labelled in *Escherichia coli* BL21 DE3-Gold cells as previously described[6,7]; an identical protocol was used to produce purified samples of I27 and HRAS. RNC constructs comprised an arrest-enhanced variant of the SecM stalling sequence, FSTPVWIWWWPRIRGPP, as previously described[6]. Purification of isolated FLN5 $A_3A_3$ was performed by affinity chromatography followed by size-exclusion chromatography in the presence of 6 M urea prior to buffer exchange into Tico buffer (10 mM Hepes, 30 mM $NH_4Cl$, 12 mM $MgCl_2$, 1 mM EDTA). The full protein sequence of the FLN5 $A_3A_3$ is deposited together with its chemical shift assignment on the BMRB (entry 51023). For the RDC, pulse-field gradient NMR (PFG-NMR) and PRE-NMR experiments, the additional mutation C747V (referred to FLN5 $A_3A_3$ V747) was introduced to yield a cysteine-less construct for site-specific spin labelling. The protein concentration was determined using the BCA assay according to the manufacturer's instructions. RNCs were expressed, isotopically labelled uniformly with [15]N, or site-specifically with [19]F, and purified as previously described[6,7]. For samples for intermolecular PRE-NMR experiments involving ribosome labelling, we generated modified *E. coli* BL21 strains with cysteine mutations in uL23 and uL24 using CRISPR as previously described[2]. RNC samples were prepared in Tico buffer for experiments. Western blot analyses were undertaken with an anti-hexahistidine horseradish peroxidase-linked antibody (Invitrogen, 1:5,000 dilution).

### Fluorescent and PEG-maleimide labelling of 70S and RNC samples

Ribosomes and RNCs were first reduced using 2 mM TCEP overnight at 277 K, then buffer exchanged into labelling buffer. For fluorescein-5-maleimide and PEG-maleimide, labelling was performed in Tico at pH 7.5. ABD-MTS labelling was performed in labelling buffer (50 mM HEPES, 12 mM $MgCl_2$, 20 mM $NH_4Cl$, 1 mM EDTA, pH 8.0). Samples were labelled using a 10x molar excess of ABD-MTS, or fluorescein-5-maleimide. Cysteine mass-tagging by PEGylation was performed as previously described with 10,000-fold molar excess of PEG over sample[7]. ABD-MTS and PEGylation reactions were analysed using 12% Bis-Tris SDS–PAGE gels[61]. The fluorescein-labelled reactions were run on a 20% Tricine SDS–PAGE gel, modified from ref. 62.

### NMR spectroscopy

All NMR experiments were recorded with Topspin 3.5pl2. NMR experiments of FLN5 $A_3A_3$ were performed in Tico buffer at pH 7.5 and 283 K. Chemical shifts were previously assigned[7] and obtained from data recorded on a Bruker Avance III operating at 700 and 800 MHz equipped with TCI cryoprobes. All samples contained 10% (v/v) $D_2O$ and 0.001% (w/v) DSS as a reference. Data were processed analysed using NMRPipe[63] (v11.7), CCPN[64] (v2.4) and MATLAB (R2017a, Mathworks).

Amide [1]H and [15]N chemical shifts were obtained from two-dimensional [1]H–[15]N SOFAST-HMQC experiments[65] using an acquisition time of 50 ms in the direct dimension. The inter-scan delay was 50 ms. Cα chemical shifts were obtained from 3D BEST-HNCA experiments recorded at 800 MHz with acquisition times of ~50 ms and inter-scan delays of 150 ms. C′ chemical shifts were obtained from BEST HNCO experiments recorded at 700 MHz using acquisition times of ~50 ms and inter-scan delays of 200 ms. RNC samples were doped with 20 mM NiDO2A (Ni(II) 1,4,7,10-tetraazacyclododecane-1,7-bis(acetic acid)) to enhance sensitivity[66]. Cosine-squared window functions were used in processing the spectra.

For PRE-NMR experiments, we used a cysteine-less construct with the C747V mutation and introduced six and eight labelling sites in the isolated and ribosome-bound protein, respectively. Samples were reduced overnight at 277 K in Tico supplemented with 2 mM TCEP. TCEP was then removed by buffer exchange into labelling buffer (50 mM HEPES, 12 mM $MgCl_2$, 20 mM $NH_4Cl$, 1 mM EDTA, pH 8.0) and subsequently labelled overnight at 277 K with 10× molar excess of MTSL. Following labelling, excess MTSL was removed by buffer exchanging the sample back into Tico buffer for NMR. The same labelling protocol was used for isolated protein and RNC samples. To measure the PREs, we recorded the signal intensities with MTSL in the paramagnetic and diamagnetic state. Direct measurements of relaxation rates proved not feasible for RNC samples due sensitivity limitations. 2D [1]H–[15]N SOFAST-HMQC experiments[65] were recorded at 800 MHz and 283 K using ~100 µM of protein or ~10 µM of RNC. Experiments were recorded with an acquisition time of 100 ms and 35 ms, in the direct and indirect dimension, respectively. The inter-scan delay was 450 ms to allow for complete relaxation. To acquire the diamagnetic data, the sample was reduced with 2.5 mM (RNC) or 100× molar excess (isolated) sodium ascorbate. Following complete reduction, the same HMQC experiment was recorded. To extract the PREs, spectral peaks were first fitted to a Lorentzian shape in both the direct and indirect dimension using NMRPipe[63]. Errors were obtained from the spectral noise (RMSE). From the fitted peaks, intensity ratios of $I_{para}/I_{dia}$ were calculated and converted to PRE rates for Bayesian ensemble reweighting by numerically solving equation S34 (see Supplementary Notes 3–4) for $\Gamma_2$. Sample integrity was monitored using interleaved [1]H,[15]N SORDID diffusion measurements as previously described[7].

PFG-NMR experiments were used to measure the diffusion coefficients and the $R_h$ of FLN5 variants. 1D [1]H,[15]N-XTSE diffusion measurements were recorded at 700 (FLN5, FLN5 Δ6) and 800 MHz (FLN5 $A_3A_3$). Eight to sixteen gradient strengths ranging linearly from 5% to 95% of the maximum gradient strength of 0.556 T m$^{-1}$ were used. By measuring the signal intensity at each gradient strength, diffusion coefficients could be obtained by fitting the data to the Stejskal–Tanner equation[67], which were converted to $R_h$ using the Stokes–Einstein equation.

RDCs for isolated FLN5 $A_3A_3$ were measured in Tico buffer at 283 K and pH 7.5 in a PEG/octanol mixture[68]. RDCs are reported as the splitting of the isotropic splitting subtracted from the aligned splitting, corrected for the negative gyromagnetic ratio of [15]N. RDCs were measured by preparing a solution containing 4.6% (w/w) pentaethylene glycol monooctyl ether ($C_8E_5$), 1-octanal (molar ratio 1-octanol:$C_8E_5$ = 0.94) and 110 µM of protein. Alignment was confirmed by measuring the $D_2O$ deuterium splitting at 283 K (17.6 Hz). All RDC NMR experiments were acquired on a Bruker Avance III HD 800 MHz spectrometer equipped with a TCI cryoprobe. A set of four different RDCs ($^1D_{NH}$, $^1D_{C\alpha CO}$, $^1D_{C\alpha H\alpha}$ and $^2D_{HNCO}$) was measured per sample (isotropic and anisotropic) using the 3D BEST HNCO (JCOH and JCC) or BEST-HNCOCA (JCAHA) experiments[69–71]. The one-bond [1]H–[15]N coupling was determined by recording two [15]N-HSQC sub-spectra, in-phase (IP) and anti-phase (AP). For the measurement of the [1]H–[13]CO coupling constants a BEST HNCO-JCOH experiment was used with an introduced DIPSAP filter. Such *J*-mismatch compensated DIPSAP spin-state filter offers an attractive approach for accurate measurement of small spin–spin coupling constants[72]. For that, three separate experiments were recorded with different filter lengths ($2\tau = 1/J$) for each anisotropic and isotropic media, where the sub-spectra associated to the separated spin states (two in phase and one anti-phase) are combined using a linear relation $k$ (IP) + ($k - 1$) (IP) ± (AP) with $k = 0.73$, the theoretical optimized scaling factor. The spectra were recorded with 144 × 104 × 1,536 complex points in the $^{13}C(t_1)/^{15}N(t_2)/^1H(t_3)$ dimensions, respectively, and with the spectral widths set to 15,244 Hz ([1]H), 2,070 Hz ([15]N) and 1,510 Hz ([13]C) for the HNCO-JCOH. For the HNCO-JCC and HNCOCA-JCAHA 256 × 200 × 1,536 complex points were acquired

in the $^{13}C(t_1)/^{15}N(t_2)/^1H(t_3)$ dimensions, with spectral widths of 15,244 Hz ($^1H$), 1,900 Hz ($^{15}N$) and 1,214 Hz/5050 Hz ($^{13}C$). The recycle delay was set to 200 ms, the acquisition time to 100 ms with 16 scans per increment, and the data was acquired in the non-uniform sampling format (2246 points for HNCO-JCOH and 7680 for the HNCO-JCC/HNCOCA-JCAHA experiments were sampled using the schedule generator from the web portal nus@HMS (http://gwagner.med.harvard.edu/intranet/hmsIST/). The time domain data was converted into the NMRPipe[63] format and reconstructed using the sparse multidimensional iterative lineshape-enhanced method (SMILE)[73]. Coupling constants were obtained from line splitting in the $^{13}C$ or $^{15}N$ dimension obtained with CCPN analysis software[64].

$^{19}F$ NMR experiments were recorded on a 500 MHz Bruker Avance III spectrometer equipped with a TCI cryoprobe at 298 K (unless otherwise indicated) using a 350 ms acquisition time and 1.5–3 s recycle delay as previously described[6]. We used an amber-suppression strategy to incorporate the unnatural amino acid tfmF, as previously described[6]. Multiple experiments were recorded in succession to monitor sample integrity over time also as previously described[6]. Data were processed using NMRPipe[63]. Spectra were baseline corrected, peaks were fit to Lorentzian functions and errors of the linewidths and integrals (that is, populations) were estimated using bootstrapping (200 iterations, calculating the standard error of the mean), or from the spectral noise for states whose resonance was not detectable, in MATLAB[6]. $^{19}F$-translational diffusion experiments were performed as previously described[6].

Thermodynamic parameters of folding ($\Delta H$, $\Delta S$ and $\Delta C_p$) were obtained from a nonlinear fit to a modified Gibbs–Helmholtz equation, assuming $\Delta C_p$ remains constant across the experimental temperature range:

$$\ln(K_{eq,T}) = -\left(\frac{\Delta H_{T_0} + \Delta C_p(T - T_0)}{R}\right)\left(\frac{1}{T}\right) + \left(\frac{\Delta S_{T_0} + \Delta C_p\ln\left(\frac{T}{T_0}\right)}{R}\right) \quad (S1)$$

$K_{eq}$ is the equilibrium constant, $T$ is the temperature in Kelvin and $T_0$ is the standard temperature (298 K). We also fitted the data to the linear van't Hoff equation (assuming $\Delta C_p = 0$).

$$\ln(K_{eq}) = -\left(\frac{\Delta H}{R}\right)\left(\frac{1}{T}\right) + \frac{\Delta S}{R} \quad (S2)$$

The Scipy package with optimize.curve_fit function was used to perform the fits[74] and errors were estimated as one s.d. from the diagonal elements of the parameter covariance matrix. All parameters ($\Delta H$, $\Delta S$, $\Delta C_p$) generally showed strong correlations with each other ($r \geq 0.8$), and thus, their uncertainties correlate also. These parameter correlations are expected[75]. The magnitudes of $\Delta H$ and $-T\Delta S$ are also expected to correlate because we study the temperature dependence of folding in a range where $\Delta G$ of folding is close to 0.

Folding free energies were calculated from the experimental populations using $\Delta G = -RT\ln(K)$. The folding free energy of the FLN5+67 wild-type RNC, where no unfolded state is observable, was estimated on the basis of two destabilizing mutants FLN5(V664A/F665A) and FLN5(V707A). The stability of these mutants was measured using $^{19}F$ NMR on and off the ribosome. The FLN5+67 wild-type folding energy ($\Delta G_{N-U}$) was then calculated as the average from the V664A/F665A and V707A mutants using $\Delta G_{WT,+67} = \Delta G_{mut,+67} - \Delta\Delta G_{mut-WT,iso}$, where $\Delta\Delta G_{mut-WT,iso}$ is the experimentally measured destabilization in isolation. Given that at FLN5+34, both mutants show a weaker destabilization than in isolation, we reasoned that this estimate of the FLN5+67 wild-type folding free energy is its lower bound (most negative).

$^{19}F$ transverse relaxation rate ($R_2$) measurements were recorded using a Hahn-echo sequence and acquired as pseudo-2D experiments with relaxation delays of 0.1 to 200 ms. Data were processed using NMRPipe

and analysed using MATLAB. Data were fit to lineshapes and $R_2$ was obtained by fitting the integrals to single exponential functions. We also orthogonally determined $R_2$ from linewidth measurements of spectra acquired by 1D $^{19}F$ pulse-acquire experiments, which showed excellent correlations. The lineshape-derived $R_2$ values also showed good correlation with previously determined rotational correlation times[25] ($\tau_C$). We additionally determined the $S^2\tau_C$ of FLN5 in 60% glycerol at 278 K by measurements of triple quantum build-up and single quantum relaxation as previously described[76,77]. Thus, our $R_2$ values can be used to determine rotational correlation times ($\tau_{C,exp}$). The obtained $\tau_{C,exp}$ was used to estimate the bound population as $\frac{\tau_{C,exp} - \tau_{C,iso}}{\tau_{C,bound} - \tau_{C,iso}}$, where $\tau_{C,iso}$ is the rotational correlation time of the isolated protein[25] (7.7 ns at 298 K) and $\tau_{C,bound}$ is the expected rotational correlation time of the bound state. $\tau_{C,bound}$ is taken as the rotational correlation time of the ribosome itself (~3,000 ns at 298 K) for a fully rigid bound state. From the bound populations ($p_B$), the resulting change in the folding free energies of the intermediates was calculated as $\Delta\Delta G_{I-U,RNC-iso} = RT(\ln(1 - p_B))$. We report the estimate for a fully rigid bound state in the main text ($S^2_{bound} = 1.0$) but note that even one order of magnitude more flexibility in the bound state ($S^2_{bound} = 0.1$) only accounts for up to $1.1 \pm 0.6$ and $0.4 \pm 0.1$ kcal mol$^{-1}$ of stabilization for I1 and I2 on the ribosome at FLN5+47, respectively. These estimates still cannot account for the >4 kcal mol$^{-1}$ of intermediate stabilization observed on the ribosome[6].

All NMR experiments of RNCs are recorded and continuously interleaved with a series of 1D $^1H/^{19}F$ spectra and $^1H,^{15}N/^{19}F$ diffusion measurements[6,7,61,78]. These provide the most sensitive means to assess changes in the sample, and when alterations in signal intensities or linewidths (that is, transverse relaxation rates), chemical shifts or translational diffusion measurements of the nascent chain are observed, data acquisition is halted. Only data corresponding to intact RNCs are summed together and subjected to a final round of analysis. Where signal-to-noise remains low, datasets from multiple samples are compared to ensure identical spectra, before summation together into a single NMR spectrum. Biochemical assays provide an orthogonal means to assess nascent chain attachment to the ribosome. Identical samples incubated in parallel with NMR samples are analysed by SDS–PAGE (under low pH conditions[61]) and detected with nascent-chain-specific antibodies. Ribosome-bound species migrate with an addition ~17-kDa band-shift relative to released nascent chains due to the presence of the tRNA covalently linked to the nascent chain. Combined with time-resolved NMR measurements, these analyses confirm that the reported NMR resonances originate exclusively from intact RNCs.

## Mass spectrometry

FLN5 A$_3$A$_3$ was buffer exchanged into 100 mM (NH$_4$)$_2$CO$_3$ at pH 6.8 (using formic acid for pH adjustment). Analyses were run on the Agilent 6510 QTOF LC–MS system at the UCL Chemistry Mass Spectrometry Facility. Samples contained ~10–20 µM of protein and 10 µl were injected onto a liquid chromatography column (PLRP-S, 1,000 Å, 8 µm, 150 mm × 2.1 mm, maintained at 60 °C). The liquid chromatography was run using water with 0.1% formic acid as mobile phase A and acetonitrile with 0.1% formic acid as phase B with a gradient elution and a flow rate of 0.3 ml min$^{-1}$. ESI mass spectra were continuously acquired. The data were processed to zero charge mass spectra with the MassHunter software, utilizing the maximum entropy deconvolution algorithm.

## Small-angle X-ray scattering

We measured SAXS of an isolated FLN5 A$_3$A$_3$ C747V sample in Tico buffer supplemented with 1% (w/v) glycerol. Data collection was performed at the DIAMOND B21 beamline (UK)[79] with a beam wavelength of 0.9408 Å, flux of $4 \times 10^{12}$ photons s$^{-1}$ and an EigerX 4 M (Dectris) detector distanced at 3.712 m from the sample. A capillary with a 1.5 mm

diameter kept at 283 K was used for data acquisition. We acquired SAXS data at multiple protein concentrations (5.5, 2.75, 1.38, 0.69, 0.34 and 0.17 mg ml$^{-1}$) to assess whether the sample exhibited signs of aggregation or interparticle interference. At 5 mg ml$^{-1}$, we observed weak signs of interparticle interference in the low $q$ region of the scattering profile, which is also reflected in the $R_g$ obtained by Guinier analysis (using the autorg tool from ATSAS[80]; Supplementary Table 1). Data were recorded as a series of frames, non-defective frames were averaged, and buffer subtracted with PRIMUS[80]. Size-exclusion chromatography–SAXS (SEC–SAXS) experiments were additionally performed in Tico buffer with 1% (w/v) glycerol using a KW402.5 (Shodex) column to confirm the monodispersity of the sample. We chose the 2.75 mg ml$^{-1}$ dataset as the final dataset to compare with our molecular dynamics simulations. This dataset exhibited the highest signal to noise ratio and did not show signs of interparticle interference, and accordingly, the $R_g$ obtained from the 2.75 mg ml$^{-1}$ dataset is consistent with the value obtained from lower concentrations and the main SEC–SAXS peak (Supplementary Table 1).

## Circular dichroism spectroscopy

The circular dichroism (CD) spectrum of isolated FLN5 A$_3$A$_3$ V747 was acquired in 10 mM Na$_2$HPO$_4$ pH 7.5 at 283 K. A Chirascan-plus CD spectrometer (Applied Photophysics), a protein concentration of 44 μM and a cuvette with a 0.5 cm pathlength were used.

## HRAS refolding experiments

HRAS refolding experiments were performed with the HRAS G-domain (residues 1–166). The protein was unfolded overnight at 298 K in Tico buffer with 2 mM β-mercaptoethanol, 8 M urea and protein concentration of 15 μM. The protein was then refolded by rapidly diluting into Tico buffer (supplemented with 2 mM β-mercaptoethanol and 50 μM GDP) to reach final urea and protein concentrations of 0.94 M and 1.76 μM, respectively, and allowed to incubate at 298 K for 24 h. For NMR analyses of refolded samples, we prepared 18 μM of refolded protein with the same urea concentrations and dilutions.

We assayed the functional/activity state of HRAS using GDP/GTP nucleotide exchange ('activity') assay[81] with fluorescently labelled GTP that exhibits higher fluorescence when bound to HRAS than free in solution (BODIPY FL GTP, ThermoFisher). 0.4 μM of HRAS, 0.01 μM of BODIPY GTP and 1 μM of SOS$_{cat}$ (the catalytic domain of Son of sevenless) were incubated at room temperature and the maximum (plateau) fluorescence recorded and normalized by the signal of the buffer (signal/noise ratio). SOS$_{cat}$ was produced as previously described[82]. Fluorescence measurements were performed using the CLARIOstar microplate reader (BMG Labtech) with excitation and emission wavelengths set to 488 and 514 nm, respectively.

The proteolytic stability of HRAS was assayed with thermolysin at a concentration of 0.05 mg ml$^{-1}$ incubated with HRAS samples over the course of 5 h in vitro and 9 h in rabbit reticulocyte lysate (RRL, TNT coupled reticulocyte lysate, Promega). Reactions were quenched with 23 mM EDTA. Timepoints were analysed by western blot analysis using a pan-RAS polyclonal antibody (ThermoFisher, 1:1,000 dilution), utilizing an anti-rabbit IgG horseradish peroxidase-linked secondary antibody (Cell Signaling Technology, 1:1,000 dilution). Densitometry analyses were performed with ImageJ[83]. For the RRL experiments, refolding reactions were performed in RRL for 24 h at 298 K and a final HRAS concentration of 1.6 μM followed by pulse proteolysis and we quantified the relative band intensities (refolded/control) for each time point to account for increased background on the western blot during the proteolysis reaction.

## Molecular dynamics simulations

We used the FLN5 A$_3$A$_3$ C747V sequence for all simulations. A reliability and reproducibility checklist is provided in Supplementary Table 8. GROMACS (version 2021)[84] was used for all all-atom molecular dynamics

simulations in explicit solvent. We employed the Charmm36m force field in combination with the CHARMM TIP3P water model (C36m) and the CHARMM TIP3P water model with an increased water hydrogen LJ well-depth (denoted here as C36m+W)[85]. We also used the a99sb-disp force field together the a99sb-disp TIP4P-D water model[86]. Default protonation states were used in all cases. Starting from a random extended conformation, for all force field combinations the system was solvated in a dodecahedron box with 151,135 water molecules and 12 mM MgCl$_2$ (resulting in 455,116 atoms and an initial box volume of 4,688 nm$^3$). Systems were then energy minimized using the steepest-decent algorithm. For the following dynamics simulations, we used the LINCS algorithm[87] to constrain all bonds connected to hydrogen and a timestep of 2 fs using the leap-frog algorithm for integration. Nonbonded interactions were calculated with a cut-off at 1.2 nm (including a switching function at 1.0 nm for van der Waals interactions) and the particle mesh Ewald (PME) method[88] was used for long-range electrostatic calculations. We then equilibrated the systems in two phases. First, we performed a 500 ps equilibration simulation in the NVT ensemble with position restraints on all protein heavy atoms. The temperature was kept at 283 K using the velocity rescaling algorithm[89] and a time constant of 0.1 ps. Next, we further equilibrated the systems for 500 ps in the NPT ensemble at 283 K and a pressure of 1 bar with a compressibility of $4.5 \times 10^{-5}$ bar$^{-1}$ using the Berendsen barostat[90]. Following equilibration, we relaxed our initial structure for 100 ns at 283 K without any position restraints using the Parrinello–Rahman algorithm[91] and then picked five structures from this simulation for production simulations. We ran a total of 5× 2 μs (with different initial coordinates and velocities) yielding a total of 10 μs of sampling per force field. For the C36m+W combination we ran an additional 5× 2 μs starting from 5 new starting structures yielding 20 μs in total.

We also generated a prior ensemble with a physics-based coarse-grained (C-alpha) model. We generated the C-alpha model template from the FLN5 crystal structure using SMOG 2.3[92], where all bonded terms have a global energy minimum at the values taken in the crystal structure[93]. Nonbonded van der Waals interactions were modelled using a 10–12 Lennard-Jones potential with $\sigma$ and $\lambda$ parameters described in the M1 parameter determined by Tesei et al.[94] (equation (S3)). We used the arithmetic mean of two residues to determine $\sigma$ and $\lambda$. Electrostatic interactions were modelled using the Debye–Hückel theory with parameters described previously[7]. Interactions between Cα beads separated by less than four residues were excluded. We ran initial simulations at a range of reduced temperatures to determine the effect on the average compactness and ran final simulations at a reduced temperature of 1.247 (150 K in GROMACS) as we did not observe a significant increase in average $R_g$ beyond this temperature. Simulations were run for a total of $3 \times 10^9$ steps with GROMACS (v2018.3).

$$u_{LJ} = \sum_i^N \lambda \left[ 5\left(\frac{\sigma}{r}\right)^{12} - 6\left(\frac{\sigma}{r}\right)^{10} \right] \quad (S3)$$

After simulations, the coarse-grained ensemble was backmapped to an all-atom structure using PULCHRA (v3.06)[95].

RNC simulations were parameterized using the C36m+W force field/water model combination[85,96]. We modelled the ribosome using the structure PDB 4YBB[97] as a template, which we previously refined against a cryo-EM map containing an FLN5 RNC[98]. As in our previous work, we only retained ribosome atoms around the nascent chain exit tunnel and accessible surface outside the vestibule[7]. The FLN6 linker and SecM sequence were initially modelled using a cryo-EM map of a FLN5+47 RNC (Mitropoulou et al., manuscript in preparation). The rest of the nascent chain (MHHHHHAS N-terminal tag and FLN5) was then built using PyMol version 2.3 (The PyMol Molecular Graphics System, Schrödinger) and we generated a random initial starting structure with a short simulation using a structure-based force field, SMOG2.3[92], without native contacts. The FLN5+31 A$_3$A$_3$ RNC (containing the

C747V mutation) complex was then centred in a dodecahedral box, solvated using 1,030,527 water molecules and neutralized with 706 $Mg^{2+}$ ions, resulting in a final system size of 3,163,127 atoms. The initial box volume was 32,117 $nm^3$. The large box size was necessary to accommodate the highly expanded unfolded state. We then used the same cut-offs and simulation methods as for the isolated protein. We initially also ran a 500 ps equilibration simulation in the NVT ensemble using position restraints on all heavy atoms using a force constant of 1,000 kJ $mol^{-1}$ $nm^2$ in along the $x$, $y$ and $z$ axes. We used a temperature of 283 K, which was held constant using the velocity rescaling algorithm[89] and a time constant of 0.1 ps. Then, we ran a 500 ps equilibration simulation in the NPT ensemble at 283 K using the same position restraints. The pressure was kept at 1 bar with a compressibility of $4.5 \times 10^{-5}$ $bar^{-1}$ using the Berendsen barostat[90]. The position restraints for all nascent chain atoms (except the terminal residue at the PTC in the ribosome) were then removed, while the ribosome atoms kept being position restrained. In this setup, we ran a 1 ns equilibration simulation at 283 K and 1 bar, using the Parrinello–Rahman algorithm[91]. All production simulations were performed using position restraints for the ribosome atoms and C-terminal nascent chain residue at the PTC. Using the equilibrated configuration, we then ran two simulations of ~100 ns to picked ten starting structures for production simulations. Then, ten production simulations of 1.5 μs each (15 μs) were initiated from these different starting structures using random initial velocities. Before the production simulation, each structure was re-equilibrated at 283 K and 1 bar with a 500 ps NVT and 500 ps NPT simulation.

Lastly, to compare our C36m+W simulations with a model that only considers steric exclusion as a nonbonded interaction, we also ran simulations of a simple all-atom model, based on a structure-based model template[92]. We used the FLN5 crystal structure to define the energy minima of all bond and dihedral angles and removed all native contacts. Simulations of isolated and ribosome-bound $FLN5A_3A_3$ were run for $1 \times 10^9$ steps and 100,000 frames were harvested for analysis. This ensemble was used to compare the expansion of the ensemble, ribosome interactions and conformational entropy with the C36m+W simulations.

## Calculation of PREs

The transverse PRE rates of backbone amide groups, $\Gamma_2$, were back-calculated from the ensembles using the Solomon–Bloembergen equation[99,100]

$$\Gamma_2 = \frac{1}{15}\left(\frac{\mu_0}{4\pi}\right)^2 \gamma_H^2 g_e^2 \mu_B^2 S(S-1)[4J(0) + 3J(\omega_H)] \tag{S4}$$

where $\mu_0$ is the permeability of space, $\gamma_H$ is the gyromagnetic ratio of the proton, $g_e$ is the electron g-factor, $\gamma_B$ is the Bohr magneton, $S$ is the proton nuclear spin and $J(\omega_0)$ is the generalized spectral density function. For flexible spin labels attached via rotatable bonds the spectral density can be expressed as in equation (S5)[101].

$$J(\omega_H) = \langle r^{-6}\rangle \left[\frac{S^2 \tau_c}{1+(\omega_H \tau_c)^2} + \frac{(1-S^2)\tau_t}{1+(\omega_H \tau_t)^2}\right] \tag{S5}$$

where $\langle r^{-6}\rangle$ is the average of the electron–hydrogen distance ($r$) distribution, $S^2$ is the generalized order parameter for the electron–hydrogen interaction vector, $\tau_c$ is the correlation time defined in terms of the rotational correlation time of the protein ($\tau_r$) and the electron spin relaxation time ($\tau_s$):

$$\tau_c = (\tau_r^{-1} + \tau_s^{-1})^{-1} \tag{S6}$$

$\tau_t$ is the total correlation time defined as:

$$\tau_t = (\tau_r^{-1} + \tau_s^{-1} + \tau_i^{-1})^{-1} \tag{S7}$$

$\tau_i$ is the internal correlation time of the spin label. Since for nitroxide labels electron spin relaxation occurs on a much slower timescale than rotational tumbling[101,102], $\tau_C$ can be approximated to $\tau_r$, such that expression for $\tau_t$ simplifies to

$$\tau_t = (\tau_C^{-1} + \tau_i^{-1})^{-1} \tag{S8}$$

Given that $\tau_c$ is not known a priori, we iteratively scanned $\tau_c$ values in the range of 1 to 15 ns to find a value for which optimal agreement with the experimental data is achieved (as judged by the reduced $\chi^2$)[94,103]. The spin label correlation time[104], $\tau_i$ was set to 500 ps, in agreement with molecular dynamics simulations[105] and electron spin resonance measurement[106].

The generalized order parameter $S^2$ for the electron–hydrogen interaction vector can be decomposed into its radial and angular components[107]:

$$S_{PRE}^2 \approx S_{PRE,angular}^2 S_{PRE,radial}^2 \tag{S9}$$

where the individual components are defined as

$$S_{PRE,angular}^2 = \frac{4\pi}{5} \sum_{m=-2}^{2} |\langle Y_2^m(\Omega^{mol})\rangle|^2 \tag{S10}$$

$$S_{PRE,radial}^2 = \langle r^{-6}\rangle^{-1}\langle r^{-3}\rangle^2 \tag{S11}$$

and $Y_2^m$ are the second order spherical harmonics and $\Omega^{mol}$ are the Euler angles in the frame. A weighted ensemble average of $S^2$ can be calculated by taking a weighted ensemble average of the individual radial and angular components.

A previously published rotamer library containing 216 MTSL rotamers[108] was used to explicitly model the flexibility of the spin label, similar to other existing methods[109,110]. The rotamer library was aligned to all employed labelling sites for each conformer using the backbone atoms of the labelling site and Cys-MTSL moiety. Clashing rotamers were discarded, where a steric clash between the rotamer and the protein was defined using a 2.5 Å cut-off distance. Only backbone and Cβ atoms were considered for the protein, assuming sidechains can rearrange to accommodate the MTSL rotamer[111]. For MTSL, only the sidechain was included (heavy atoms beyond the Cβ atom). Protein frames for which at least one labelling position cannot sterically allow any MTSL rotamers were discarded. The rotamer library was used to calculate a weighted ensemble-averaged $\Gamma_2$ over the rotamer ensemble for each protein conformer in the protein ensemble using equations (S3–S11). The protein ensemble average can then be calculated by averaging $\Gamma_2$ over the ensemble.

PRE intensity ratios were then calculated from the ensemble-averaged PRE rate, $\langle \Gamma_2 \rangle$, using

$$\frac{I_{para}}{I_{dia}} = \frac{R_2 e^{-2\Delta\langle\Gamma_2\rangle}}{R_2 + \langle\Gamma_2\rangle} \times \frac{R_{2,MQ}}{R_{2,MQ} + \langle\Gamma_2\rangle} \tag{S12}$$

where $R_2$ is the linewidth in the proton dimension (residue-specific), $R_{2,MQ}$ is the linewidth in the nitrogen dimension (multiple-quantum term) and $\Delta$ is the delay time in the HMQC experiment (5.43 ms). See Supplementary Note 3 for additional details.

For RNCs, we considered that that ribosome tethering may increase the correlation time of the electron–amide interaction vector due to restricted molecular tumbling near the exit tunnel. We therefore calculated an order parameter, $S_{NC}^2$, which quantifies the motion of the electron-interaction vector over the entire nascent chain conformer ensemble ($S_{NC}^2$ is distinct from the order parameter $S^2$ that quantifies the motion of the MTSL rotamer library attached to a labelling site for a specific protein conformer; equation (S9)). $S^2$ is given by

$$S^2_{NC} \approx S^2_{NC,angular} S^2_{NC,radial} \tag{S13}$$

where $S^2_{NC,angular}$ and $S^2_{NC,radial}$ are given by

$$S^2_{PRE,angular} = \frac{4\pi}{5} \sum_{m=-2}^{2} |\langle Y_2^m(\Omega^{mol}) \rangle|^2 \tag{S14}$$

$$S^2_{PRE,radial} = \langle r^{-6} \rangle^{-1} \langle r^{-3} \rangle^2 \tag{S15}$$

and $Y_2^m$ are the second order spherical harmonics and $\Omega^{mol}$ are the Euler angles in the frame. We approximated the position of the free electron with the Cα atom of the labelling site in this case. A $S^2_{NC}$ value of 0 indicates that the vector tumbles completely independent of the ribosome and that the correlation time of the electron–amide vector is the same as for the isolated protein, $\tau_{C,iso}$. A $S^2_{NC}$ value of 1 means that the vector tumbles with the same rotational correlation time as the ribosome ($\tau_{r,70S}$ = 3.3 μs per cP, as determined by fluorescence depolarization[112], and $\tau_{r,70S}$ = 4.3 μs at 283 K in $H_2O$). The effective correlation time, $\tau_{C,eff}$, of each amide-electron vector is given by

$$\tau_{C,eff} = S^2_{NC}\tau_{r,70S} + (1 - S^2_{NC})\tau_{C,iso} \tag{S16}$$

We used a value of 3 ns for $\tau_{C,iso}$, which was the optimal value determined for isolated FLN5 $A_3A_3$. Generally, $\tau_C$ (equation (S6)) is approximated as $\tau_C \approx \tau_r$ because the electron spin relaxation time, $\tau_s$, occurs on a much slower timescale. In fact, measurements of the spin relaxation time of nitroxides have been measured to be on a timescale from hundreds of nanoseconds to several microseconds[113–115]. The calculated values of $\tau_{C,eff}$ are predominantly below 100 ns except for labelling sites C744, uL23 G90C and uL24 N53C, where values of up to ~250 ns are observed (Supplementary Tables 5 and 6). Thus, we still expect $\tau_C$ to be dominated by $\tau_r$ and make use of the $\tau_C \approx \tau$ approximation.

Finally, reference PRE profiles for a fully extended peptide were calculated from a linear polyalanine chain using a $\tau_C$ of 5 ns and $R_{2,H}/R_{2,MQ}$ of 100 Hz.

## Bayesian inference reweighting

We performed ensemble refinement by reweighting the molecular dynamics-derived ensembles against the experimentally deduced $\Gamma_2$ rates using the Bayesian Inference of Ensembles (BioEn) software and method described in the corresponding paper[116,117]. These calculations were performed using in-house scripts of the software with the modification to incorporate upper and lower bound restraints in addition to regular restraints with gaussian errors. To this end, these inequality restraints were treated as normal gaussian restraints but subjected to a conditional statement. Lower bound restraints ($\Gamma_2 > 64.5$ s$^{-1}$ for isolated FLN5 $A_3A_3$; $\Gamma_2 > 96.0$ s$^{-1}$ for the RNCs) were applied only if the back-calculated $\Gamma_2$ was below the lower bound value. Similarly, upper bound restraints ($\Gamma_2 < 2.2$ s$^{-1}$ for isolated FLN5 $A_3A_3$; $\Gamma_2 < 3.7$ s$^{-1}$ for the RNCs) were applied only if the back-calculated average was above the upper bound. This effectively allows the back-calculated value to vary freely above the lower bound and below the upper bound but imposes a penalty if the inequality condition is not met. The errors of the lower and upper bound values were taken as the combined relative error of that datapoint (that is, the intensity ratio).

As described by Köfinger et al.[117], the reweighting optimization problem can be efficiently solved by minimizing the negative log-posterior function ($L$).

$$L = \theta S_{KL} + \sum_{i=1}^{M} \frac{\left(\sum_{\alpha=1}^{N} w_\alpha y_i^\alpha - Y_i\right)^2}{2\sigma_i^2} \tag{S17}$$

$\theta$ expresses the confidence in the initial ensemble, $N$ is the ensemble size, $M$ is the number of experimental restraints, $w_\alpha$ is the vector of weights for the conformers in the ensemble, $y_i^\alpha$ is the back-calculated experimental value $i$, $Y_i$ is the experimental restraint $i$, $\sigma_i$ is the uncertainty of experimental restraint $i$, and $S_{KL}$ is the Kullback–Leibler divergence defined as

$$S_{KL} = \sum_{\alpha=1}^{N} w_\alpha \ln\frac{w_\alpha}{w_\alpha^0} \tag{S18}$$

$w_\alpha^0$ is the vector of initial weights (which were uniform). We used the log-weights method to minimize the negative log-posterior[117] and performed reweighting calculations for a range of $\theta$ values, as the optimal value of $\theta$ cannot be known a priori. Therefore, we conduct L-curve analysis[117,118] by plotting $S_{KL}$ (entropy) on the $x$ axis and the goodness of fit, quantified by the reduced $\chi^2$ value, on the $y$ axis. The reduced $\chi^2$ was calculated against the experimental intensity ratios ($I_{para}/I_{dia}$). This is an effective method to prevent overfitting and introducing a minimal amount of bias into the prior ensemble[117,119]. After reweighting, we also calculated the effective fraction of frames contributing to the ensemble average[119] as an indication of the extent of fitting.

$$N_{eff} = \exp(-S_{KL}) \tag{S19}$$

For RNCs, we used the same approach with an additional modification. Since the PRE depends on $\tau_{C,eff}$ and $S^2_{NC}$ which are a function of the weights of individual structures in the ensemble, this consequently leads to changes in $\tau_{C,eff}$ and $S^2_{NC}$ when reweighting is performed. Therefore, the conformer-specific PRE values that were used for reweighting are not the same anymore after reweighting. To account for this, we performed 20 iterative rounds of reweighting where each additional round receives input weights and $\tau_{C,eff}$ from the previous round. We found that this leads to convergence of the weights and conformer-specific PREs.

We found that for the ribosomal labelling sites, uL23 G90C and uL24 N53C, the reweighting results are sensitive to the specific ribosome structure used to fit the MTSL rotamer library to, since small variations in the local structure of the labelling site can lead to different rotamer distributions. We tested two different rotamer distributions for the ribosomal labelling sites (Extended Data Fig. 5a), finding that one of them (referred to as R2) gives better agreement with the intermolecular PRE data after reweighting and fits better into the expected density or MTSL rotamers when rotamers are fitted to ten high-resolution ribosome structures (Extended Data Fig. 5a). The R2 rotamer distribution is more representative of the expected variation from structural changes in the labelling sites and was therefore used for our final reweighting calculations.

## Calculation of RDCs, $R_h$ and chemical shifts from molecular dynamics simulations

To back-calculate the $R_h$ from static structures we used an approximate relationship between $R_g$ and $R_h$ values[120], the latter being calculated from the programme HYDROPRO[121]. Thus, we calculated the $R_g$ from Cα atoms using MDAnalysis[122] and then converted it to $R_h$ using

$$R_h = \frac{R_g}{\frac{\alpha_1(R_g - \alpha_2 N^{0.33})}{N^{0.60} - N^{0.33}} + \alpha_3} \tag{S20}$$

$N$ is the number of amino acids, $\alpha_1$ takes a value of 0.216 Å$^{-1}$, $\alpha_2$ takes a value of 4.06 Å, and $\alpha_3$ has a value of 0.821. The estimated value of $R_h$ (relative to the HYDROPRO calculation) has an average relative uncertainty[120] of 3%. HYDROPRO itself has a relative uncertainty of ±4% with respect to experimental values[121]. Therefore, we treat the back-calculated ensemble-average $R_h$ with a total relative uncertainty

of ±5%. The ensemble average was calculated as previously described by Ahmed et al. for back-calculation of PFG-NMR derived values[123] of $R_h$.

$$\langle R_h \rangle = \ln(\langle \exp(-R_h^{-1}) \rangle)^{-1}. \tag{S21}$$

Chemical shifts were calculated using the SHIFTX2 software[124] and RDCs were calculated using the global alignment prediction method implemented in PALES[125]. We then scaled the magnitude (that is, the extent of alignment) of the calculated RDCs by a global factor to optimize the Q-factor for each ensemble.

## Calculation of SAXS profiles from molecular dynamics simulations

We used Pepsi-SAXS[126] to compute the theoretical scattering profiles of each conformer in the molecular dynamics ensembles. We treated the contrast of the hydration layer ($\delta_p$) and the effective atomic radius ($r_0$) as global parameters and used values of 3.34 e$^-$ nm$^{-3}$ and $1.025 \times r_m$ ($r_m$ = average atomic radius of the protein) in line with previous work that showed these parameters to well suited for flexible proteins[127]. The constant background and scale factor were also fitted globally using least-squares regression[103,127]. The goodness of fit was assessed using the reduced $\chi^2$ metric, where $n$ is the number of datapoints, $q$ is the scattering angle, $I_q^{calc}$ and $I_q^{exp}$ are the calculated and experimental scattering intensities, respectively, and $\sigma_q$ is the experimental error:

$$\chi_r^2 = \frac{1}{n} \sum_q^n \frac{(I_q^{calc} - I_q^{exp})^2}{\sigma_q^2} \tag{S22}$$

## Structural analysis

The Python package MDAnalysis[122] and MDTraj[128] were used for general analysis of the ensembles involving atomic coordinates. For native contact analysis, we calculated the fraction of native contacts (relative to the native FLN5 crystal structure) as[129]

$$Q(X) = \frac{1}{N} \sum_{ij} \frac{1}{1 + e^{(\beta(r_{i,j} - \lambda r_{i,j}^0))}} \tag{S23}$$

where $r_{i,j}$ and $r_{i,j}^0$ are the distances between atoms $i$ and $j$ in frame $X$ and the template structure, respectively, $\beta$ modulates the smoothness of the switching function (default value 5 Å$^{-1}$ used) and $\lambda$ is a factor allowing for fluctuations of the contact distance (default value 1.8 used).

Asphericity was calculated using MDAnalysis[122] as defined by Dima and Thirumalai[130]:

$$\Delta = \frac{3}{2} \frac{\sum_{n=1}^{3} (\lambda_i - \bar{\lambda})^2}{trT^2} \tag{S24}$$

$\bar{\lambda}$ represents the mean eigenvalue obtained from the inertia tensor $\bar{\lambda} = \frac{trT}{3}$.

For the intrachain contact analysis, we defined contacts between Cα−Cα distances of less than 10 Å. The contact features qualitatively were unchanged when using lower cut-off values or when calculating contacts between all heavy atoms. Secondary structure populations were calculated using DSSP[131] implemented in MDTraj. The SASA was calculated using GROMACS[84]. Clustering was also performed in GROMACS using the GROMOS algorithm[132] and Cα RMSD cut-offs in the range of 1.2–1.8 nm.

## Error analysis from ensembles

Errors from the molecular dynamics ensembles were estimated using a block analysis of the full concatenated ensembles (composed of multiple statistically independent simulations). We performed block analysis for the concatenated ensembles to verify that the estimate of the standard error of the mean (s.e.m.) plateaus/fluctuates at block sizes

larger than blocks corresponding to the individual trajectories. The final block size was chosen either in the plateau region of block analysis plots or corresponding to the blocks of the statistically independent simulation (10 independent simulations were run for the isolated and RNC systems with the C36m+W force field, and thus 10 blocks were chosen for block analysis and error estimation). The error after reweighting with PRE-NMR data was calculated the same way using a weighted standard error, where blocks are weighted according to the weights obtained from reweighting with PRE-NMR data. Exemplar block analysis plots are shown in Supplementary Fig. 8.

## Energetic analyses from structural ensembles

The conformational entropy was calculated as defined by Baxa et al.[133]. Proline, glycine and alanine entropies were calculated from the backbone probability distribution $P_i(\Phi,\Psi)$. Residues with a maximum of two sidechain torsion angles, $X_n$, the entropy was calculated from the probability distribution $P_i(\Phi,\Psi,X_1,X_2)$, while residues with more sidechain torsion angles was calculated from the sum of entropies obtained using the $P_i(\Phi,\Psi,X_1)$, and $P_i(X_n)$, after subtraction of the entropy obtained from $P_i(X_1)$. Entropies were calculated from probability distributions using $S = -k_B \sum_{i=1}^{n} P_i \ln(P_i)$. We used a block analysis from the pooled ensembles (i.e., all individual trajectories concatenated together) to check that the entropy difference between on and off the ribosome is robust with respect to sampling by calculating entropy changes with increasing amounts of total sampling (from the 15 and 20 µs of concatenated sampling for the RNC and isolated protein, respectively). The errors were then also estimated from the same sampling/block sizes up to 7.5 µs of molecular dynamics sampling. This is because the estimate of entropy differences trend increases up to total sampling times of 7.5 µs (Extended Data Fig. 6g).

The energetic contributions due to changes in solvation were estimated based on empirical relationships between changes in the polar and apolar accessible surface area[75,134] ($\Delta ASA_{polar}$ and $\Delta ASA_{apolar}$). The apolar and polar surface area of the protein were defined based on the atomic partial charges in the C36m force field[85]. Atoms with an absolute charge of less than or equal to 0.3 were defined as apolar. The change in heat capacity of hydration is related to these quantities by

$$\Delta C = \Delta C_{apolar} + \Delta C_{polar} = \alpha \times \Delta ASA_{apolar} + \beta \times \Delta ASA_{polar} \tag{S25}$$

where $\alpha$ and $\beta$ are 0.34 ± 0.11 and −0.12 ± 0.12 cal mol$^{-1}$ K$^{-1}$ Å$^{-2}$, respectively. We obtained these values as an average and standard deviation of parameters previously reported in the literature as summarized in ref. 135 to account for the uncertainty of the parameters in addition to the uncertainty coming from conformational sampling in our simulations. The enthalpy change due to solvation is then obtained from[75]

$$\Delta H_{solv}(333 \text{ K}) = \gamma \times \Delta ASA_{apolar} + \delta \times \Delta ASA_{polar} \tag{S26}$$

$$\Delta H_{solv}(T) = \Delta H_{solv}(333 \text{ K}) + \Delta C(T - 333 \text{ K}) \tag{S27}$$

$T$ is the temperature and $\gamma$ and $\delta$ constants taking on values of −8.44 and 31.4 cal mol$^{-1}$ Å$^{-2}$, respectively. While we are not aware of alternative parameter sets for the solvation enthalpy (equation (S26)) in the literature, we treated these parameters with a relative uncertainty of 50% to show that even with such high levels of uncertainty our conclusions are not affected. Finally, the solvation entropy and change in free energy are then calculated using

$$\Delta S_{333 \text{ K, solv}} = \Delta C_{apolar} \ln\left(\frac{T}{T_{apolar}}\right) - \Delta C_{polar} \ln\left(\frac{T}{T_{polar}}\right) \tag{S28}$$

$$\Delta G_{solv} = \Delta H_{solv} - T\Delta S_{solv} \tag{S29}$$

where $T_{apolar}$ and $T_{polar}$ are the temperatures at which $\Delta S_{solv,apolar}$ and $\Delta S_{solv,polar}$ are 0 (385 K and 335 K, respectively). Our previous work indicated ribosome solvation changes during coTF is not a major factor in coTF thermodynamics (see Supplementary Note 9), we estimated the above quantities using surface areas calculated excluding the ribosome. We regard these absolute quantities as an estimated upper bound for $\Delta G_{solv}$ because it is likely that folding intermediates and the native state also interact with the ribosome[6], thus effectively cancelling out any reduction in SASA of the unfolded state due to ribosome interactions. However, in the following section we describe an alternative, more direct approach for the solvation entropy that does not rely on this assumption.

## Calculation of solvation entropy changes using the 2PT method

The water and solvation entropy changes were also assessed more directly from molecular dynamics simulations using the two-phase thermodynamic (2PT) method[136] implemented in the DoSPT code (https://dospt.org/index.php/DoSPT)[137]. For these calculations, we chose five snapshots from our isolated FLN5 $A_3A_3$ V747 simulations detailed above (that is, with different initial protein conformations and solvent configuration) and use these to initiate short molecular dynamics simulations for entropy calculations. We first re-equilibrated the boxes for 10 ns at the target temperature in the NPT ensemble at 1 bar using the velocity rescaling algorithm[89] and the Parrinello–Rahman algorithm[91] as detailed above and the velocity Verlet integration algorithm (md-vv in GROMACS[84]). Production simulations were then run in the NVT ensemble at 283 K and 298 K (to assess the effect of temperature on the water entropy calculations) for 20 ps using the md-vv integrator and saving coordinates and velocities for analysis every 4 fs. Control simulations of pure TIP3P (CHARMM TIP3P) water in a cubic box with a box vector length of 5 nm, resulting in 4,055 water molecules. Five independent simulations were performed by first energy minimizing the system using the steepest-decent algorithm. Then, using a 2 fs timestep and thermostat/barostat settings as for the protein and the md-vv integrator we equilibrated the water box first in the NVT ensemble for 1 ns, followed by 1 ns in the NPT ensemble using the Berendson barostat[90]. The water box was then further equilibrated in the NVT ensemble for 1 ns prior to the production simulation in the NVT ensemble for 20 ps, saving coordinates and velocities every 4 fs. These production simulations were also performed at 283 K and 298 K and then used to calculate the molar entropies of pure water at these temperatures with DoSPT.

For water entropy calculations in the protein system, we first analysed the radial distribution function water surrounding the protein molecule using our 15 μs and 20 μs molecular dynamics ensembles of the isolated protein and RNC and the GROMACS rdf functionality[84] to identify the region of the first two hydration shells that show significantly reduced water dynamics. Using this analysis, we chose a distance cut-off of 3.5 Å between the protein and water centre of mass to define the hydration layer around the protein. With this criterion we then calculated the probability distribution and average number of water molecules in the hydration layer to assess the difference in solvation on and off the ribosome. Water molecules that remain within a defined distance range from the protein during the entire 20 ps production simulation were then selected to calculate the average molar entropy per molecule of water in different environments with DoSPT. The accessible volume for this subsystem was estimated by using the average volume occupied per water molecule in a pure water box under identical conditions multiplied by the number of molecules. To obtain the change in solvation entropy (difference between the RNC and isolated system, $\Delta S_{solv,RNC\text{-}iso}$), we used

$$\Delta S_{solv,RNC-iso} = N_{diff}\Delta S_{solv,water} \tag{S30}$$

where $N_{diff}$ is the average difference in the number of water molecules in the hydration layer (RNC-iso) and $\Delta S_{solv,water}$ is the entropy difference between water molecules in the hydration layer (0–3.5 Å from the protein) and water molecules in bulk solution (defined here as 36–46 Å from the protein).

## Reporting summary

Further information on research design is available in the Nature Portfolio Reporting Summary linked to this article.

## Data availability

Data are available as source data with the figures. The NMR assignment of FLN5 $A_3A_3$ has been previously deposited in the Biological Magnetic Resonance Data Bank (BMRB) under the entry code 51023. The structural ensembles of the unfolded states have been deposited on Zenodo (https://doi.org/10.5281/zenodo.11618750 (ref. 138)). This study made use of the following public datasets deposited in the protein databank (PDB, https://www.rcsb.org/): 4YBB, 6PJ6, 6XZ7, 7K00, 7LVK, 7N1P, 7O1A, 7PJS, 7Z20, 7ZP8, 1QFH, 1TIT and 4Q21. Source data are provided with this paper.

## Code availability

Python scripts used to calculate PRE-NMR data from the ensembles and to refine the ensembles by reweighting are available on Github (https://github.com/julian-streit/PREreweighting). NMR pulse sequences are available on Github (https://github.com/chriswaudby/pp). Codes used to fit the $^{19}F$ NMR spectra are available on Github (https://github.com/shschan/NMR-fit).

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

**Acknowledgements** This work was supported by a Wellcome Trust Investigator Award (to J.C., 206409/Z/17/Z). The authors thank I. Chen (St. Jude Research) for helpful general comments on the manuscript; M. Smith (Université de Montréal) for the gift of the plasmid encoding the protein SOS$_{cat}$; and S. Mukherjee and L. Schäfer (Ruhr University Bochum) for advice on the water entropy calculations. We acknowledge use of the UCL Biomolecular NMR Centre. The Francis Crick Institute is also acknowledged for provision of access to the MRC Biomedial NMR Centre and receives its core funding from Cancer Research UK (FC001029), the UK Medical Research Council (FC001029) and the Wellcome Trust (FC001029). J.O.S. was supported by a BBSRC London Interdisciplinary Biosciences Doctoral Programme studentship. L.F.W. and C.R.H. were supported by MRC Doctoral Training studentships. We thank Diamond and N. Cowieson for access and technical help to acquire the SAXS data on the B21 beamline. This project made use of time on HPC resources on Archer2 (ARCHER2 UK National Supercomputing service, https://www.archer2.ac.uk) granted via the UK High-End Computing Consortium for Biomolecular Simulation, HECBioSim (https://www.hecbiosim.ac.uk), supported by EPSRC (grant no. EP/R029407/1 and EP/X035603/1). We also acknowledge the EuroHPC Joint Undertaking for awarding this project access to the EuroHPC supercomputer LUMI, hosted by CSC (Finland) and the LUMI consortium through a EuroHPC Regular Access call and the Baskerville Tier 2 HPC service (https://www.baskerville.ac.uk/). Baskerville was funded by the EPSRC and UKRI through the World Class Labs scheme (EP/T022221/1) and the Digital Research Infrastructure programme (EP/W032244/1) and is operated by Advanced Research Computing at the University of Birmingham. We additionally acknowledge the use of

the UCL Myriad and Kathleen High Performance Computing Facility (Myriad@UCL and Kathleen@UCL), and associated support services, in the completion of this work.

**Author contributions** Conceptualization: J.O.S., I.V.B., S.H.S.C., T.W., C.A.W., A.M.E.C., L.D.C. and J.C. Methodology: J.O.S., I.V.B., S.H.S.C., C.A.W., T.W., A.M.E.C., L.D.C. and J.C. Investigation: J.O.S., I.V.B., S.H.S.C., S.B., L.F.W., A.M.F., G.J., H.K.S., C.R.H., A.M.E.C., L.D.C. and J.C. Visualization: J.O.S., I.V.B. and S.H.S.C. Funding acquisition: J.O.S., A.M.E.C., T.W., L.D.C. and J.C. Project administration: A.M.E.C., L.D.C. and J.C. Supervision: S.H.S.C., T.W., C.A.W., A.M.E.C., L.D.C. and J.C. Writing the original draft: J.O.S., L.D.C. and J.C. Reviewing the paper and editing: J.O.S., I.V.B., S.H.S.C., L.D.C. and J.C.

**Competing interests** The authors declare no competing interests.

**Additional information**
**Correspondence and requests for materials** should be addressed to Sammy H. S. Chan, Anaïs M. E. Cassaignau or John Christodoulou.

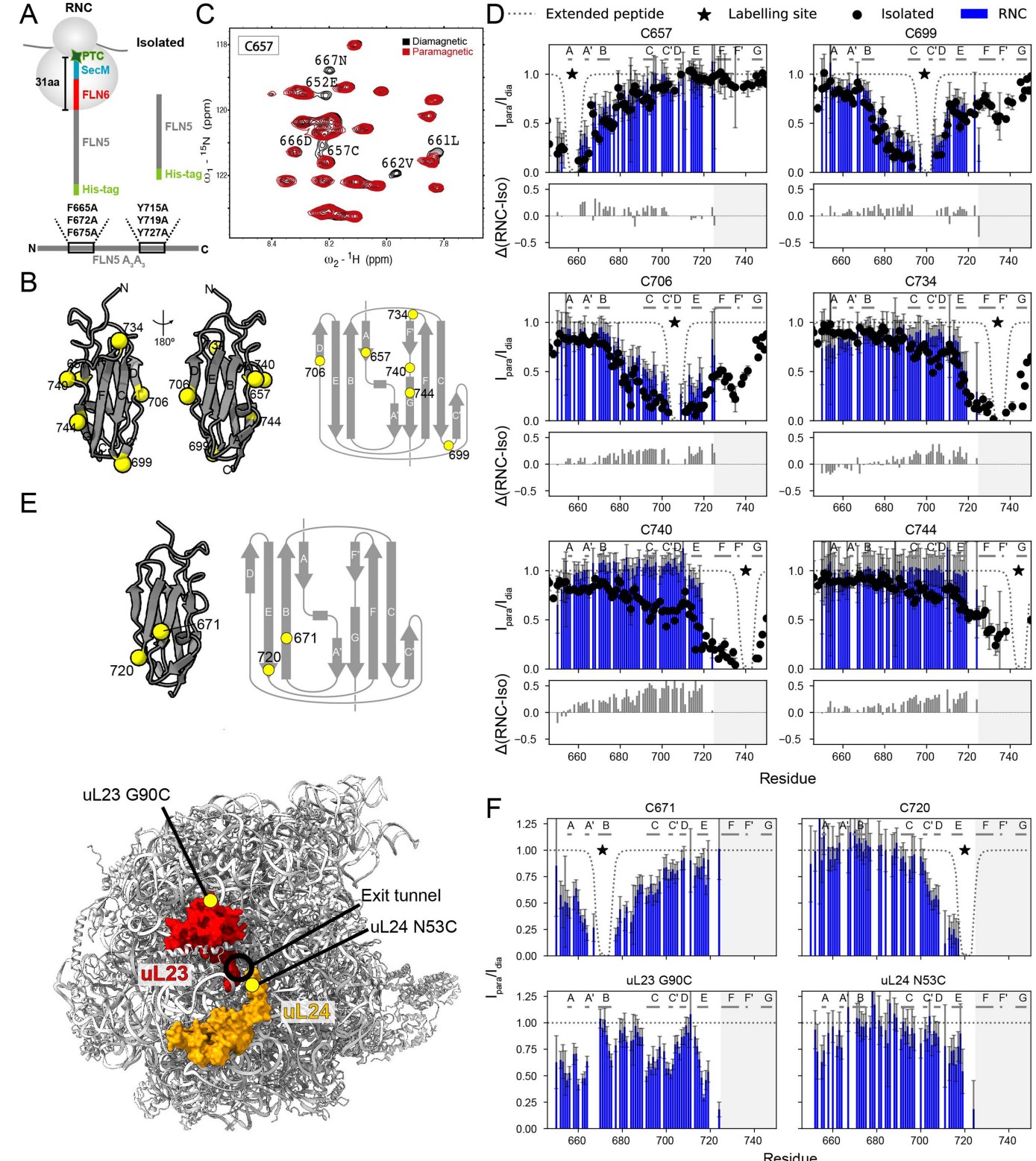

**Extended Data Fig. 1** | See next page for caption.

**Extended Data Fig. 1 | PRE analysis of unfolded FLN5 on and off the ribosome.** **(A)** Schematics of the constructs used for the PRE experiments. The RNC is comprised of an N-terminal His-tag (for purification), FLN5 $A_3A_3$, the subsequent domain FLN6, and an enhanced version of the SecM-AE1 stalling sequence[6,7]. The FLN5 $A_3A_3$ mutant was previously described[7]. **(B)** (Left) The annotated crystal structure (PDB 1QFH[93]) is shown from two views towards the two main β-sheets, highlighting the PRE labelling sites used for both the isolated protein and the RNC. (Right) The secondary structure of folded FLN5 and labelling sites are shown. **(C)** Region of an exemplar $^1$H-$^{15}$N HMQC NMR spectrum of isolated FLN5 $A_3A_3$ spin-labelled at C657 (see Supplementary Fig. 1 for full spectrum). The paramagnetic and diamagnetic spectrum are overlayed. **(D)** PRE intensity ratio profiles for six different labelling sites (indicated with the black star) on (blue) and off (black) the ribosome. NMR data were recorded at 800 MHz, 283 K.

Theoretical reference profiles expected for a fully extended polypeptide are also shown as dashed lines (see methods). The secondary structure elements (β-strands) of native FLN5 are indicated at the top. The shaded region at the C-terminus represents the region of FLN5 that is broadening beyond detection through ribosome interactions (N730-K746, in the RNC)[7]. The second panel with grey bars under each dataset shows the difference between the RNC and isolated data. **(E)** (Top) The annotated crystal structure (PDB 1QFH[93]) of FLN5 is shown with two additional labelling sites used for the RNC construct. (Bottom) Annotated MTSL labelling sites (yellow circles) on the ribosome structure near the exit tunnel. **(F)** PRE intensity ratio profiles for the two addition labelling sites within FLN5 $A_3A_3$ and two ribosomal MTSL labelling sites recorded at 800 MHz, 283 K. All data show the fitted mean NMR intensities ± RMSE propagated from spectral noise. See Supplementary Fig. 1 for NMR spectra.

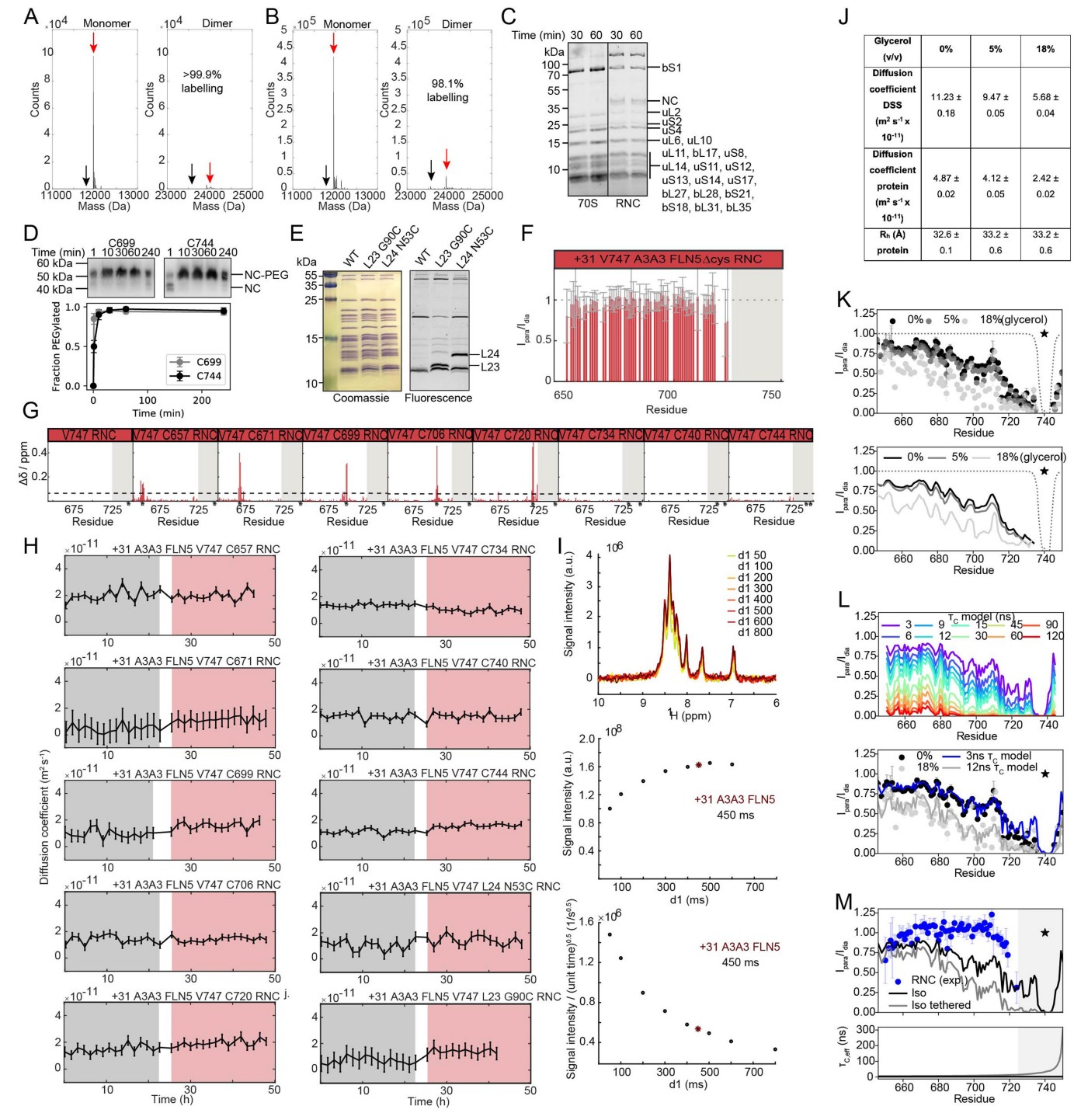

**Extended Data Fig. 2** | See next page for caption.

**Extended Data Fig. 2 | MTSL labelling, quality control and optimisation of PRE-NMR experiments. (A-B)** Mass spectrometry analysis of MTSL-labelled FLN5 $A_3A_3$ cysteine variants C699 V747 (A) and C744 V747 (B). Black arrows indicate the mass of unlabelled FLN5 $A_3A_3$ and red arrows the mass of MTSL-labelled protein. **(C)** Fluorescent gel (12% BisTris) of purified 70 S and RNC (FLN5 + 31 $A_3A_3$ C699 V747) samples labelled with a fluorescent MTSL analogue (ABD-MTS) at pH 8.0 for the indicated time. The gel shows a distinct band for the NC in addition to the ribosome background. Ribosomal proteins are also annotated based on molecular weight estimates. The experiment was performed three times (n = 3) and a representative gel image is shown (see supplementary information, Supplementary Fig. 2 for uncropped gel images). **(D)** Representative anti-hexahistidine western blot (12% BisTris gel) of FLN5 + 31 $A_3A_3$ V747 with a cysteine at C699 and C744 during reaction time-course with molar excess (10000x) of PEG maleimide at pH 7.5 to probe the accessibility and reactivity of the cysteine variants. The fraction PEGylated (mean ± SD; n = 2 for C699; n = 3 for C744) was estimated by densitometry and plotted as a function of time (see supplementary information, Supplementary Fig. 3 for uncropped gel images). **(E)** A representative Coomassie and fluorescent gel (20% Tricine) of purified WT, L23 G90C and L24 N53C 70 S ribosomes after overnight incubation with 10x molar excess fluorescein maleimide at pH 7.5. (See supplementary information, Supplementary Fig. 4 for uncropped gel images; n = 2 for L23 G90C and n = 3 for WT and L24 N53C). **(F)** PRE intensity ratios of the FLN5 + 31 $A_3A_3$ variant without any cysteines in the NC (C747V, Δcys). **(G)** Chemical shift perturbations (CSPs) along the protein sequence for all MTSL-labelled isolated protein (upper row) and RNC (lower row) variants measured in the $^1$H-$^{15}$N SOFAST-HMQC spectra of FLN5 + 31 $A_3A_3$ RNC cysteine variants relative to the isolated FLN5 $A_3A_3$ protein and the FLN5 + 31 $A_3A_3$ RNC, respectively. The labelling sites are indicated with a star (*). The dotted line indicates a threshold of 0.06 ppm. **(H)** Integrity of RNCs during PRE experiments was monitored with $^{15}$N-SORDID diffusion measurements. The calculated diffusion coefficient D is shown throughout NMR acquisition (centre), highlighting the paramagnetic (grey) and the diamagnetic acquisition timeframe (red). **(I)** Optimisation of the recycle delay (d1) time chosen for PRE SOFAST-HMQC experiments to provide maximum sensitivity while also allowing the signal to relax completely before the subsequent scan is initiated. 1D $^1$H spectra at $d_1$ values ranging from 50-800 ms (top, yellow to red gradient); total signal intensity dependence on the $d_1$ value (middle); time-averaged signal (bottom). 450 ms was chosen for PRE experiments. **(J)** Diffusion coefficients of the DSS reference and isolated FLN5 $A_3A_3$ in different concentrations of glycerol. The extracted radius of hydration ($R_h$) for the protein is also shown. The values at 5% and 18% of glycerol were calculated taking into account the increase in viscosity from the DSS diffusion measurements. **(K)** PRE analysis of isolated FLN5 $A_3A_3$ C740 V747 in different concentrations of glycerol. The upper panel shows all individual datapoints while the lower panel shows the data averaged over a window of three residues for ease of visualisation. **(L)** Theoretical effect of increasing viscosity on the PRE intensity ratios ($I_{para}/I_{dia}$). The upper panel shows the predicted PRE profile of the FLN5 $A_3A_3$ ensemble obtained after reweighting using different values of $\tau_c$ (shown in legend in nanoseconds) and the lower panel shows an overlay of the experimental data at 0 and 18% glycerol with the MD profiles using $\tau_c$ of 3 and 12 ns. **(M)** Theoretical effect of increasing residue-specific $\tau_c$ values towards the C-terminus for a tethered polymer, using Eq. S16 and $S^2_{NC} = (1/d) \times S^2_{NC,max}$ where $d$ is the distance to the C-terminal residue (in amino acids) and $S^2_{NC,max}$ the maximum order parameter that the C-terminal residue can reach (set to 0.1 for this illustrative example). The top plot shows the experimental RNC PRE-NMR data and isolated PREs (computed from the reweighted MD ensemble) with either a uniform $\tau_c$ of 3 ns across the sequence or the tethering $\tau_c$ values from the panel below. Unless otherwise indicated, all NMR data are presented as the fitted mean ± RMSE propagated from the spectral noise.

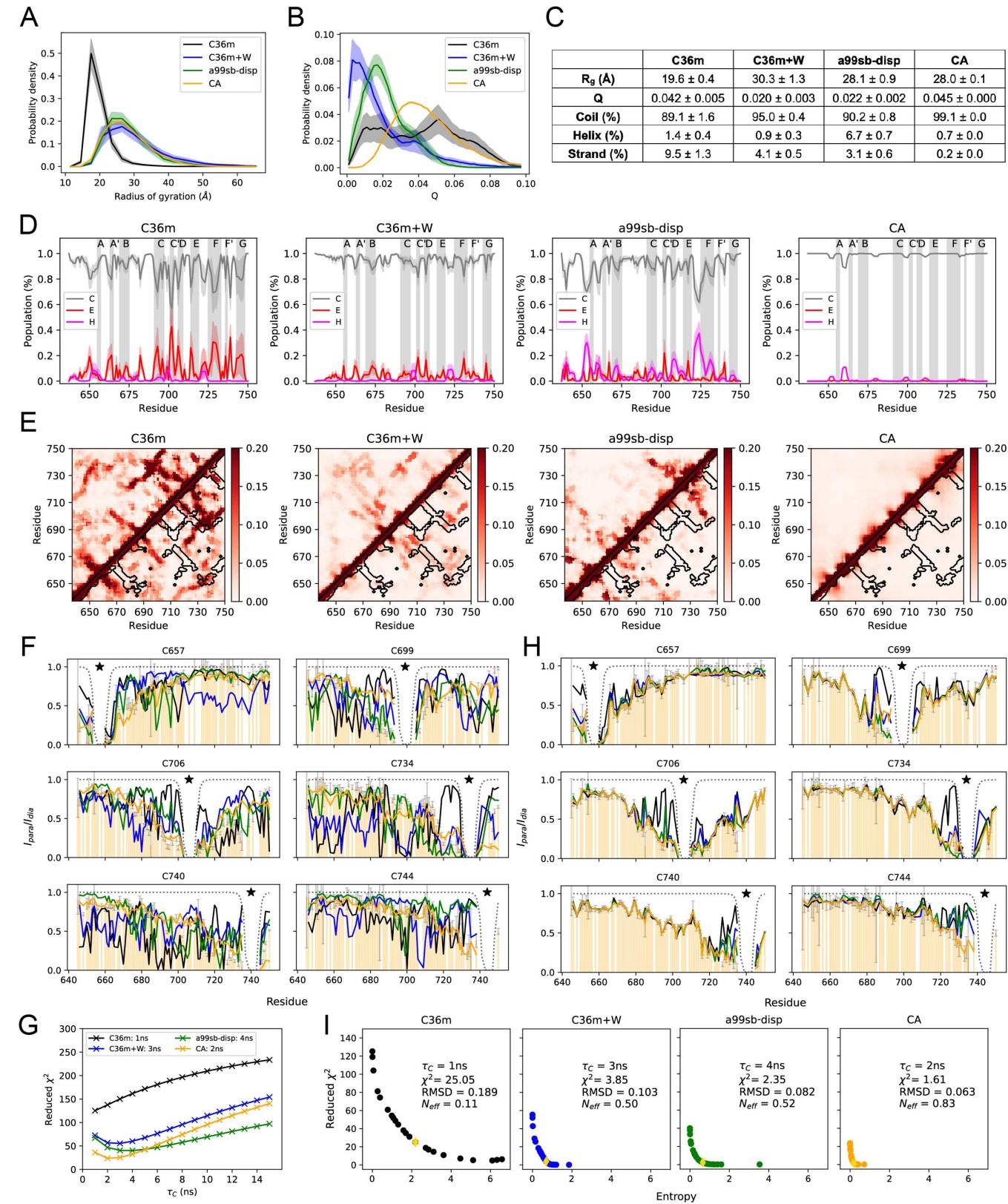

**Extended Data Fig. 3** | See next page for caption.

**Extended Data Fig. 3 | Analysis and reweighting of MD simulations for isolated FLN5 A$_3$A$_3$.** **(A)** Probability distributions of the all-atom radius of gyration (R$_g$) for the different ensembles (mean ± SEM from block averaging). **(B)** Probability distributions of the fraction of native contacts (Q, relative to natively folded FLN5, mean ± SEM from block averaging). **(C)** Ensemble-averaged properties including Rg, Q and secondary structure populations are summarised (mean ± SEM from block averaging). **(D)** Average secondary structure propensities (mean ± SEM from block averaging) along the protein sequence determined using the DSSP algorithm (C = coil, E = strand, H = helix)[131]. The vertical shaded areas highlight the regions of β-strands (annotated as strands A-G) in natively folded FLN5. **(E)** Average contact maps of the ensembles (zoomed in to a probability of 0.2 for clarity). Contacts were defined as Cα-Cα distances of less than 10 Å. The black contours highlight the native contact map of folded FLN5. Above and below the diagonal are identical. **(F)** Overlay of experimental data (shown in transparent orange bars) with the calculated PREs of the four ensemble before and after **(H)** reweighting. Colours are as in panels A-B. **(G)** Determination of optimal τ$_c$ for each ensemble by computing the reduced χ$^2$ statistic against the experimental PRE-NMR data (Extended Data Fig. 1). Values of τ$_c$ were scanned in steps of 1 ns from 1 to 15 ns and the optimal value found is displayed in the figure legend. Colours are as in panels A-B. **(I)** L-curve analysis to identify an optimal balance between the prior ensemble and agreement with experimental data[117]. The entropy term on the x-axis represents the Kullback-Leibler divergence and quantifies the extent of deviation from the prior ensemble. The optimal value of τ$_c$ as determined from the prior ensemble as well as the χ$^2$, RMSD and $N_{eff}$ (fraction of effective frames contributing to the ensemble average calculated as ln(-Entropy)[119]) are displayed in each panel for the corresponding elbow of the L-curve, which is the final solution chosen from the reweighting analysis (see methods).

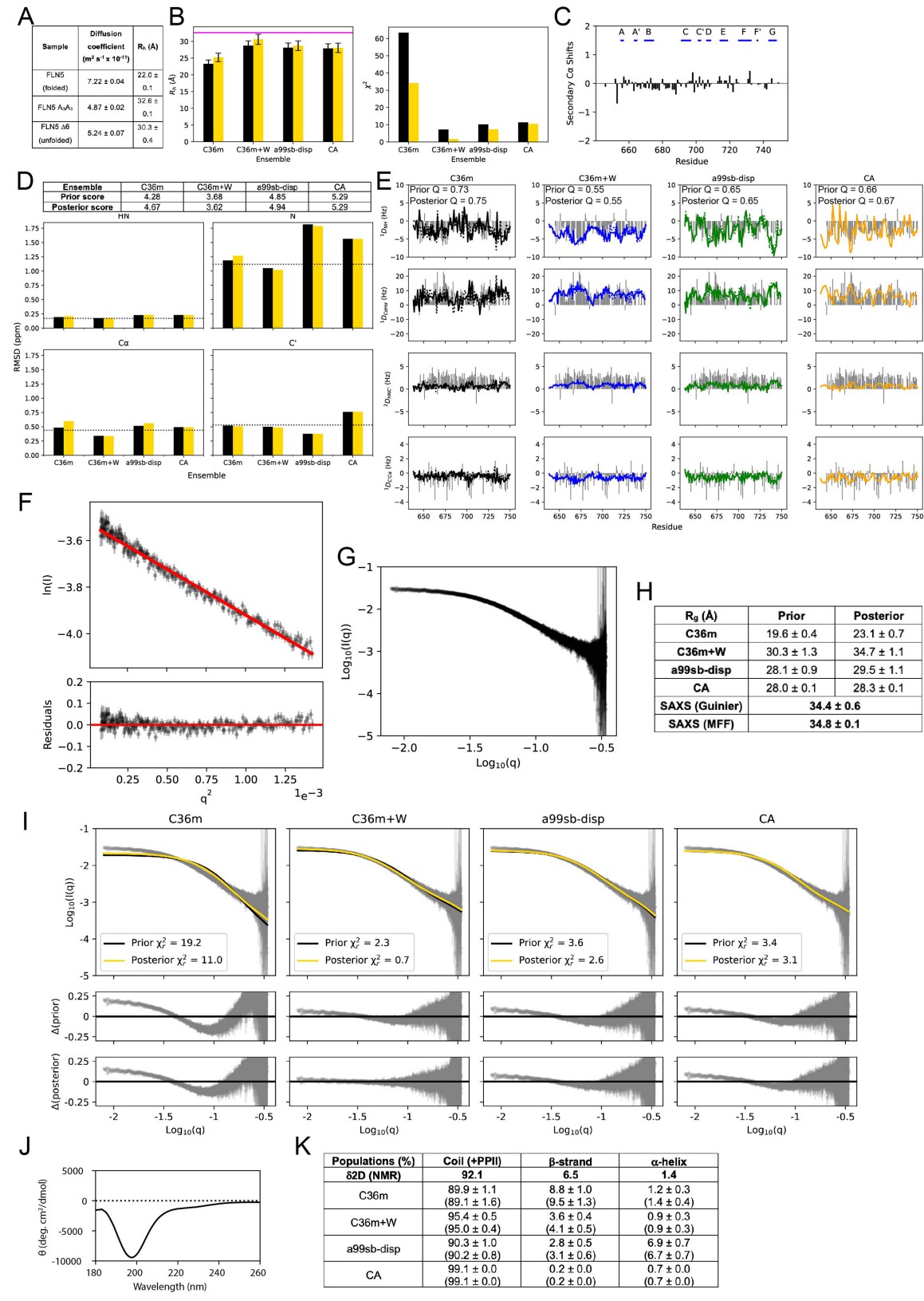

**Extended Data Fig. 4** | See next page for caption.

**Extended Data Fig. 4 | Validation of the ensembles against orthogonal data not used in the reweighting process. (A)** Diffusion coefficients (mean ± RMSE propagated from NMR intensity fits) and radius of hydration ($R_h$) (see methods) as measured for folded FLN5, FLN5 $A_3A_3$ and the unfolded state of FLN5Δ6, a previously characterised truncation variant[24]. **(B)** Comparison between the experimental $R_h$ (32.6 ± 0.1 Å, plotted as a horizontal line in magenta) and the calculated $R_h$ of the ensembles before (black bar) and after (yellow bar) reweighting. The error bars represent the uncertainty around the ensemble average expected from the forward model (see methods). The right panel shows the corresponding $\chi^2$ values, quantifying the agreement with the experimental data. **(C)** Secondary Cα chemical shifts of FLN5 $A_3A_3$ using the random coil shifts predicted by POTENCI[139]. **(D)** Comparison between experimental and calculated chemical shifts from the MD ensembles before (black bars) and after (yellow bar) reweighting for each nucleus. The table above the plot summarises a global agreement score, calculated by adding the nucleus specific RMSD values normalised by the error of the forward model. The forward model error is plotted as a horizontal line in the bar plots, taken as the RMSE values reported by the method[124]. **(E)** Comparison between the experimental RDCs (grey bars) measured in PEG/octanol with the simulated RDCs before (dotted line) and after reweighting with the PRE data (solid line). The RDC Q-factors are used to quantify the agreement. **(F)** Guinier region and linear fit (red line) to the experimental SAXS data (black circles). The bottom plot shows the residuals. **(G)** Experimental SAXS profile shown as a double log plot (mean ± errors propagated as determined by the ATSAS package[80]). **(H)** Ensemble-averaged $R_g$ values obtained from the MD ensembles before (prior) and after reweighting (posterior) compared with the experimental value from the Guinier analysis in panel F, obtained with the autorg tool[80], and the molecular form factor (MFF) analysis[140]. **(I)** Comparison of the experimental and theoretical SAXS profiles obtained from the MD ensembles before and after reweighting. The goodness of fit is quantified with the reduced $\chi^2$ and residuals are shown below the main plot for the prior and posterior ensembles. **(J)** CD spectrum of isolated FLN5 $A_3A_3$ recorded at 283 K. **(K)** Secondary structure populations obtained from the NMR chemical shifts with δ2D[141] compared with average populations observed in the MD ensembles before (in parantheses) and after reweighting (mean ± SEM from block averaging).

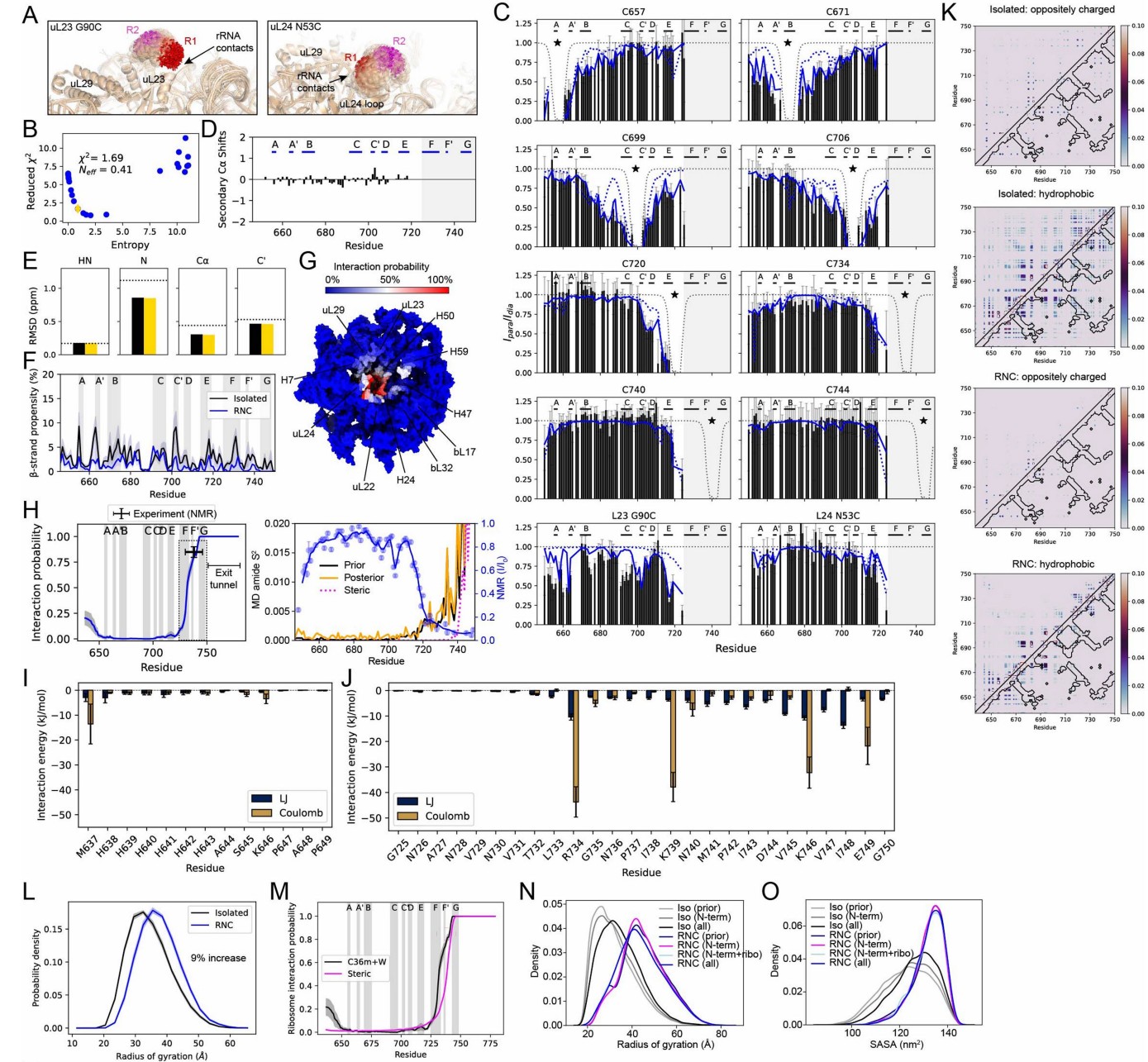

**Extended Data Fig. 5** | See next page for caption.

**Extended Data Fig. 5 | Analysis of unfolded state ensemble on the ribosome obtained from all-atom MD simulations. (A)** Modelling of MTSL rotamer distribution on ribosome labelling sites uL23 G90C and uL24 N53C. Ten *E. coli* ribosome PDB models (highest resolution models available to date: 4YBB, 6PJ6, 6XZ7, 7K00, 7LVK, 7N1P, 7O1A, 7PJS, 7Z20, 7ZP8) were aligned to the simulation ribosome frame in PyMOL (v2.3). For each ribosome model, MTSL rotamers were fitted to the labelling sites as described in methods. The transparent cloud represents the rotamer cloud from these ten ribosome models, highlighting how small fluctuations in the labelling site can lead to different rotamer distributions. R1 represents the rotamer distribution fitted to the ribosome model utilised in the all-atom MD simulations, while R2 is the rotamer distribution fitted to the ribosome model utilised in our previous work[6]. We find the RNC ensembles to be in better agreement after reweighting with the R2 rotamer distribution compared to the R1 distribution and used the R2 distribution for the results presented here. **(B)** Bayesian reweighting of the FLN5 + 31 $A_3A_3$ RNC ensemble using the experimental PRE data is shown (see methods). The final $\chi^2$ and $N_{eff}$ obtained at the elbow of the curve are shown on the plot. **(C)** Comparison of back-calculated PREs from MD and the experimental data (black bars, Extended Data Fig. 1) before (dotted blue line) and after reweighting (solid blue line). **(D)** Secondary C$\alpha$ chemical shifts of FLN5 + 31 $A_3A_3$ measured at 283 K using the POTENCI random coil values[139]. **(E)** Average agreement (reported as the RMSD in ppm) between MD (calculated) and experimental chemical shifts before (black) and after (yellow) reweighting with the PRE data. The dotted horizontal line represents the error of the forward model[124]. **(F)** β-strand secondary structure propensity (mean ± SEM from block averaging). **(G)** NC interactions with the ribosome mapped onto the surface of the ribosome. **(H)** Left: Interactions between the NC and ribosome surface along the protein sequence (mean ± SEM from block averaging). The black cross indicates the experimentally estimated interaction for the C-terminal binding site (within the dotted rectangle) from our previous work[7]. Right: A comparison of amide $S^2$ order parameters from MD simulations with relative NMR intensities[7] further supports the accuracy of NC-ribosome interactions observed in the MD simulations. The decrease in NMR intensities towards the C-terminus around residue 720 coincides with an increase in the amide $S^2$ (restricted dynamics due to ribosome binding). A steric-only model (see methods) does not predict this increase correctly, only showing an increase in the amide $S^2$ around at -residue 740. **(I-J)** The residue-specific interaction contributions from Lennard-Jones (LJ) and Coulombic energies (mean ± SEM from block averaging) of the N-terminal (I) and C-terminal (J) ribosome-binding segments are shown. Ribosome interactions are driven by positively charged C-terminal residues (R734, K739, K746) with the rRNA and E749 interacting with RNA-bound $Mg^{2+}$ ions and K47 within the uL24 loop. **(K)** Analysis of intramolecular contacts within FLN5 $A_3A_3$ on and off the ribosome between different types of residues (oppositely charged and hydrophobic). **(L-M)** Probability distributions of the FLN5 $A_3A_3$ steric-only model on and off the ribosome and comparison between the steric-only model and C36m+W ensemble of the NC-ribosome interaction probability along the FLN5 sequence (mean ± SEM from block averaging). **(N)** $R_g$ and **(O)** SASA probability distributions for isolated and RNC FLN5 $A_3A_3$ before reweighting (prior) and after reweighting with different datasets (see Supplementary Tables 2–4).

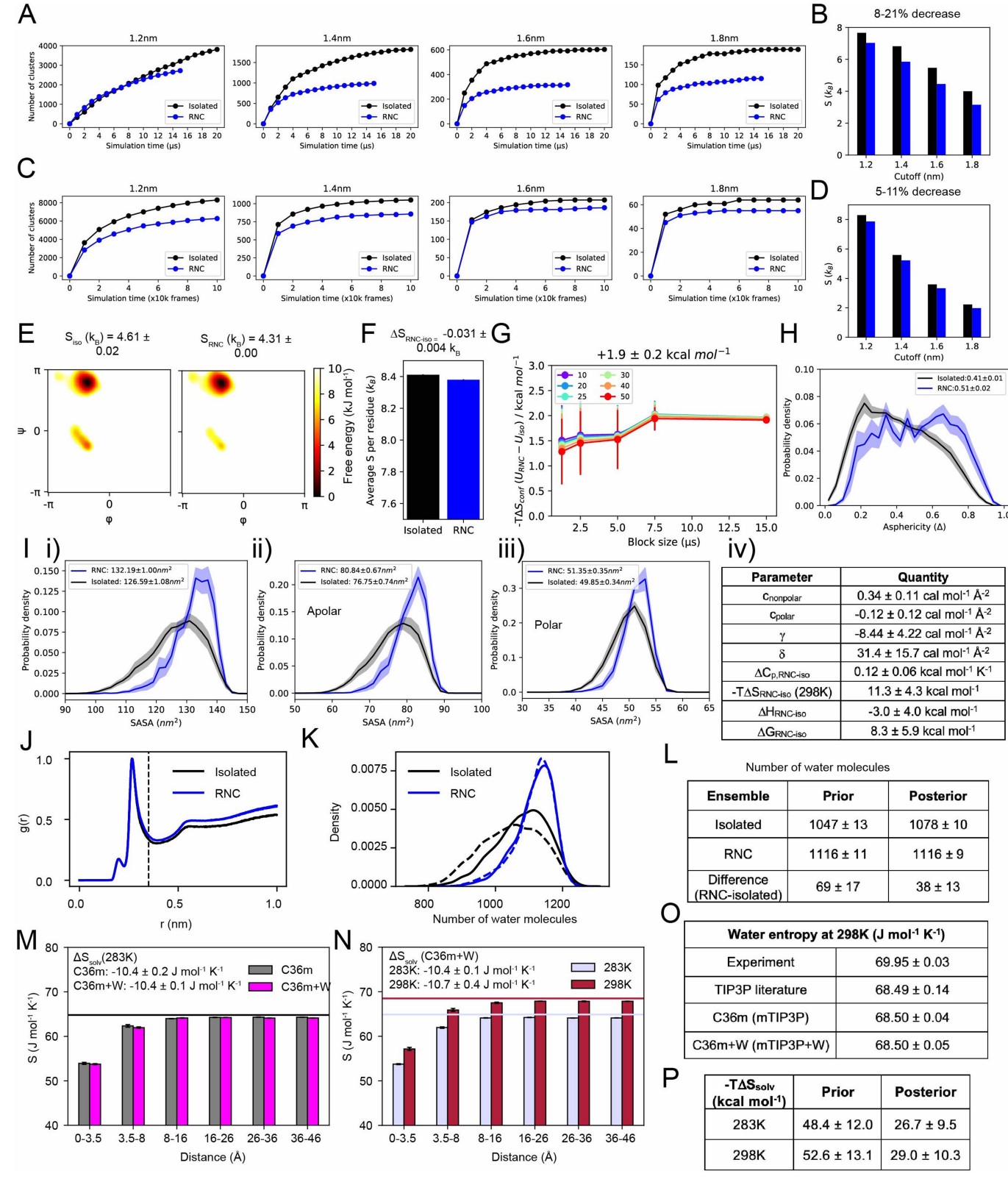

**Extended Data Fig. 6** | See next page for caption.

**Extended Data Fig. 6 | Entropy analysis of the unfolded state on and off the ribosome. (A)** Convergence of the number of clusters visited (see methods for clustering details) for several different cut-off values was assessed by plotting number of clusters as a function of simulation time. This confirmed that for the higher cut-off values (1.4–1.8 nm), sampling has been sufficient to reach a plateau in the number of clusters visited. This was analysed to ensure that differences between the RNC and isolated protein are not due to differences in sampling. **(B)** The average Gibbs entropy ($-\sum_i^n p_i \times \ln(p_i)$, where n is the number of clusters/microstates and p the population of each microstate) was then estimated from the full ensembles after reweighting with the PRE data. **(C)** and **(D)** show the same analysis as in panels A-B but for a simple all-atom steric model of the unfolded state (see Methods). **(E)** Exemplar Ramachandran free energy landscapes of A721 on and off the ribosome. **(F)** The average entropy (S) summed over all residues for each ensemble is shown (mean ± SEM from block averaging). The average difference per residue is shown above the plot. Structures were sampled every 20 ps with equal statistical weights (to avoid differences due to differences in reweighting between the ensembles). **(G)** The resulting effect on free energy (−TΔS for the entire protein at 298 K, mean ± SEM) was calculated using different block sizes of total sampling and number of bins (legend of plot). We observe a convergence towards +1.9 ± 0.2 kcal mol⁻¹ (estimated from 7.5 μs sampling and 50 bins). **(H)** Asphericity (Δ, see methods) of the ensembles shown as probability distributions (mean ± SEM from block averaging). **(I)** Probability distribution (mean ± SEM from block averaging) of the total (i), apolar (ii) and polar (iii) solvent-accessible surface area (SASA) of FLN5 (residues 646–750) is shown for each ensemble. (iv) The thermodynamic parameters of the solvation free energy difference between the unfolded state on and off the ribosome were calculated based on the apolar and polar changes in surface area and experimentally-parameterised functions of the heat capacity, $C_p$, entropy, $S$, and enthalpy, $H$[75,134,135] (see methods for more details). **(J)** Average radial distribution function of the protein (all atoms) to water (centre of mass) distance for the isolated and RNC ensemble. The vertical line represents the 3.5 Å distance cut-off chosen to define the hydration layer consisting of the first and second hydration shell. **(K)** Probability distributions of the number of water molecules in the first hydration layer before (dashed line) and after (solid line) reweighting with PRE-NMR data and **(L)** ensemble-averaged number of water molecules in the hydration layer (mean ± SEM from block averaging). **(M)** Molar water entropy of obtained with the two-phase thermodynamic method (2PT) as a function of distance from the FLN5 $A_3A_3$ protein at 283 K for both the C36m and C36m+W parameters (which differ only in their water hydrogen LJ parameter). The horizontal line represents the bulk molar entropy of water obtained from a pure water box at 283 K (panel O). The solvation entropy ($S_{solv}$) is the difference of the molar entropy of water in the hydration layer (0–3.5 Å) and in bulk (36–46 Å value used). Values are shown as mean ± SEM obtained from five independent simulations (n = 5, see Methods). **(N)** Molar water entropy as a function of distance from the FLN5 $A_3A_3$ protein with the C36m+W force field at 283 and 298 K (mean ± SEM from n = 5). Their respective bulk values obtained from pure water boxes (panel P) are shown as horizontal lines. **(O)** Comparison of molar entropy of water obtained from experiments[142], in previous work in the literature with the TIP3P water model[136], and values obtained in this work with C36m and C36m+W at 298 K (mean ± SEM form n = 5). **(P)** Difference in solvation entropy on and off the ribosome (RNC-isolated, mean ± SEM) obtained by using the solvation entropies per water molecule from panel N and difference in the number of water molecules in the hydration shells of the RNC and isolated ensemble (see methods). This quantity is shown for the ensembles before (prior) and after (posterior) reweighting with PRE-NMR data.

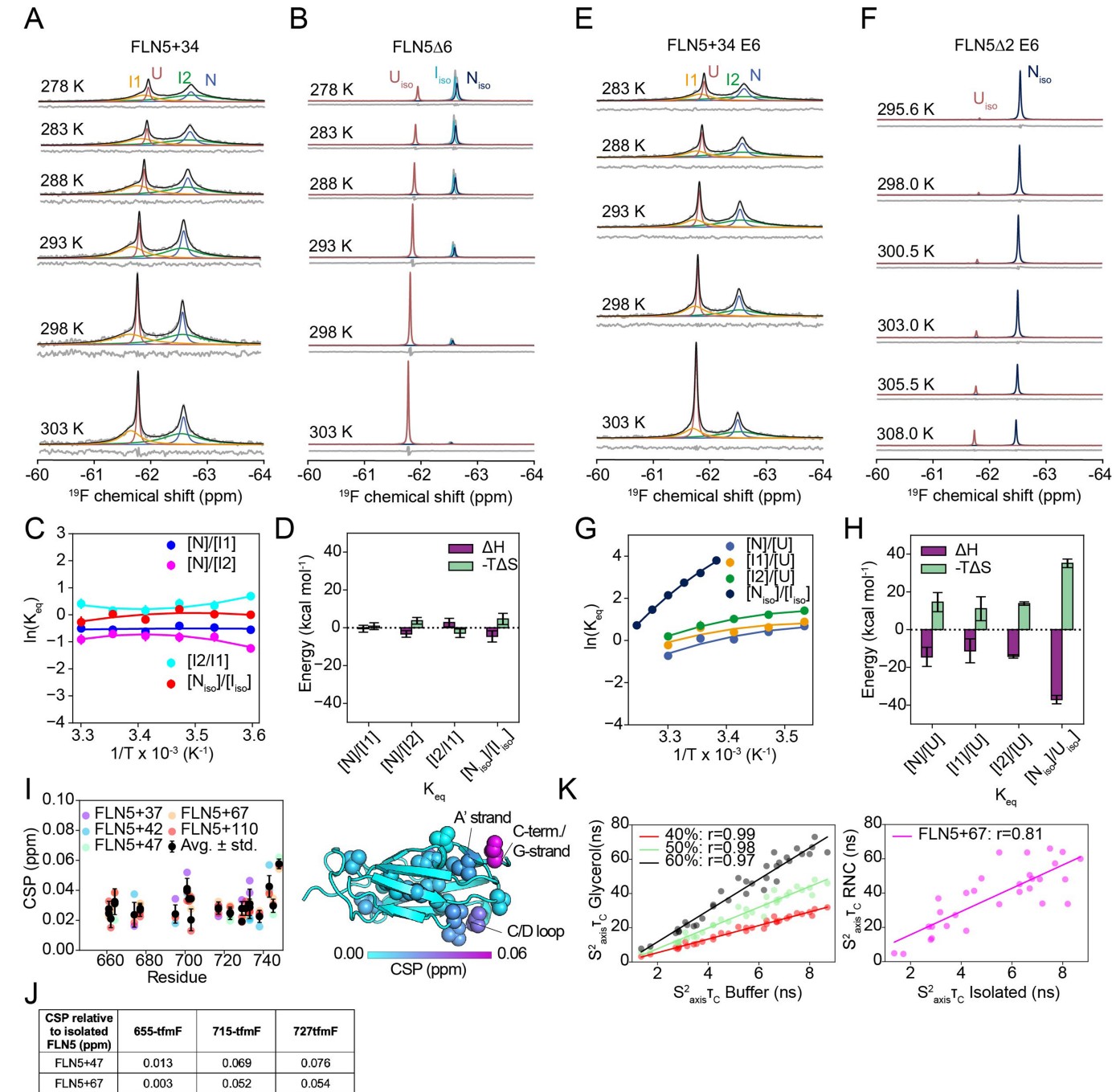

**Extended Data Fig. 7 | Dependence of the folding equilibrium constant on temperature and structural perturbations observed in the native state on the ribosome.** (A-B) [19]F NMR spectra of FLN5 on and off (Δ6 truncation) the ribosome recorded at a [19]F-Larmor frequency of 470 MHz. Raw spectra are shown in grey, lineshape fits in colour and the total fit in black. Residuals after fitting are shown below each spectrum. (C-D) Nonlinear fits to a modified Gibbs-Helmholtz equation (see methods) of the equilibrium constants on and off the ribosome measured by [19]F NMR (from panels A-B) shown as the mean ± SEM propagated from NMR line shape fits (panel C) and the resulting thermodynamic parameters (mean ± SD from fits, panel D). (E-F) [19]F NMR spectra of the FLN5 mutant E6 on and off the ribosome (Δ2 truncation) recorded at a [19]F-Larmor frequency of 470 MHz. The FLN5Δ2 E6 was chosen due to its suitable stability in this temperature range to quantify both [U] and [N]. Raw spectra are shown in grey, lineshape fits in colour and the total fit in black. Residues after fitting are shown below each spectrum. (G-H) Nonlinear fits

to a modified Gibbs-Helmholtz equation (see methods) of the equilibrium constants on and off the ribosome measured by [19]F NMR (from panels E-F) shown as the mean ± SEM propagated from NMR line shape fits (panel G) and the resulting thermodynamic parameters (mean ± SD from fits, panel H). (I) Left: Chemical shift perturbations (CSPs) measured by NMR ([1]H-[13]C HMQC) for methyl groups of natively folded FLN5 (RNCs relative to the isolated protein)[25]. The black datapoints represent the mean ± SD from five different RNC lengths for ease for visualisation. Right: Average CSPs mapped on the crystal structure of FLN5[93]. (J) CSPs (RNC relative to isolated protein) measured for FLN5 labelled with three different [19]F-tfmF labelling sites by [19]F NMR at linker lengths of 47 and 67 amino acids[6]. (K) Correlation plots (along with Pearson correlation coefficients) of methyl relaxation parameters ($S^2_{axis}\tau_c$) for natively folded FLN5[25] in different concentrations of glycerol (left panel) and correlating FLN5 on and off the ribosome (right panel).

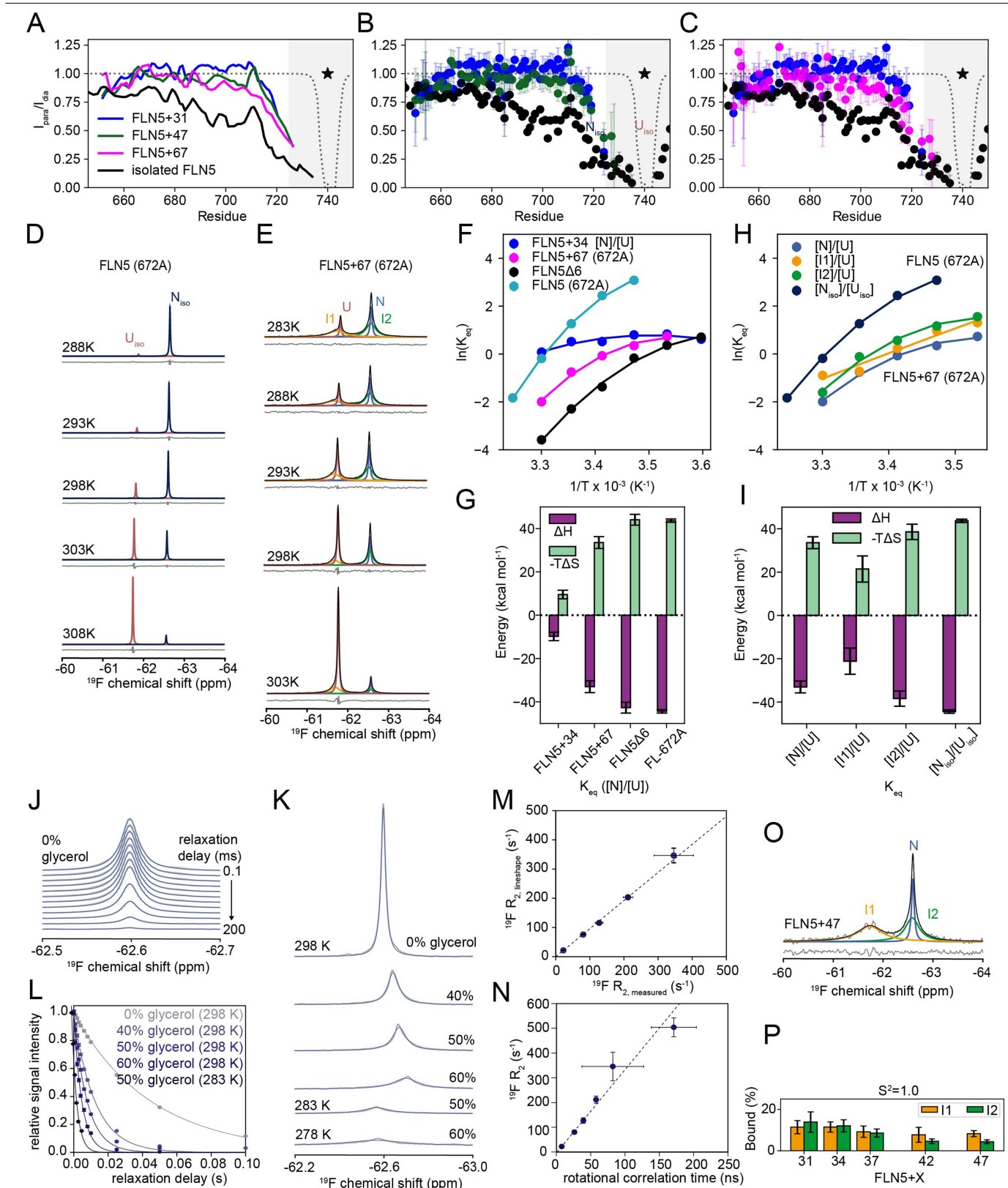

**Extended Data Fig. 8** | See next page for caption.

**Extended Data Fig. 8 | Expansion and entropic destabilisation of the unfolded state on the ribosome persist at longer NC linker lengths.**
**(A-C)** PRE-NMR analysis of FLN5 $A_3A_3$ (labelled at C740, black star) in isolation and at three different RNC linker lengths (FLN5 + 31, FLN5 + 47, FLN5 + 67). Panel A shows a window average over three residues for ease of visualisation. Panels B and C show all datapoints as the fitted mean ± RMSE propagated from spectral noise. The colour scheme in panels B-C is the same as in panel A. Theoretical reference profiles expected for a fully extended polypeptide are also shown as dashed lines. The shaded region at the C-terminus represents the region of FLN5 that is broadening beyond detection through ribosome interactions (N730-K746, in the RNC)[7]. **(D-E)** [19]F NMR spectra of FLN5 (F672A) on and off the ribosome recorded at a [19]F-Larmor frequency of 470 MHz. A destabilising variant (F672A) is used to enable measurements of the unfolded state populations at FLN5 + 67. Raw spectra are shown in grey, lineshape fits in colour and the total fit in black. Residuals after fitting are shown below each spectrum. **(F)** Nonlinear fit to a modified Gibbs-Helmholtz equation of the equilibrium constants on and off the ribosome measured by [19]F NMR (mean ± SEM propagated from NMR line shape fits). **(G)** Thermodynamic parameters estimated from the nonlinear fits in panel F (mean ± SD). FLN5 F672A and FLN5 Δ6 have indistinguishable thermodynamics, validating 672A as a pseudo wild-type system. **(H)** Nonlinear fit to a modified Gibbs-Helmholtz equation of the equilibrium constants (all constants relative to the unfolded state) on and off the ribosome measured by [19]F NMR (mean ± SEM propagated from NMR line shape fits). **(I)** Thermodynamic parameters estimated from the nonlinear fits in panel H (mean ± SD). **(J)** Transverse relaxation rate ($R_2$) measurements of isolated full-length (FL) FLN5 labelled at position 655 with tfmF recorded at a [19]F-Larmor frequency of 470 MHz and 298 K. **(K)** 1D [19]F NMR spectra of isolated, full-length FLN5 in different concentrations of glycerol, fitted spectra in blue, raw spectra in grey. **(L)** Fitting of $R_2$ rates for FL-FLN5 in different concentrations of glycerol. **(M)** Correlation between measured $R_2$ rates (panel L) and those obtained from the linewidths of the peaks in the 1D spectra (panel K). Points are shown as the mean ± SEM propagated from NMR line shape fits. **(N)** Correlation between the [19]F linewidth/$R_2$ rate obtained from line shape fitting (mean ± SEM) and previously determined rotational correlation times of FLN5 in different concentrations of glycerol[25]. **(O)** 1D [19]F NMR spectrum of FLN5 + 47 used in panel (P). **(P)** Estimated populations of coTF intermediates I1 and I2 bound to the ribosome based on the experimental [19]F linewidth at 298 K[6] and linear correlation between linewidth and rotational correlation time (panel N). The ribosome-bound populations were estimated with an $S_{bound}^2 = 1.0$ ($\tau_{R,bound} = 3003$ ns) and are shown as the mean ± SEM propagated from fitted NMR linewidths.

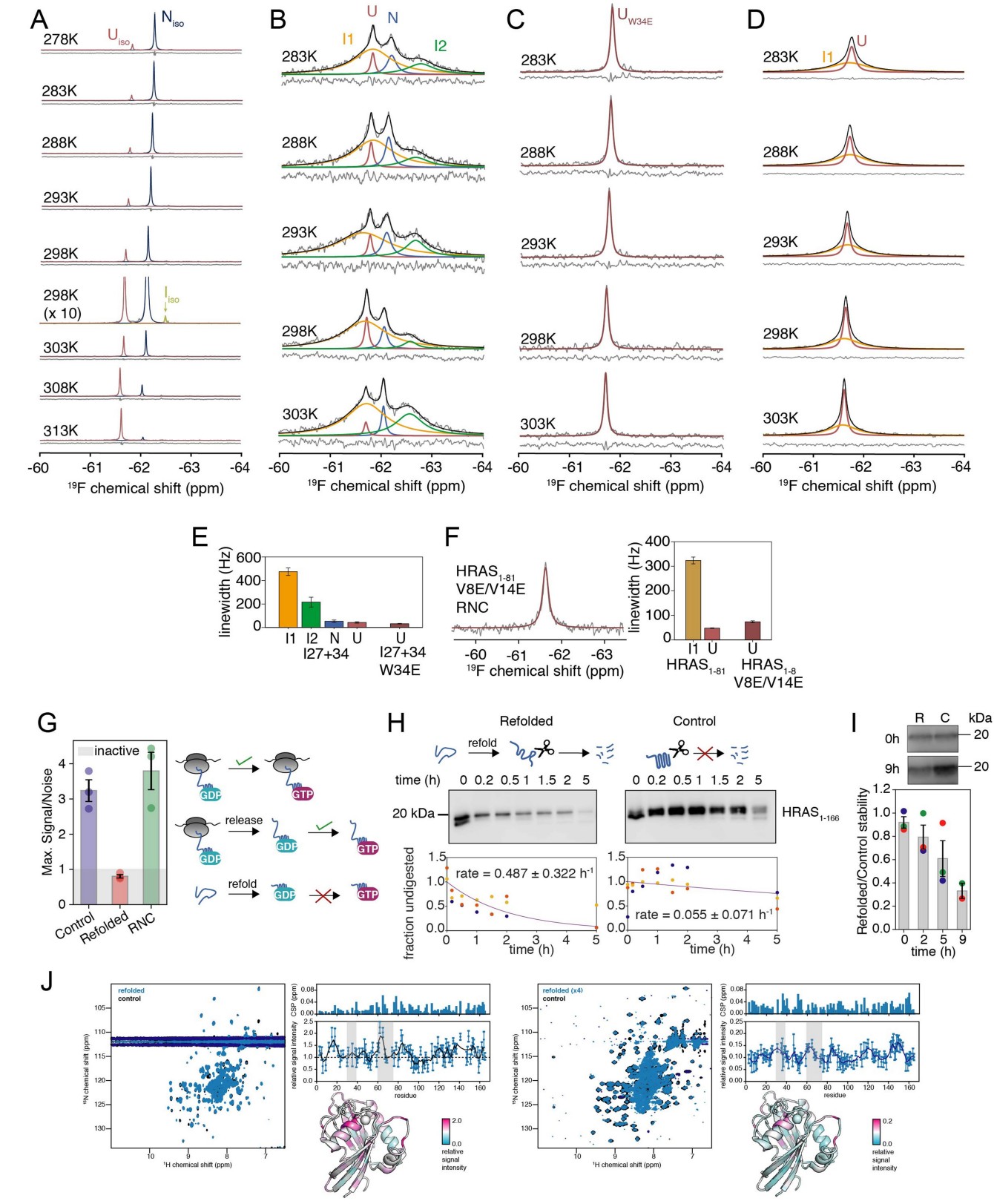

**Extended Data Fig. 9** | See next page for caption.

**Extended Data Fig. 9 | Co- and post-translational folding thermodynamics of I27 and HRAS. (A)** $^{19}$F NMR spectra of isolated titin I27 (F73A variant), **(B)** titin I27 + 34 RNC, **(C)** titin I27 + 34 W34E RNC (a fully unfolded variant[37]) and **(D)** HRAS$_{1-81}$ on the ribosome recorded at different temperatures (at a $^{19}$F-Larmor frequency of 470 MHz). **(E)** The linewidths of all four states in the wild-type and unfolded state of the mutant I27 + 34 RNC are shown as the mean ± SEM from fitted NMR lineshapes. **(F)** $^{19}$F NMR spectrum of HRAS$_{1-81}$ on the ribosome with two destabilising mutations V8E/V14E recorded at 298 K and a $^{19}$F-Larmor frequency of 470 MHz. Analysis of the NMR data in the time domain (as described in ref. 6) shows that the fit is better for a single state compared to two states for the mutant (BIC = 6,897 and BIC = 6,894, respectively). Wild-type HRAS$_{1-81}$ fits better to two states than a single state (BIC = 17,900 and BIC = 17,721, respectively). The right panel shows the linewidths of the two states in wild-type HRAS$_{1-81}$ (Fig. 4d) and the mutant shown here. The bars represent the mean ± SEM from fitted NMR lineshapes. **(G)** HRAS GDP/GTP nucleotide exchange assay (schematic on top shows exchange from GDP- to GTP-bound state for RNC, released (control) and refolded HRAS). The plot shows the GDP/GTP exchange activity (mean ± SEM) from three independent refolding reactions (n = 3). We measured the activity as the maximum signal/noise fluorescence ratio obtained relative to buffer (see Methods). Values of ≤ 1 signify no activity.

**(H)** Pulse proteolysis experiments of refolded and native (control) HRAS. The proteolytic stability of HRAS was assayed with thermolysin (see schematic on top). Exemplar western blots are shown and densitometry analyses from three independent refolding repeats (n = 3) are globally fit to an exponential decay with the obtained degradation rate indicated on the plot (mean ± SD from fitted parameters are shown). See Supplementary Fig. 6 for uncropped gel images. **(I)** Pulse proteolysis experiments (with thermolysin) of refolded (R) and native (control, C) HRAS in rabbit reticulocyte lysate (RRL). Exemplar western blots are shown comparing relative refolded/GDP band intensities at 0 and 9 h time points. Densitometry analyses (mean ± SEM) with n = 3 for the 0, 2 and 5 h time points and n = 2 refolding reactions for the 9 h time point are shown in the bottom bar plot. See Supplementary Fig. 7 for uncropped gel images. **(J)** $^1$H-$^{15}$N SOFAST-HMQC NMR spectra of refolded and native (control) HRAS for two independent refolding reactions (left and right, recorded at 298 K and 700 and 800 MHz, respectively). The chemical shift perturbations (CSPs) and signal intensities (mean ± RMSE obtained from spectral noise) of refolded relative to native HRAS are shown below the spectra. The shaded grey areas highlight switch regions 1 and 2, respectively, and the relative signal intensities are also coloured on the HRAS structure (PDB 4Q21).

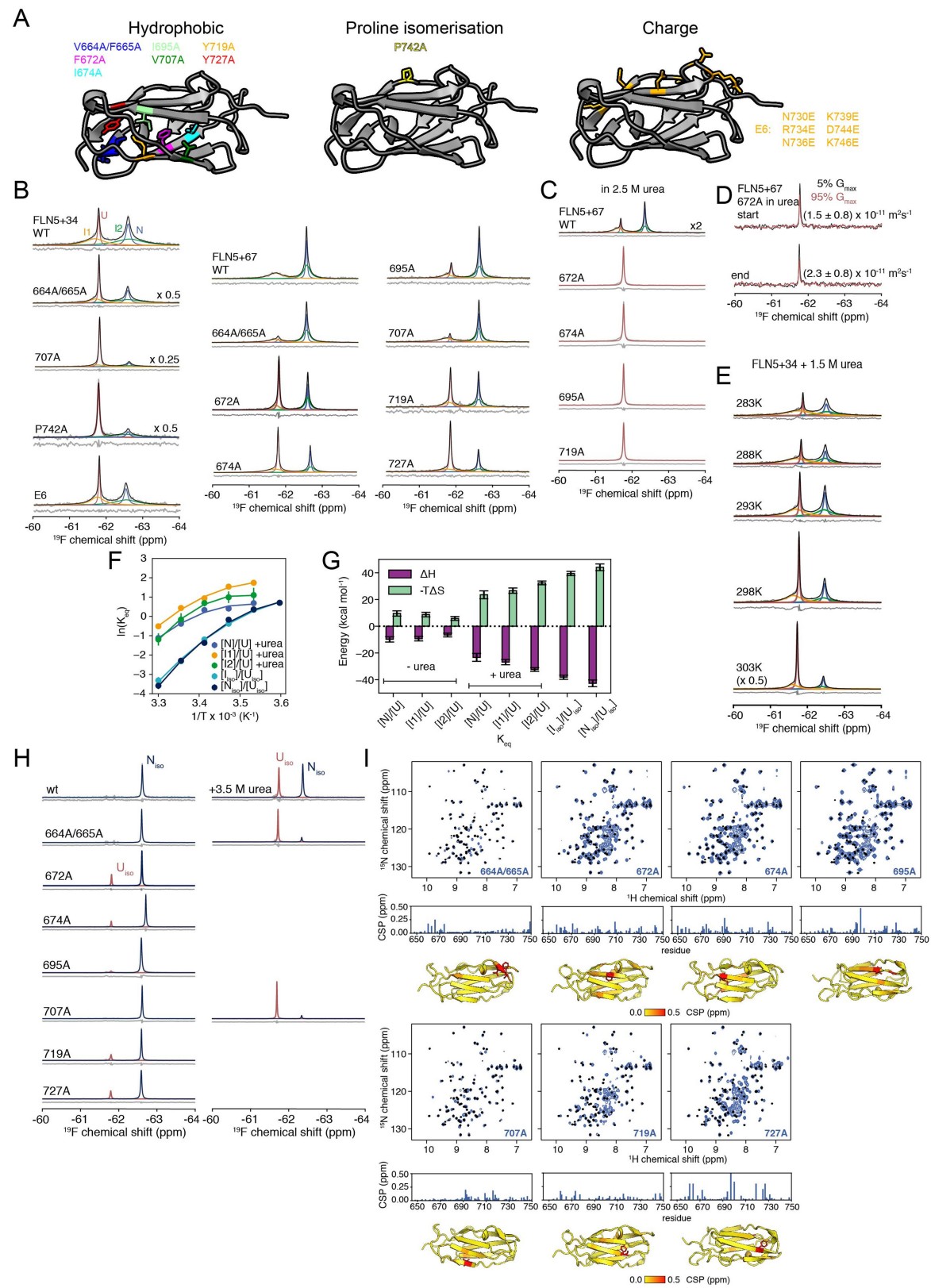

**Extended Data Fig. 10 |** See next page for caption.

**Extended Data Fig. 10 | NMR analyses of destabilising FLN5 mutants on and off the ribosome.** All data were recorded at a [1]H-Larmor frequency of 500 MHz ([19]F-Larmor frequency of 470 MHz), 298 K. **(A)** Mutations mapped on the structure of FLN5[93]. **(B)** [19]F NMR spectra of wild-type and mutant FLN5 RNCs. The spectrum of FLN5 + 34 P742A was previously reported[6]. **(C)** [19]F NMR spectra of wild-type and four mutant FLN5 RNCs in the presence of 2.5 M Urea. The spectral noise was used to estimate the maximum population of the native state to calculate a lower bound of its folding free energy in urea. **(D)** [19]F NMR translational diffusion experiment on FLN5 + 67 672 A RNC in 2.5 M urea to monitor the integrity of the sample in urea. The diffusion coefficient does not change significantly throughout the course of the NMR experiment and is consistent with a ribosome-bound species[6]. **(E)** [19]F NMR spectra of the FLN5 + 34 RNC in 1.5 M urea at different temperatures recorded at a [19]F-Larmor frequency of 470 MHz. Raw spectra are shown in grey, lineshape fits in colour and the total fit in black. Residuals after fitting are shown below each spectrum. **(F)** Nonlinear fits to a modified Gibbs-Helmholtz equation for FLN5 + 34 in 1.5 M urea and isolated FLN5Δ6 as a reference. Values are shown as the mean ± SEM propagated from NMR line shape fits. **(G)** The resulting thermodynamic parameters including the ones of FLN5 + 34 without urea (−urea) for reference are shown as mean ± SD obtained from the fits. **(H)** [19]F NMR spectra of wild-type and mutant FLN5 in isolation. Stabilities were quantified from the unfolded and folded state populations under native conditions, and where 3.5 M urea was used to quantify the stability of less destabilising variants relative to wild-type (assuming a constant m-value[7]). **(I)** [1]H-[15]N SOFAST-HMQC spectra of mutant FLN5 variants in isolation (purple) overlaid with wild-type (black). The chemical shift perturbations (CSPs) are mapped onto the crystal structure of FLN5. The thermodynamic stability and CSPs of isolated FLN5 variants P742A and E6 were previously reported and characterised[7,24].

# Reporting Summary

## Statistics

For all statistical analyses, confirm that the following items are present in the figure legend, table legend, main text, or Methods section.

| n/a | Confirmed | |
|---|---|---|
| ☐ | ☒ | The exact sample size (*n*) for each experimental group/condition, given as a discrete number and unit of measurement |
| ☐ | ☒ | A statement on whether measurements were taken from distinct samples or whether the same sample was measured repeatedly |
| ☒ | ☐ | The statistical test(s) used AND whether they are one- or two-sided *Only common tests should be described solely by name; describe more complex techniques in the Methods section.* |
| ☒ | ☐ | A description of all covariates tested |
| ☐ | ☒ | A description of any assumptions or corrections, such as tests of normality and adjustment for multiple comparisons |
| ☐ | ☒ | A full description of the statistical parameters including central tendency (e.g. means) or other basic estimates (e.g. regression coefficient) AND variation (e.g. standard deviation) or associated estimates of uncertainty (e.g. confidence intervals) |
| ☒ | ☐ | For null hypothesis testing, the test statistic (e.g. *F*, *t*, *r*) with confidence intervals, effect sizes, degrees of freedom and *P* value noted *Give P values as exact values whenever suitable.* |
| ☐ | ☒ | For Bayesian analysis, information on the choice of priors and Markov chain Monte Carlo settings |
| ☒ | ☐ | For hierarchical and complex designs, identification of the appropriate level for tests and full reporting of outcomes |
| ☒ | ☐ | Estimates of effect sizes (e.g. Cohen's *d*, Pearson's *r*), indicating how they were calculated |

*Our web collection on statistics for biologists contains articles on many of the points above.*

## Software and code

Policy information about availability of computer code

| Data collection | NMR data were recorded using Topspin 3.5pl2, pulse sequences available on https://github.com/chriswaudby/pp. Further details are provided in the Methods. MD simulations were performed and processed with GROMACS version 2021.3. To generate the C-alpha (CA) topologies for MD (with GROMACS version 2018.3), SMOG version 2.3 was used and PULCHRA (version 3.06) for all-atom backmapping. For model building and visualisation, PyMol version 2.3 was used. |
|---|---|
| Data analysis | NMR data were analysed using CCPN (version 2.4), nmrPipe (version 11.7) and MATLAB (R2017b, The MathWorks Inc.), codes are available on github.com/shschan/NMR-fit. Python analyses utilised version 3.7.  Python scripts used to calculate PRE-NMR data from the ensembles and to refine the ensembles by reweighting are available on Github (https://github.com/julian-streit/PREreweighting). SAXS data were processed and analysed with PRIMUS/ATSAS version 3.2.1. Chemical shifts were calculated with SHIFTX2 (version 1.10A). RDCs were calculated with PALES (Linux version 10.0). Pepsi-SAXS (version 3.0) was used to calculate SAXS scattering profiles. Blot image densitometry analyses were performed with ImageJ version 1.51. Water entropy calculations were performed with DoSPT version 0.2.2. |

For manuscripts utilizing custom algorithms or software that are central to the research but not yet described in published literature, software must be made available to editors and reviewers. We strongly encourage code deposition in a community repository (e.g. GitHub). See the Nature Portfolio guidelines for submitting code & software for further information.

## Data

Policy information about availability of data

All manuscripts must include a data availability statement. This statement should provide the following information, where applicable:
- Accession codes, unique identifiers, or web links for publicly available datasets
- A description of any restrictions on data availability
- For clinical datasets or third party data, please ensure that the statement adheres to our policy

> Data supporting the findings of this study are included in the article, source data, and extended data. The NMR assignment of FLN5 A3A3 has been previously deposited in the BMRB under the entry code 51023. The structural ensembles of the unfolded states are available on Zenodo (DOI: 10.5281/zenodo.11618750).

## Research involving human participants, their data, or biological material

Policy information about studies with human participants or human data. See also policy information about sex, gender (identity/presentation), and sexual orientation and race, ethnicity and racism.

| | |
|---|---|
| Reporting on sex and gender | N/A |
| Reporting on race, ethnicity, or other socially relevant groupings | N/A |
| Population characteristics | N/A |
| Recruitment | N/A |
| Ethics oversight | N/A |

Note that full information on the approval of the study protocol must also be provided in the manuscript.

# Field-specific reporting

Please select the one below that is the best fit for your research. If you are not sure, read the appropriate sections before making your selection.

☒ Life sciences  ☐ Behavioural & social sciences  ☐ Ecological, evolutionary & environmental sciences

For a reference copy of the document with all sections, see nature.com/documents/nr-reporting-summary-flat.pdf

# Life sciences study design

All studies must disclose on these points even when the disclosure is negative.

| | |
|---|---|
| Sample size | No samples sizes were predetermined. Samples sizes were chosen according to standards generally accepted in the protein folding/structural & computational biology fields. NMR experiments were summed from multiple experiments (generally >20) until signal/noise was sufficiently high, which is typical for NMR studies. All samples undergo rigorous biochemical and NMR quality control measurements, as described. For PRE-NMR experiments, we performed duplicates (n=2) for isolated FLN5 A3A3 labelled at C740 and the FLN5+31 A3A3 RNC labelled at C699, C740 and C744. Repeats were biological repeats (independent sample purifications) and reproduced the data within uncertainty of the measurements. HRAS refolding experiments were performed in triplicate (n=3). These samples sizes were deemed sufficient as repeats led to identical conclusions. |
| Data exclusions | No data were excluded. |
| Replication | All independent attempts to replicate the PRE-NMR data (sample size listed above) were successful. The MD ensembles were concatenated from 10 independent trajectories initiated from different initial coordinates and velocities. |
| Randomization | N/A, as typical for NMR and structural biology studies. Experiments and simulations were rationally designed to be systematic and answer specific technical and biological questions and therefore randomization was not applicable. All experiments and simulations were performed under well-controlled conditions. |
| Blinding | N/A, as typical for NMR and structural biology studies. Our data analysis was systematic without any possible prior knowledge about the result and, thus, blinding was not applicable. |

# Reporting for specific materials, systems and methods

We require information from authors about some types of materials, experimental systems and methods used in many studies. Here, indicate whether each material, system or method listed is relevant to your study. If you are not sure if a list item applies to your research, read the appropriate section before selecting a response.

## Materials & experimental systems

| n/a | Involved in the study |
|-----|----------------------|
| ☐ | ☒ Antibodies |
| ☒ | ☐ Eukaryotic cell lines |
| ☒ | ☐ Palaeontology and archaeology |
| ☒ | ☐ Animals and other organisms |
| ☒ | ☐ Clinical data |
| ☒ | ☐ Dual use research of concern |
| ☒ | ☐ Plants |

## Methods

| n/a | Involved in the study |
|-----|----------------------|
| ☒ | ☐ ChIP-seq |
| ☒ | ☐ Flow cytometry |
| ☒ | ☐ MRI-based neuroimaging |

## Antibodies

| | |
|---|---|
| Antibodies used | Anti-histidine tag (1:5000 dilution, Invitrogen MA1-21315-HRP, lot WK337821), Pan-Ras Polyclonal Antibody (ThermoFisher, PA5-104464, rabbit IgG, 1:1000 dilution), anti-rabbit IgG HRP-linked (Cell Signalling Technology #7074, 1:1000 dilution) |
| Validation | Western blot visualisation as described on the manufacturers' websites:<br>The anti-histidine antibody was verified by relative expression and cell treatment to confirm specificity to the antigen.<br>The Pan-Ras antibody was verified by knockdown to ensure that the antibody binds to the target antigen.<br>The anti-rabbit antibody was validated with Cell Signaling Technology primary antibodies. |

