## [Peer Review file · Nature]

Manuscript Title: The ribosome lowers the entropic penalty of protein folding

Redactions – unpublished data

Reviewer Comments & Author Rebuttals

Reviewer Reports on the Initial Version:

Referees' comments:

Referee #1 (Remarks to the Author):

The manuscript by Streit and colleagues describes an impressive set of experiments and analyses aiming to quantify, to my knowledge for the first time, how much the ribosome affects the folding thermodynamics of proteins and explain why. The authors combine a wide range of biophysical and biochemical experiments as well as simulations and analyses of thermodynamics to shed light on the mechanisms of how the ribosome perturbs folding.

These are interesting and important problems in molecular biology. While we now understand most of the fundamental aspects of protein folding of small proteins, and can predict the structure of many these from sequence, we still have an incomplete understanding of protein folding more broadly in a cellular context. A number of experiments have shown that protein folding in the cell can be substantially more complex than in vitro; in particular it is now clear that many proteins do not fold reversibly in a cellular context. This in turn makes the first folding events special, so that if the ribosome could—for example—aid in folding or protect against misfolding, this could have important consequences for the cell.

A number of experiments and computational analyses have over the years demonstrated that co-translational folding on the ribosome can be different than in free solution, but the molecular origins have mostly remained unclear. Most importantly, it is not clear whether there are general mechanisms that underlie how the ribosome could affect folding of a wider range of proteins. It is against this backdrop that Streit and colleagues studied the structure of the unfolded state on the ribosome and the folding thermodynamics, and compared to the results of the free proteins.

With these points made, my main concern about this work is that it is very hard to evaluate the accuracy of the thermodynamic consequences and analyses that are presented in the paper. The analyses hinge upon an elegant combination of many somewhat uncertain analyses, and it is not completely clear to me how errors in each of these (both experimental uncertainty, but most centrally uncertainty in the assumptions) add up and combine.

Major

1.

The thermodynamic analysis relies on a series of empirical relationships and assumptions as outlined on p. 25 and 26. I realize it is a very vague question, but perhaps the authors could nonetheless outline what kinds of uncertainty one might expect from combining all of these approximate

relationships. Presumably, the errors go well beyond those from the conformational ensembles (e.g. the parameters in the relationships and the approximations involved).

2.

A substantial part of the mechanistic interpretations depends on the conformational ensembles of the unfolded protein on the ribosome. This includes both the conceptual result of a more expanded chain, as well as the following analyses of solvent accessibility and entropies. These ensembles were generated by combining a series of complex NMR experiments with large-scale simulations and reweighting methods. While the authors do a good job of discussing the uncertainty from many of these parts, I would still like to see some additional analyses. More specifically, which of the experiments support the notion of the expanded ensemble, and which of the experiments could be compatible with both an expanded and a more compact unfolded state.

As far as I can see, the expanded ensemble is mostly supported by the PRE data, so that a major part of the results depends on the analyses of these. Given the complexity of the system it is not easy to do orthogonal experiments, and the authors do a good job of explaining some of the uncertainties that come from the interpretation of the PREs. Nevertheless, it would have been good to see whether for example the RDCs alone would also support an expanded unfolded state over a more compact one.

Also, it appears that the expansion of the unfolded state (relative to the free protein) is seen mostly for the labels that are located close to ribosome (Fig. S1D). Why are these effects not seen as clearly when the labels are placed near the N-terminus or middle of the protein? From what I see it is really mostly C740 and C744 that show a major difference. Why?

What do the ensembles look like if the PREs from these label sites are not included? It would be more convincing if the authors could show more clearly that the expansion isn't just supported by the PREs from one or two labelling sites.

As the authors discuss, the PREs depend both on the distribution of structures, but also on the dynamics in complex ways. I realize why the authors cannot measure the actual PRE and instead resort to intensity ratios, but this of course adds additional uncertainty. Presumably, the timescales for motions of the chain (and label) are very different near the ribosome and near the N-terminus (hence the broadening in the last residues). How does this differential chain dynamics throughout the chain affect the measured intensity ratios and the relationship to the PRE and the actual distance distributions?

Do I understand correctly that the previous experiments (Cassaignau et al)—as well as the 19F results presented here—imply that the chain is bound to the ribosome ca. 85% of the time? How does this (slower?) dynamics affect the measured PREs.

I realize that it would be a lot of work to perform PRE experiments on the E6 variant and so I don't think it is a necessary experiment. But it would be good with some convincing argument why such slower motions would not complicate interpretations of the intensity ratios.

Incidentally, did the authors measure actual Γ_2 values for the isolated protein and compare to the values inferred from the intensity ratios? This of course doesn't solve the issue of complex dynamics on the ribosome, but at least is a simple test of the accuracy.

Also, as an aside, are the differences in solvent accessibility seen before the reweighting?

3.

The authors use the conformational ensembles to estimate entropic effects. I very much like this kind of analysis of the ensembles, but I have a hard time understanding the uncertainty involved. The analyses are based on a number of simplifications and assumptions as well as empirical parameterizations, and it is not clear how big the errors of these are. For example, there are a number of methods to relate changes in SASA to entropy and heat capacity, and it is not clear why the specific methods were chosen and what the consequences were.

4.

One of the major findings from this work is that the mutational effects of destabilizing mutations are smaller on the ribosome than off the ribosome. This is an intriguing observation, and the authors suggest that there could be an evolutionary advantage of this. I still don't understand the mechanism of how this would occur. The standard interpretation of why these deletion mutations affect stability is that they diminish the hydrophobic driving force for folding. If the unfolded state is more solvated on the ribosome, why would this effect be smaller?

5.

The authors assess the convergence of the molecular dynamics by comparing the results from 10us of sampling with that of 20us that include the first 10us of aggregate simulations. A more reasonable estimate of the convergence would be to compare the averages of the 10us with the second set of 10 us. What are the averages (and errors) from these two semi-independent ensembles? Perhaps the authors could provide something like the table in Fig. S5 panel I of these two 10 us sets both before and after reweighting.

Also, the authors should provide a bit more detail on how the error bars from the simulations were estimated. They write that they use blocking and in some places write that they used a 7.5 us block size. How was this size determined? And as far as I can see, most simulations are 1.5us or 2us long, so I don't understand how they can use a 7.5us block size. Also, how are the errors estimated after reweighting?

Minor

1.

Throughout the manuscript the authors discuss "stability", but in some/many cases it is not always completely clear what they mean. Stability (to me) roughly refers to the free energy difference between two states, and so it is important to make it clear what those states are. For example, in the introduction (p. 2) the authors write "reveal an entropically-driven destabilisation of the unfolded state on the ribosome", and the reader is then left to guess relative to what. Is it against the unfolded state off the ribosome or against the folded state on the ribosome. Of course, these are all

connected, but I would suggest that the authors go through the manuscript to see whether there are places where the states could be more clearly defined.

2.

The authors perform fitting of thermodynamic models to estimate thermodynamic parameters. It is not clear how the errors of the fits are determined and whether they have examined potential parameter correlations. For example, on page 10, the authors write “DDHN-U,RNC-iso = $+28.8 \pm 10.1$ kcal mol⁻¹ at 298K), and a lower entropic penalty of at least 18 kcal mol⁻¹ on the ribosome (- TDDSN-U,RNC-iso = -28.5 ± 10.1 kcal mol⁻¹ at 298K).”

First, how were the errors on DDH and DDS estimated? Second, are there potential correlations between the fitted parameters?

Similarly, how accurate are the heat capacities determined and are those parameters correlated with the other thermodynamic parameters?

3.

How robust is the conformational clustering? The cutoffs used are very large and RMSD is perhaps not the most meaningful metric for similarity of unfolded structures. They could optionally consider e.g. this clustering method: <https://doi.org/10.1021/acs.jctc.3c00224>

Minor minor

1.

p. 7 Should “analogues” be “analogous”?

2.

p. 28 the authors write “The transverse PRE relaxation rates (Γ) can be obtained by analytically solving equation S32 (performed in MATLAB).” Do they mean “numerically”? If not, what does this relationship look like?

3.

The authors present error estimates in a somewhat unconventional way with very high precision of the errors. For example (but this is just one of many example) the authors write $25.2 \pm 1.26A$ on page 29, which I would normally expect to have been presented as $25 \pm 1 A$.

4.

tfmF-> trifluoromethyl-phenylalanine ?

5.

p. 50 Guthub might be a good name for a restaurant

Referee #2 (Remarks to the Author):

In this manuscript, the authors combine solution NMR and all-atom molecular dynamics simulations to study co-translational folding thermodynamics of a model protein (FLN5) on and off the ribosome. The main experimental approach is to use ^{19}F -NMR to deconvolute temperature-dependent populations of native, unfolded, and intermediate states with and without the ribosome. The experimental protocol and data analysis to extract populations as well as the binding free energy has been previously reported in *Nat Chem* 14, 1165-1173 (2022), also from the Christodoulou group. Then the authors fit these populations using a modified Gibbs-Helmholtz equation to extract enthalpy, entropy, and heat capacity parameters. On the computational side, Bayesian interference reweighting is used to model structural ensembles of an unfolded variant of FLN5 and a ribosome-nascent chain complex (RNC) of FLN5 based on measured ^{15}NH PRE ratios. The ensemble modeling method has been developed as described in Refs. 111, 112 and 114. The ^{15}NH -PRE and ^{19}F -NMR experiments were well performed and carefully analyzed using well established methods. The technical novelty of either experiments or computational modeling is not strong in this reviewer's opinion.

The main scientific conclusion is that the ribosome re-balances the enthalpy-entropy compensation of protein co-translational folding in the RNC complex (Fig. 6). The new finding is that solvation entropy dominates the destabilization of unfolded states, which outcompetes the enthalpic stabilization due to ribosome interactions. The physical picture is clear and appears appealing, although these conclusions are in sharp contrast to one early MD paper also from Christodoulou (Ref. 38). My enthusiasm is further dampened by the following points:

1. The observation of long-range PRE is clear in Fig. 1B. For C-terminal residues in FLN5+31A3A3 close to the ribosome exit channel, the amide PRE for residues 730-746 should be very informative for reweighting ensemble modeling and selecting a suitable force field; however, these key amide resonances are broadened out due to electrostatics from ribosome interactions, as previously reported in *Nat Chem* 13, 1214-1220 (2021). Therefore, the ensemble modeling for this C-terminal segment is done without PRE constraints, regardless of the MTSL labeling sites (Fig. S1D, S6C). Can authors obtain methyl or ^{19}F PRE for residues beyond 730? This way, they can further refine the ensemble modeling for the ribosome tethered FLN5 complex.
2. I appreciate that the authors use different protein systems (Fig. 4) and various destabilizing mutants (Fig. 5) to generalize their physical picture that ribosome re-balances the enthalpy-entropy compensation. If true, the contribution of solvent entropy should depend on the length of amino acids from the ribosome peptidyl transferase center (PTC) and can be tested by appending sequences at the N-terminus of FLN5 that are intrinsically disordered. Based on the well-established MD framework, this could form a predictive understanding for the RNC folding thermodynamics, which would be more convincing and powerful.
3. The authors use an array of orthogonal NMR-derived observables to cross-validate the structural ensemble (Fig. S5), which is nicely done. Can the authors use different experimental techniques to cross-check some parameters that they have already calculated using MD? For example, they can use SAXS to measure R_g and CD for secondary structures for cross comparison.
4. What is the experimental evidence for "fewer" conformations for the RNC complex compared to the free FLN5 peptide (Fig. 6)?

Minor points:

1. A typo: The free energy increase of unfolded state on the ribosome in the last paragraph in page 5 should be from Figure 2F, not 3F.
2. In the middle of the second paragraph on page 1, ...“found that the ribosome drastically expands the conformational ensemble...”. The use of “drastically” is subjective given Fig. 1C-H that is based on MD simulations. Also, “expands” the conformational ensemble could also mean increase the number of conformations. The authors should specify it by adding “structurally” prior to “expands”.
3. The error bar definition is missing for Fig. 5C legend.
4. Fluorine labeling site is missing for Fig. 2G legend.
5. I could not find the reporting summary associated with methods and statistics.

Referee #3 (Remarks to the Author):

In this manuscript Streit and coauthors explore conformational statistics and energetics of nascent polypeptide chain using NMR to assess the conformation and MD to assess energetics and combination of both to get the ensemble using NMR coupling as constraints for MD simulations. The authors use small IG-fold proteins as a model. Earlier proteins from this family were shown to fold spontaneously in vitro so that in principle comparison between in vitro and cotranslational folding could be made. The authors assess change in enthalpy and entropy (compared to free unfolded state) using the algorithms to estimate these quantities from conformational ensembles and changes in ASA compared to fully solvated forms. Based on this analysis the authors conclude that cotranslational folding stabilizes the native state of the proteins, stabilizes cotranslational folding intermediates and buffers detrimental effects of mutations.

The paper presents a tour de force in NMR exploration of conformational statistics of partly folded proteins associated with the ribosome. The MD analysis is much less impressive and is rather routine with several conclusions that are not as solid as underlying NMR analysis and data (see below).

Below I list my concerns in decreasing order of relevance.

- 1) The effect of ribosome on protein stability would be biologically relevant if cotranslationally folded state gets "locked" upon exit from the ribosome. That actually might happen with proteins of very high molecular weight and complex topologies, especially for multimer and most importantly for proteins with disulfides in oxidizing environment. However, Ig fold proteins/ domains can fold and unfold through normal thermal fluctuations in the cytoplasm and their fate (degradation, aggregation etc.) is determined by their stability sans ribosome. Thus the effect of the ribosome on stability of such spontaneously folding proteins as studies here appears irrelevant.
- 2) Even more so is the effect of mutations on stability. It is irrelevant whether ribosome "buffers" such effects as their biological effects are determined by stability of the variants in the cytoplasm without the ribosome.
- 3) The authors analyses of stability effects in the unfolded state appears to contradict basic principles of thermodynamics. They argue that the unfolded state on the Rs is entropically depleted and so that the corresponding increase in free energy cannot be compensated by energy of RS

binding. But that means that binding (or more generally constraints imposed by Rs) goes uphill in free energy compared to the unconstrained unfolded state unbound to the Rs. In this scenario it is this state that will be observed as it is lower in free energy rather than unfolded fragment of the protein bound to the Rs. The fact that actually Rs-bound state is observed means that some estimates of free energies are off.

4) While the authors explore energetic consequences of unfolded state binding to the RS they omit from consideration the effect of Rs on the folded state of the nascent chain. This is a serious omission as the free energy balance of the transient state is determined by both effect of the Rs on both folded and unfolded states and intermediates, But as pointed out in (1) the free energy of the transient Rs bound state is of little biological significance.

The paper is clearly written, and statistical analyses appear adequate.

All in all while this work reports valuable data on structure of the ensemble of Rs bound nascent chain its conclusions and significance are not clear cut.

Author Rebuttals to Initial Comments:

The ribosome lowers the entropic penalty of protein folding, Streit *et al.*

We thank the reviewers for the many helpful comments and suggestions to improve this manuscript. We believe that the revision fully addresses the comments and concerns raised by the reviewers, as detailed in our response below (reviewers' comments in green, our response in black). In the revised manuscript, we have coloured text corresponding to new additions/data in blue.

Referee #1 (Remarks to the Author):

The manuscript by Streit and colleagues describes an impressive set of experiments and analyses aiming to quantify, to my knowledge for the first time, how much the ribosome affects the folding thermodynamics of proteins and explain why. The authors combine a wide range of biophysical and biochemical experiments as well as simulations and analyses of thermodynamics to shed light on the mechanisms of how the ribosome perturbs folding.

These are interesting and important problems in molecular biology. While we now understand most of the fundamental aspects of protein folding of small proteins, and can predict the structure of many these from sequence, we still have an incomplete understanding of protein folding more broadly in a cellular context. A number of experiments have shown that protein folding in the cell can be substantially more complex than in vitro; in particular it is now clear that many proteins do not fold reversibly in a cellular context. This in turn makes the first folding events special, so that if the ribosome could—for example—aid in folding or protect against misfolding, this could have important consequences for the cell.

A number of experiments and computational analyses have over the years demonstrated that co-translational folding on the ribosome can be different than in free solution, but the molecular origins have mostly remained unclear. Most importantly, it is not clear whether there are general mechanisms that underlie how the ribosome could affect folding of a wider range of proteins. It is against this backdrop that Streit and colleagues studied the structure of the unfolded state on the ribosome and the folding thermodynamics, and compared to the results of the free proteins.

We thank the reviewer for their appreciative comments.

With these points made, my main concern about this work is that it is very hard to evaluate the accuracy of the thermodynamic consequences and analyses that are presented in the paper. The analyses hinge upon an elegant combination of many somewhat uncertain analyses, and it is not completely clear to me how errors in each of these (both experimental uncertainty, but most centrally uncertainty in the assumptions) add up and combine.

We take this point very well as we were faced with the same concerns in pursuit of the most appropriate analyses of our experimental data and simulations. We have now added additional analyses and tried to clarify uncertainties in the thermodynamic parameters and ensure that our thermodynamic predictions from the conformational ensembles are robust. The additional clarifications and analyses are described further below in response to point 1 and 3.

Major

1.

The thermodynamic analysis relies on a series of empirical relationships and assumptions as outlined on p. 25 and 26. I realize it is a very vague question, but perhaps the authors could nonetheless outline what kinds of uncertainty one might expect from combining all of these approximate relationships. Presumably, the errors go well beyond those from the conformational ensembles (e.g. the parameters in the relationships and the approximations involved).

The reviewer is correct in that the errors of the thermodynamic analysis go beyond those from the ensembles. In the revision, we discuss the assumptions and uncertainties in Supplementary Note 9. Moreover, we present additional analyses in response to point 3 below.

We stress that the calculations from the ensembles are, of course, estimations/predictions to understand the approximate magnitudes and main factors of the thermodynamic effects at play (and in fact led to the thermodynamic experiments presented), and we have ensured to clearly state this in the revised manuscript (e.g., page 5/line 5, page 5/line 31, page 5/lines38-39, page 6, line 1). We also note that these predictions and the main conclusions of our manuscript are ultimately validated by experimental data, including:

- a) The entropy of protein folding is significantly lower on the ribosome compared to isolation, supported by the ¹⁹F NMR data showing a clear and large difference in temperature dependence (Figure 2H-I).
- b) We have added additional experimental data (Figure 5D in the revised manuscript) showing that the temperature dependence of coTF is highly solvation dependent and can be strongly modulated with even relatively small (1.5M) amounts of urea in the buffer, also suggesting that NC hydration is a key driver of the thermodynamic effects on the ribosome. See also response to point 4 below.

- c) The PRE-NMR and MD data (both independently and together through reweighting the structures, see below in response to point 2iv) support the conclusion that the unfolded state is structurally expanded on the ribosome relative to in isolation.
- d) We have added additional experimental data (Figure 3A-B in the revised manuscript) showing that the structural expansion observed by PRE-NMR experiments correlates with the observed folding entropy/temperature dependence by ^{19}F NMR by studying an FLN5 RNC with a longer linker length (described below in response to point 2vii). This demonstrates that there is a direct relationship between the unfolded nascent polypeptide structural ensemble and coTF thermodynamics.

2.

i) A substantial part of the mechanistic interpretations depends on the conformational ensembles of the unfolded protein on the ribosome. This includes both the conceptual result of a more expanded chain, as well as the following analyses of solvent accessibility and entropies. These ensembles were generated by combining a series of complex NMR experiments with large-scale simulations and reweighting methods. While the authors do a good job of discussing the uncertainty from many of these parts, I would still like to see some additional analyses. More specifically, which of the experiments support the notion of the expanded ensemble, and which of the experiments could be compatible with both an expanded and a more compact unfolded state.

The PRE-NMR analyses alone of the disordered state structures on and off the ribosome support the expanded ensemble for RNC and a more compact unfolded state for the isolated protein. Specifically, all six intramolecular PRE-NMR datasets for both the RNC and isolated protein (C657, C699, C706, C734, C740, and C744) consistently show stronger PREs for the isolated protein than for the RNC. To make that clearer in the revision, we have added a second panel under each dataset in Figure S1D that plots the difference (RNC-isolated) and shows a systematic reduction in PRE-effects across the RNCs and spin-labels.

One potential caveat is that the PRE depends on τ_c , the correlation time constant for the for the inter-atomic vectors connecting the unpaired electron (spin labels) and nuclei detected by NMR (Eq. S4-S5 in the manuscript). Our previous work showed that all NC resonances have reduced intensities relative to the isolated protein¹. This means that the τ_c parameter is either the same or higher on the ribosome compared to in isolation. Theoretically, an increased τ_c , even for an identical distance distribution sampled in the ensemble, would give rise to stronger PREs (experimentally confirmed below with the isolated protein in glycerol solutions, see response to point 2v). However, the RNC exhibits weaker PREs despite residues very likely having an increased τ_c parameter. This is far more compatible with a more expanded ensemble on the ribosome, and not the opposite. Even if the τ_c parameter were identical on and off the ribosome, the trend across residues consistently showing stronger PRE effects for the isolated protein indicates that the isolated protein is more compact than the RNC. We describe these considerations in supplementary note 2 of the revised manuscript.

ii) As far as I can see, the expanded ensemble is mostly supported by the PRE data, so that a major part of the results depends on the analyses of these. Given the complexity of the system it is not easy to do orthogonal experiments, and the authors do a good job of explaining some of the uncertainties that come from the interpretation of the PREs. Nevertheless, it would have been good to see whether for example the RDCs alone would also support an expanded unfolded state over a more compact one.

We acknowledge the reviewer's point about orthogonal experiments and while we have been able to perform such experiments for the isolated protein, including RDCs, SAXS and PFG-NMR (Figure S5), PRE-NMR is the only technically feasible NMR experiment to perform on RNCs that can also report on distances and the structural expansion. We have explored RDCs of RNCs, but these suffer from high experimental errors for RNC samples (see ref.²), and the commonly used alignment medium PEG/octanol fails to produce homogenous alignments for ribosome samples³. Moreover, RDCs are less straightforward to interpret in terms of compaction/average R_g , particularly given that the alignment tensor of the NC may be different even if its structures were identical to the isolated unfolded state due to the influence of the ribosome particle. As discussed in response to point 2iv below, our MD simulations (without any reweighting) also independently predict that the NC is structurally expanded relative to the isolated protein.

iii) Also, it appears that the expansion of the unfolded state (relative to the free protein) is seen mostly for the labels that are located close to ribosome (Fig. S1D). Why are these effects not seen as clearly when the labels are placed near the N-terminus or middle of the protein? From what I see it is really mostly C740 and C744 that show a major difference. Why?

The reviewer is correct that the differences are most visible for C740, C744, and to a lesser extent for C706 and C734 (see revised Figure S1D). Our MD ensembles also show that the structural expansion of the unfolded state on the ribosome is not uniform, with the NC more expanded towards its C-terminus, near the ribosome surface, both in terms of local R_g and long-range contacts (Figures 1H/2A). This is likely why the C-terminal sites depict stronger differences in their PRE profiles.

The larger differences in the PRE-NMR profiles between the RNC and isolated protein for NC/spin label residues closer to the ribosome surface is in line with the distance-dependent steric exclusion of the NC from the large ribosomal particle, which is expected to be more noticeable near the ribosome. To investigate that further, we performed additional PRE-NMR experiments (Figure 3A and S10A-C in the revised manuscript, described below in response to point 2vii) and found that PRE effects become stronger (i.e., lower PRE intensity ratios) when the NC elongates from a 31 amino acid linker to a 47 and 67 amino acid linker, which is consistent with the steric exclusion interpretation.

(iv) What do the ensembles look like if the PREs from these label sites are not included? It would be more convincing if the authors could show more clearly that the expansion isn't just supported by the PREs from one or two labelling sites.

We have compared the expansion (R_g) and solvation (SASA) observed with the ensembles after reweighting while excluding the C-terminal labelling sites (C734, C740, C744). The structural expansion and increased solvation (SASA) are clearly observed in all cases, with or without the C-terminal datasets. Moreover, the MD simulations also predict an expansion without any reweighting. These analyses have been added as supplementary information in the manuscript (tables S2-S4, Figure S7N-O and supplementary note 8).

v) As the authors discuss, the PREs depend both on the distribution of structures, but also on the dynamics in complex ways. I realize why the authors cannot measure the actual PRE and instead resort to intensity ratios, but this of course adds additional uncertainty. Presumably, the timescales for motions of the chain (and label) are very different near the ribosome and near the N-terminus (hence the broadening in the last residues). How does this differential chain dynamics throughout the chain affect the measured intensity ratios and the relationship to the PRE and the actual distance distributions?

This is a point that we had considered carefully. Indeed, the timescales for motions of residues relative to the spin-label (τ_C) are certainly different near the ribosome compared to farther away such as at the N-terminus of the chain. As common for PRE studies of disordered proteins, the exact τ_C are not known *a priori*. We therefore considered how the ribosome affects this parameter and the dynamics. Our previous analyses showed that the NMR signal intensities of all residues across the sequence of the NC are attenuated relative to isolated protein, indicating slower dynamics due to tethering and ribosome interactions¹. We computationally approximated this contribution of the ribosome to τ_C and, specifically, its effect on attenuating NC dynamics on the resulting PREs. This is described with equations S13-S16 in the methods section. In the revision, we have added the τ_C values obtained for the NC ensemble before/after reweighting (supplementary tables S5-S6). The τ_C values increase relative to those of an isolated protein as expected, particularly for the C-terminal labelling sites (C734, C740, C744) and ribosomal labelling sites (L23 and L24).

Moreover, we have undertaken additional experiments to illustrate the expected effect of chain motion on PREs. Using the isolated FLN5 A₃A₃ protein (spin-labelled with MTSL at position C740) we performed NMR experiments under varying glycerol concentrations (mimicking the slower dynamics that the chain experiences on the ribosome⁴). These experiments show that slower dynamics result in stronger PREs throughout the sequence, even though the compaction of the ensemble does not change (verified by NMR diffusion measurements). Therefore, the slower dynamics near the ribosome surface would enhance the PRE effect. However, our PRE-NMR data clearly demonstrate that the RNC exhibits weaker PREs than the isolated protein. This means that even with the additional effect of PRE broadening originating from slower dynamics on the ribosome, the RNC PRE-NMR data cannot be explained by the same level of compaction (or stronger compaction) as observed in the isolated state. These analyses have been added to Figure S2J-M and are described in supplementary note 2 of the revised manuscript.

vi) Do I understand correctly that the previous experiments (Cassaignau et al)—as well as the 19F results presented here—imply that the chain is bound to the ribosome ca. 85% of the time? How does this (slower?) dynamics affect the measured PREs.

This is correct, the C-terminus of the chain is bound $85 \pm 5\%$ based on our previous work¹. As described in the response to point 2v above, these interactions result in increased τ_C values for C-terminal residues which is expected to strengthen the PRE effects on the ribosome relative to in isolation, even if the ensembles were identical. This strengthens the argument that the experimental data (PRE-NMR) are only compatible with a structural expansion on the ribosome.

vii) I realize that it would be a lot of work to perform PRE experiments on the E6 variant and so I don't think it is a necessary experiment. But it would be good with some convincing argument why such slower motions would not complicate interpretations of the intensity ratios.

The suggestion of performing a PRE experiment on the E6 variant is one we considered early in our study but decided against: we reasoned that E6 RNC variant would require comparison with its isolated E6 variant, as the E6 mutations may also affect the structural compaction and contacts in the isolated protein.

Instead, we have opted to run PRE experiments on two longer FLN5 RNCs (FLN5+47 A₃A₃ and FLN5+67 A₃A₃) in the revision. While at FLN5+31 the chain is bound $85 \pm 5\%$ of the time, the E6 variant at the same length only binds $10 \pm 2\%$. The FLN5+47 and FLN5+67 A₃A₃ RNCs bind 57 ± 2 and $26 \pm 2\%$, respectively, as shown in our previous work¹.

Thus, by increasing the linker length to FLN5+67 we achieve a similar reduction in interactions as would be achieved with FLN5+31 E6 and further reduce the steric restriction on the chain by moving it further away from the ribosome particle. These data have been added to the revised manuscript in Figure 3A and Figure S10A-C and are described in the main text (page 8, lines 26-33). The data show that the PREs observed for the longer RNCs are weaker than for the isolated protein, further substantiating the notion that the chain is structurally expanded on the ribosome, and this is also observed when compounding effects of dynamics and ribosome interactions are reduced.

viii) Incidentally, did the authors measure actual Γ_2 values for the isolated protein and compare to the values inferred from the intensity ratios? This of course doesn't solve the issue of complex dynamics on the ribosome, but at least is a simple test of the accuracy.

We have indeed been investigating this point in detail and one of our co-authors is currently preparing a comprehensive manuscript on the topic of validating the accuracy of HMQC intensity-based analyses of PREs against relaxation-based measurements. The HMQC intensity-based approach generally gives very accurate results (Waudby *et al.*, in preparation): [REDACTED]

[REDACTED]

Moreover, comparisons of the MD ensembles with additional SAXS data of the isolated protein that were acquired for the revised manuscript further show that the intensity-derived PRE rates provide accurate restraints that improve the quality of the conformational ensemble, i.e., the agreement with SAXS data improves upon reweighting with PRE data and the resulting R_g of the reweighted ensemble is in excellent agreement with the SAXS-derived value. See Figure S5/S6 and supplementary note 6, page 37/lines 5-20 in the revised manuscript regarding the additional analyses with SAXS data.

ix) Also, as an aside, are the differences in solvent accessibility seen before the reweighting?

Yes, the differences in R_g and SASA are also seen before reweighting, and is discussed further above in response to point 2iv and shown in the revised supplementary tables S2-S4, Figure S7N-O and supplementary note 8.

3.

The authors use the conformational ensembles to estimate entropic effects. I very much like this kind of analysis of the ensembles, but I have a hard time understanding the uncertainty involved. The analyses are based on a number of simplifications and assumptions as well as empirical parameterizations, and it is not clear how big the errors of these are. For example, there are a number of methods to relate changes in SASA to entropy and heat capacity, and it is not clear why the specific methods were chosen and what the consequences were.

We agree with the reviewer that the SASA/solvation analyses are based on assumptions and empirical parameterisations. In the revision, we have estimated the contribution of these parameter uncertainties to the final result. As described in the updated methods section (page 30/line 15– page 31/line 4 in the manuscript), we have propagated parameter uncertainties based on a range of parameter values reported in the literature. Figure 2F and the numbers in the main text (page 5, lines 31-39) have been updated accordingly. The resulting values and errors of enthalpy, entropy and free energy changes have changed slightly in the revised manuscript, however, the conclusion of the entropically driven destabilisation of the unfolded state on the ribosome (relative to in isolation) remains robust.

As a second additional analysis, we have calculated the solvation entropy changes using an alternative approach given that it is the largest contributor to the predicted change in free energy. We reasoned that the conclusion regarding the large solvent entropy change would be more robust if also predicted/calculated with an entirely orthogonal method. (We

have also considered estimating enthalpy changes from force field potential energies, but this was not possible due to the high statistical noise). We have directly analysed and compared water entropy in the hydration shell and in bulk using the two-phase thermodynamic method (2PT)⁵. The additional 2PT/water entropy analysis orthogonally predicts large solvation entropy changes due to the structural expansion and increased solvation of the NC on the ribosome, estimated to be 29.0 ± 10.3 of kcal mol⁻¹, qualitatively consistent with the SASA-based results and remarkably close in magnitude to the experimentally measured change in folding entropy on the ribosome of -34.5 ± 3.2 kcal mol⁻¹. This analysis thus also suggests that the measured folding entropy change on the ribosome can in theory be accounted for by the increased solvation of the unfolded state, and that the solvation-driven entropic destabilisation of the unfolded state is likely the largest energetic outcome resulting from the structural expansion. The 2PT/water analysis has been added to Figure S8K-Q and main text (page 6, lines 1-8), and the approach is described in detail in the methods section of the revised manuscript on page 31 from line 19.

4.

One of the major findings from this work is that the mutational effects of destabilizing mutations are smaller on the ribosome than off the ribosome. This is an intriguing observation, and the authors suggest that there could be an evolutionary advantage of this. I still don't understand the mechanism of how this would occur. The standard interpretation of why these deletion mutations affect stability is that they diminish the hydrophobic driving force for folding. If the unfolded state is more solvated on the ribosome, why would this effect be smaller?

When we initially observed the buffering effect empirically and explored it further, we were perplexed to find that it was not dependent on the type of mutation and is also observed for charge and proline isomerisation mutations (Figure 5 in the manuscript): this finding indicated that the mechanism of buffering is not limited to hydrophobic mutants where the hydrophobic driving force is perturbed.

As the reviewer states, the physical mechanism is not rationalised with a precise mechanistic understanding of the buffering discovery remaining unresolved. Such an understanding is likely to be beyond the scope of the current work and here we merely aim to report the first quantitative observation of this effect and to discuss its impacts on coTF (see additional clarification on page 16 (lines 9-18) in the revised manuscript).

We have however performed additional experiments towards validating our interpretation of the buffering effect being caused by the change of folding thermodynamics on the ribosome. We reasoned that the addition of urea would reduce the entropic effects on the ribosome, given that the effect is dominated by hydration of the NC (Figure 2F in the revised manuscript) which is perturbed by urea. We performed ¹⁹F NMR experiments to measure the protein folding enthalpy and entropy of the FLN5+34 RNC in the presence of 1.5 M urea. These data show that the presence of urea indeed decreases the thermodynamic effects on the ribosome (relative to pure water/buffer). Thus, these data align with the interpretation from our conformational ensembles that the thermodynamic effects on the ribosome are mainly driven by increased NC hydration by water molecules. Moreover, since the buffering is also reduced in the presence of urea (Figure 5C), the temperature dependence of folding on the ribosome therefore correlates with the strength of the mutation buffering effect. We have added these analyses to Figures 5D/S13E-G and describe them on page 13 (lines 24-29) in the revised manuscript.

5.

The authors assess the convergence of the molecular dynamics by comparing the results from 10 μ s of sampling with that of 20 μ s that include the first 10 μ s of aggregate simulations. A more reasonable estimate of the convergence would be to compare the averages of the 10 μ s with the second set of 10 μ s. What are the averages (and errors) from these two semi-independent ensembles? Perhaps the authors could provide something like the table in Fig. S5 panel I of these two 10 μ s sets both before and after reweighting.

We have updated Figure S6 to include a full comparison of all three C36m+W ensembles (the first 10 μ s, the second 10 μ s and the full 20 μ s) both before (panels A-E) and after (panels F-O) reweighting with the PRE-NMR data. This comparison shows that the two halves (10 μ s datasets) are already similar before reweighting, although the first half is slightly more compact than the second half. However, their differences are reduced (e.g., the R_g values are within error of each other) after reweighting with the PRE-NMR data. This underscores a robustness in the conformational sampling, in particular in combination with the experimental data and reweighting.

Also, the authors should provide a bit more detail on how the error bars from the simulations were estimated. They write that they use blocking and in some places write that they used a 7.5 μ s block size. How was this size determined? And as far as I can see, most simulations are 1.5 μ s or 2 μ s long, so I don't understand how they can use a 7.5 μ s block size. Also, how are the errors estimated after reweighting?

We have added a paragraph to the methods section on page 29/line 34-page 30/line 7 to clarify how errors were calculated. The conformational entropy analysis and large block size mentioned by the reviewer are also further clarified on page 30 (lines 15-21) in the methods section of the revised manuscript.

Minor

1.

Throughout the manuscript the authors discuss “stability”, but in some/many cases it is not always completely clear what they mean. Stability (to me) roughly refers to the free energy difference between two states, and so it is important to make it clear what those states are. For example, in the introduction (p. 2) the authors write “reveal an entropically-driven destabilisation of the unfolded state on the ribosome”, and the reader is then left to guess relative to what. Is it against the unfolded state off the ribosome or against the folded state on the ribosome. Of course, these are all connected, but I would suggest that the authors go through the manuscript to see whether there are places where the states could be more clearly defined.

We have added additional detail throughout the main text of the manuscript to clarify this important point.

2.

The authors perform fitting of thermodynamic models to estimate thermodynamic parameters. It is not clear how the errors of the fits are determined and whether they have examined potential parameter correlations. For example, on page 10, the authors write “DDHN-U,RNC-iso = $+28.8 \pm 10.1$ kcal mol⁻¹ at 298K), and a lower entropic penalty of at least 18 kcal mol⁻¹ on the ribosome (- TDDSN-U,RNC-iso = -28.5 ± 10.1 kcal mol⁻¹ at 298K).”

First, how were the errors on DDH and DDS estimated? Second, are there potential correlations between the fitted parameters?

We used the `scipy.curve_fit` function for the fits and errors for the parameters (standard deviation) were estimated from the diagonal elements of the covariance matrix returned by the `curve_fit` function. We have added a short paragraph to the methods section on page 19 (lines 31-36) to clarify this point.

We naturally expect the fitted values of ΔH and $-T\Delta S$ to be of similar magnitude because we are working with samples in a temperature range where the $\Delta G_{\text{folding}}$ is close to 0 (as shown by our ¹⁹F NMR spectra for these experiments). Measurements of equilibrium constants necessitate both states concerned to be detectable and measurable in population (¹⁹F NMR has a detection limit of ~1%).

Parameter correlations from the fits were assessed using the covariance matrix returned by `scipy.curve_fit`. Fitted parameters generally showed strong correlations with each other ($r \geq 0.8$). Thus, these parameter correlations explain the correlation of the parameter uncertainties that we obtained, which we also clarified in the methods section on page 19 (lines 31-36).

Similarly, how accurate are the heat capacities determined and are those parameters correlated with the other thermodynamic parameters?

The heat capacities of protein folding obtained from the ¹⁹F NMR measurements generally showed strong correlations with both ΔH and ΔS in the fits as mentioned above. For example, for FLN5 $\Delta C_{p,N-U}$ showed strong correlations with both ΔH ($r = 0.91$) and ΔS ($r = 0.90$). For I27, $\Delta C_{p,N-U}$ also correlated with ΔH ($r = 0.54$) and ΔS ($r = 0.51$). Moreover, the values obtained are similar to those previously reported for the isolated proteins. This is mentioned in the manuscript on page 7 and lines 23-25 for FLN5, and we have added the literature value of isolated titin I27 on page 12 and lines 15-16 as a comparison for I27 also.

3.

How robust is the conformational clustering? The cutoffs used are very large and RMSD is perhaps not the most meaningful metric for similarity of unfolded structures. They could optionally consider e.g. this clustering method: <https://doi.org/10.1021/acs.jctc.3c00224>

We agree that the RMSD-based approach may not be sufficient for more detailed analyses and have refrained from doing so. Instead, we focus on the global properties of the ensembles (such as R_g) and their distributions and averages. We use conformational clustering in a limited context, with the primary purpose of visualizing the structures and to obtain some approximations as to the diversity of conformational space. For that purpose, clustering appears to be sufficiently robust, e.g., one can visually observe a clear distinction between the clusters shown in Figure 1D.

Minor minor

1.

p. 7 Should “analogues” be “analogous”?

Corrected!

2.

p. 28 the authors write “The transverse PRE relaxation rates (Γ) can be obtained by analytically solving equation S32 (performed in MATLAB).” Do they mean “numerically”? If not, what does this relationship look like?

This should indeed have been “numerically” and has been corrected.

3.

The authors present error estimates in a somewhat unconventional way with very high precision of the errors. For example (but this is just one of many example) the authors write 25.2 +- 1.26A on page 29, which I would normally expect to have been presented as 25 +- 1 A.

We have amended the average R_g values to be reported to one decimal place throughout the manuscript.

4.

tfmF-> trifluoromethyl-phenylalanine ?

We have added the definition of tfmF in the figure legend of figure 2G (4-trifluoromethyl-L-phenylalanine) and in the methods section.

5.

p. 50 Guthub might be a good name for a restaurant

This has now been corrected to Github.

Referee #2 (Remarks to the Author):

In this manuscript, the authors combine solution NMR and all-atom molecular dynamics simulations to study co-translational folding thermodynamics of a model protein (FLN5) on and off the ribosome. The main experimental approach is to use ^{19}F -NMR to deconvolute temperature-dependent populations of native, unfolded, and intermediate states with and without the ribosome. The experimental protocol and data analysis to extract populations as well as the binding free energy has been previously reported in Nat Chem 14, 1165-1173 (2022), also from the Christodoulou group. Then the authors fit these populations using a modified Gibbs-Helmholtz equation to extract enthalpy, entropy, and heat capacity parameters. On the computational side, Bayesian interference reweighting is used to model structural ensembles of an unfolded variant of FLN5 and a ribosome-nascent chain complex (RNC) of FLN5 based on measured ^{15}NH PRE ratios. The ensemble modeling method has been developed as described in Refs. 111, 112 and 114. The ^{15}NH -PRE and ^{19}F -NMR experiments were well performed and carefully analyzed using well established methods. The technical novelty of either experiments or computational modeling is not strong in this reviewer's opinion.

We thank the reviewer for their time and effort. While the basic methodologies individually (i.e., PRE-NMR, ^{19}F NMR, ensemble reweighting, MD simulations, etc.) are not novel in that they have been used before somewhere, we do consider that our work represents a significant advance in applying and combining these methods to uncover a major biological observation in a fundamental area and highly complex and large biological system. These investigations into the detailed structural and thermodynamic properties of RNCs are certainly very much absent in the current (rather limited) coTF understanding. The technical hurdles to study nascent chains (representing <1% of the total ribosomal complex) by any experimental/computational technique underlie the rather limited progression of the coTF field relative to major advances in understanding protein folding *in vitro*.

NMR analyses of these ribosomal complexes (>2.5 MDa) is currently the only way to obtain such a detailed structural and thermodynamic analysis and as such we are the only group doing this level of work despite many attempts. The NMR studies require bespoke preparative biochemistry to produce the 10s of mg quantities with selective and frequently site-specific isotope labelling of the nascent chain against a rendered invisible ribosomal backdrop; and combined with the resulting low solubility ($\leq 20 \mu\text{M}$, 1-2 orders of magnitude lower than typical NMR samples), the limited sample stability (<24 hours at 298K), slowed molecular tumbling resulting in resonance broadening requiring multiple days of acquisition time for even a single $^{15}\text{N}/^{19}\text{F}$ NMR spectrum. High-resolution studies able to give advanced restraints for simulations have only now become experimentally feasible after more than two decades of improvement in our RNC sample preparation and developments in NMR methodology as applied to RNCs^{1,2,6-12}, including establishing both intranascent chain and inter (CRISPR-modified) ribosome-nascent chain PREs described in this work (supplementary Figures S1-S3). The NMR data described in our manuscript alone are the result of ~100 independent, purified samples. Using ^{19}F NMR and undertaking $^1\text{H}/^{15}\text{N}$ PRE-NMR on such a breadth of carefully designed ribosomal complexes representing biosynthetic snapshots to consider folding properties alone is unique and, moreover, is used here to provide what we consider to be novel and fundamental insights into important questions in molecular biology. For example, prior to our work, it has not been clear why proteins fold via stable intermediate states on the ribosome but not in isolation, the latter often resulting in misfolding and aggregation (see the revised Introduction).

Similarly, there is sparse existing literature describing the integration of NMR data with atomistic MD simulations for such a large (3.2 million atoms; requiring 19.1 million CPU hours, CPUh), multi-component (RNA/protein) system and none for RNCs to the best of our knowledge.

Our approach of developing the necessary multi-disciplinary tools tailored to RNCs therefore enables us to present the earliest and most comprehensive structural and thermodynamic description of nascent chains, alongside the first quantitative measurements of enthalpy/entropy of coTF and mutation buffering by the ribosome. As we clarify in the

discussion (page 15/lines 21-23 and page 16/lines9-18), these effects on the ribosome can have important downstream implications for proteins.

The main scientific conclusion is that the ribosome re-balances the enthalpy-entropy compensation of protein co-translational folding in the RNC complex (Fig. 6). The new finding is that solvation entropy dominates the destabilization of unfolded states, which outcompetes the enthalpic stabilization due to ribosome interactions. The physical picture is clear and appears appealing, although these conclusions are in sharp contrast to one early MD paper also from Christodoulou (Ref. 38). My enthusiasm is further dampened by the following points:

We initially also noted the contrast of our findings to those of the 2011 paper (ref.¹³ in this document) and this caused us to take even greater care in putting forward the current work. The prior theoretical work in the 2011 paper describes interpretations from a significantly simpler coarse-grained force field and in the absence of any experimental work. The paper was essentially an early day advert about the possibilities of coTF from the Dobson group prior to the extensive alignment with high-resolution and thermodynamic experimental data that emerged subsequently from our group. The current work uses all-atom force fields with explicit solvent and is supported by and refined with extensive and direct experimental evidence for the structural expansion of the NC (NMR, Figures S1-S7). The thermodynamic conclusions of our MD study are also extensively and experimentally validated by independent, quantitative measurements of enthalpy/entropy on the ribosome by ¹⁹F NMR as described in the manuscript (Figures 2-5).

1. The observation of long-range PRE is clear in Fig. 1B. For C-terminal residues in FLN5+31A3A3 close to the ribosome exit channel, the amide PRE for residues 730-746 should be very informative for reweighting ensemble modeling and selecting a suitable force field; however, these key amide resonances are broadened out due to electrostatics from ribosome interactions, as previously reported in Nat Chem 13, 1214-1220 (2021). Therefore, the ensemble modeling for this C-terminal segment is done without PRE constraints, regardless of the MTSL labeling sites (Fig. S1D, S6C). Can authors obtain methyl or ¹⁹F PRE for residues beyond 730? This way, they can further refine the ensemble modeling for the ribosome tethered FLN5 complex.

We particularly welcome this point raised by the reviewer as we have considered 730-750 region of FLN5 extensively during our study.

Before our response to the main points made here, we would like to briefly clarify the matter of *choice of force fields* in this work: the selection was based on the ensembles of the isolated protein which could be benchmarked against extensive experimental data - PREs, RDCs, ¹⁵N/¹³C chemical shifts, R_n, SAXS; the latter are now in the revised manuscript (see below in response to point 3), and for which we could test all force fields. The ability to assess force fields against RNCs is limited by both the (in)accessible orthogonal experimental data (e.g., SAXS, RDCs) and the extremely high computational cost (i.e., ~19.1M CPUh for the RNC).

The reviewer is correct in pointing out that the absence of NMR resonances of residues 730-750 is due to electrostatic interactions with the ribosome. During the initial phase of our study, we therefore specifically designed three labelling sites in the broadened C-terminal region (C734, C740 and C744) to obtain PRE restraints that directly report on distances between residues 730-750 and short- and long-range regions upstream of the ribosome-interacting C-terminal region. This provided access to restraints relating to the C-terminal region despite it being broadened out beyond detection.

However, the reviewer points out correctly that the PRE-NMR data do not directly capture:

- a) the extent of interaction between residues 730-750 and the ribosome surface; and
- b) the local compaction of residues 730-750.

We discuss below our extensive considerations regarding whether this region is nevertheless modelled accurately in our MD ensemble.

C-terminal NC-ribosome interactions

The extent of interaction between the C-terminus of the nascent chain and the ribosome surface (point a) was assessed by calculating the fraction of the C-terminus bound to the ribosome. As shown in Figure S7H of the manuscript, the interactions observed in our MD models are entirely within the narrow range previously determined by NMR¹. To strengthen this argument, we have also added an additional panel to Figure S7H showing that the decrease in NMR intensities towards the C-terminal region coincides with an increase in the MD-calculated amide order parameter (S^2).

PRE of C-terminal residues by ¹³C/¹⁹F-NMR

The local structure and compaction within residues 730-750 (point b above) can only be experimentally restrained with PRE data measured for residues in this region when MTSL is also conjugated to a residue in this region. Whether such data could be obtained via ¹⁹F NMR or methyl labelling, as suggested by the reviewer is something we have considered and attempted, e.g., via ¹⁹F-spectra of an unfolded FLN5+34 RNC site-selectively ¹⁹F-labelled at two different positions within the C-terminal region (figure below): ¹⁹F-labelling within the C-terminal region (¹⁹F-726, ¹⁹F-732) exhibited very broad linewidths (~200 and ~300 Hz respectively) compared to a labelling site that is more N-terminal but flanking the

broadened region (^{19}F -718, ~ 40 Hz), with even more extreme line broadening expected for a shorter RNC (i.e., FLN5+31 vs FLN5+34) and also within paramagnetic samples (expected linewidths >2 ppm), limiting any accurate/quantitative measurements. This is in line with the ^1H , ^{15}N line broadening (beyond detection) of the C-terminal (residues 730-750) region and the line-broadening would be expected to also be significant for ^1H , ^{13}C -correlated spectra of the RNC in this region. Additionally, ^1H , ^{13}C -correlated spectra would be complicated by poor chemical shift dispersion resulting in the overlap of unfolded state resonances, thus, requiring position specific labels (e.g., also via Amber suppression) for each RNC. This was not explored since even single ^{19}F NMR probes exhibit severe line broadening within the C-terminal region in question (see ^{19}F -726 and ^{19}F -732 below) despite the highly sensitive, dead time-free 1D experiments.

^{19}F NMR spectra of unfolded FLN5+34 RNC labelled at C-terminal sites.

PRE of C-terminal residues by ^{15}N -NMR of longer RNCs

However, as described below (in response to point 2), we have obtained newly acquired PRE-NMR data (from ^1H - ^{15}N HMQC experiments) for labelling position C740 in FLN5+47 A_3A_3 and FLN5+67 A_3A_3 , whose longer linker lengths reduce the C-terminal interactions from $85 \pm 5\%$ at FLN5+31 to $57 \pm 2\%$ and $26 \pm 2\%$ respectively¹. Although this reduction in interactions is not sufficient to render the resonances of residues in the C-terminal region (730-750) clearly observable, the signal intensity increases for residues flanking this region. This has allowed us to quantify intensity ratios for residues E724, A727 and N728 (added as supplementary table S7 in the revised manuscript). These residues exhibit higher PRE intensity ratios on the ribosome compared to in isolation, indicating that the RNC samples longer average distances between residue 740 (the spin-label) and residues 724-728 even at these longer NC lengths where the structural expansion is reduced (see more in response to point 2 below). Consequently, these data support the conclusion that the C-terminal region is also more expanded on the ribosome compared to in isolation (as observed in our MD ensemble, Figure 2A).

2. I appreciate that the authors use different protein systems (Fig. 4) and various destabilizing mutants (Fig. 5) to generalize their physical picture that ribosome re-balances the enthalpy-entropy compensation. If true, the contribution of solvent entropy should depend on the length of amino acids from the ribosome peptidyl transferase center (PTC) and can be tested by appending sequences at the N-terminus of FLN5 that are intrinsically disordered. Based on the well-established MD framework, this could form a predictive understanding for the RNC folding thermodynamics, which would be more convincing and powerful.

This insightful suggestion to examine the hypothesis that the solvation entropy depends on the NC length (i.e., length of amino acids from the PTC) prompted us to first also test whether the structural expansion can be observed to decrease with NC length. Thus, we have performed additional PRE-NMR experiments of FLN5+47 A_3A_3 and FLN5+67 A_3A_3 , having 16 and 36 additional linking residues to the original FLN5+31 A_3A_3 samples, respectively. We selected the C740 labelling site as this site showed the greatest difference in PRE rates between the RNC and isolated protein. These experiments revealed, as expected from steric exclusion principles, that the PREs become stronger compared to FLN5+31 yet remain weaker than those observed for isolated FLN5 A_3A_3 . The data have been added to the revised manuscript in Figures 3A/S10A-C, and are described in the main text on page 8/lines 26-33. These new experiments affirm that the FLN5+67 unfolded state is still more structurally expanded than when off the ribosome. However, the NC becomes more compact as the NC length increases.

We then investigated whether the decrease in structural expansion at a longer NC length correlates with a decreased entropy effect and performed ^{19}F NMR experiments of FLN5+67 across different temperatures. To enable observation of both the folded and unfolded species, we used the destabilising mutant F672A and verified that the entropy/enthalpy changes of folding of isolated FLN5 F672A are identical to those of isolated FLN5 WT ($\Delta 6$). These experiments indeed demonstrated that the entropy of folding is still reduced at FLN5+67 (relative to isolated FLN5) but significantly less than at FLN5+34. This analysis has been added in Figures 3B/S10D-I and is described in the main text on page 8 from line 33 of the revised manuscript.

In summary, our new data conclusively confirms that the entropy effects due to increased solvation decrease as the NC elongates as expected from steric arguments. Moreover, we observe a correlation between the structural expansion of the unfolded state on the ribosome with the magnitude of the entropy effects and temperature dependence of folding.

3. The authors use an array of orthogonal NMR-derived observables to cross-validate the structural ensemble (Fig. S5), which is nicely done. Can the authors use different experimental techniques to cross-check some parameters that they have already calculated using MD? For example, they can use SAXS to measure R_g and CD for secondary structures for cross comparison.

We have measured SAXS data for isolated FLN5 A₃A₃ to further validate the overall structural compaction observed in the MD simulations before and after reweighting with PRE-NMR data. We find excellent agreement between MD and SAXS. Furthermore, we verified that FLN5 A₃A₃ is predominantly disordered with CD spectroscopy. The CD spectrum of FLN5 A₃A₃ is consistent with a highly disordered protein that does not sample significant secondary structure. We also orthogonally used the NMR chemical shifts (HN, N, CA, CO) to predict the global secondary structure populations using $\delta 2D^{14}$ and compared these with the ones observed in MD simulations before (in parentheses, panel K below) and after reweighting with PRE-NMR data for the different force fields. The NMR chemical shifts confirm that FLN5 A₃A₃ is predominantly coil with only minimal amounts of α -helical and β -strand, with a slight preference for β -strand as also observed with the C36m and C36m+W force fields. The SAXS, CD and NMR secondary structure prediction analyses have been added to Figure S5F-K and S6O in the revised manuscript and are described in supplementary note 6 of the revised manuscript.

4. What is the experimental evidence for “fewer” conformations for the RNC complex compared to the free FLN5 peptide (Fig. 6)?

The conclusion that “fewer” conformations exist on the ribosome (i.e., a reduced conformational entropy) is not directly observed by any experiment in this work. Experimentally measuring conformational entropy by NMR relaxation experiments is, however, not feasible for RNCs due to sensitivity limitations. The reduced conformational entropy on the ribosome is thus inferred from the MD ensembles, which are reweighted and validated by NMR data. Given that the contribution to the energetic changes from conformational entropy is rather small compared to solvation (Figure 2F in the manuscript), we have amended Figure 6 by showing the energetic contributions that contribute less in a lighter font to reduce emphasis of this point.

Minor points:

1. A typo: The free energy increase of unfolded state on the ribosome in the last paragraph in page 5 should be from Figure 2F, not 3F.

We have corrected this in the revised version.

2. In the middle of the second paragraph on page 1, ...“found that the ribosome drastically expands the conformational ensemble...”. The use of “drastically” is subjective given Fig. 1C-H that is based on MD simulations. Also, “expands” the conformational ensemble could also mean increase the number of conformations. The authors should specify it by adding “structurally” prior to “expands”.

We agree with the reviewer: we have removed “drastically” and clarified the statement with “structurally” in the revised manuscript. We have additionally added the following sentence on page 3:

“‘expansion’ refers to structural expansion in this article “

3. The error bar definition is missing for Fig. 5C legend.

We have now added an error bar definition in the description for panel 5A (page 14, lines 4-5 in the revised manuscript), which is the same for the rest of the figure.

4. Fluorine labeling site is missing for Fig. 2G legend.

This has now been added to the figure legend for Fig. 2G.

5. I could not find the reporting summary associated with methods and statistics.

The reporting summary associated with methods and statistics was attached to our initial submission and is also submitted (and amended according to the new data added) as part of the revised submission.

Referee #3 (Remarks to the Author):

In this manuscript Streit and coauthors explore conformational statistics and energetics of nascent polypeptide chain using NMR to assess the conformation and MD to assess energetics and combination of both to get the ensemble using NMR coupling as constraints for MD simulations. The authors use small IG-fold proteins as a model. Earlier proteins from this family were shown to fold spontaneously *in vitro* so that in principle comparison between *in vitro* and cotranslational folding could be made.

We thank the reviewer for their insightful comments (below). The reviewer is of course correct in pointing out that proteins that fold reversibly in solution can be used to make direct comparisons between *in vitro* and coTF, which dictated our choice of systems for the current study to enable quantitative on/off ribosome comparisons. We have also

used HRAS, a GTPase domain with an α/β fold, in addition to the Ig domains to generalise our conclusions to a different protein fold (Figure 4).

The authors assess change in enthalpy and entropy (compared to free unfolded state) using the algorithms to estimate these quantities from conformational ensembles and changes in ASA compared to fully solvated forms. Based on this analysis the authors conclude that cotranslational folding stabilizes the native state of the proteins, stabilizes cotranslational folding intermediates and buffers detrimental effects of mutations.

We wish to clarify the summary made by the reviewer here. Firstly, our main conclusions of the paper that the ribosome structurally expands the unfolded state and lowers the entropic penalty of coTF are, in fact, also supported by (and significantly originate from) independent experimental data. These include PRE-NMR (for the structural expansion, Figures 1B/3A/S1D) and ^{19}F NMR (for thermodynamics, Figures 2-4), the latter for three different proteins to illustrate that this is a general phenomenon (independent of sequence and fold).

The reviewer is correct that we argue for a stabilisation of coTF intermediates and buffering of mutations, however, the native state on the ribosome is *destabilised* (not stabilised) on the ribosome relative to the native state off the ribosome (Figure 3D). We have added direct structural evidence that the native state structure and/or environment on the ribosome is perturbed relative to off the ribosome in the revised manuscript to further support and clarify this (Figure S9O-Q, described in the main text on page 8, lines 2-10).

The paper presents a tour de force in NMR exploration of conformational statistics of partly folded proteins associated with the ribosome. The MD analysis is much less impressive and is rather routine with several conclusions that are not as solid as underlying NMR analysis and data (see below).

We thank the reviewer for recognising the leading NMR effort. However, regarding the MD, we would respectfully like to stress that there are only a handful of examples in the literature describing MD simulations of such large (>3 million atoms) and complex systems, and none within the coTF field to the best of our knowledge. We note that the MD simulations required ~19.1 million CPU hours on the UK's largest supercomputer (ARCHER2). Furthermore, the rigorous comparison and refinement procedure against experimental data that we undertook is challenging because of the large size of the RNC and disparate dynamics of the NC on/off the ribosome. It is remarkable that our MD simulations of both the isolated and the RNC agree very well with experimental NMR and SAXS data (Figures S4H, S5-S6, S7A-H in the manuscript). Such a detailed integrative approach of combining NMR and all-atom MD simulations in explicit solvent for a large (2.5MDa), multi-component (protein/RNA) system has not been achieved to date to the best of our knowledge (particularly in the context of coTF). The main conclusions from the MD simulations are a) that the unfolded NC is more structurally expanded than the isolated unfolded protein and b) that this results in an entropic destabilisation of the unfolded state on the ribosome (i.e., lower entropy of folding). Both of these results are experimentally validated (Figures 1B/3A for the expansion, Figures 2-4 for the entropy of folding measured by NMR). Moreover, we have added additional analyses to ensure the conclusion about the entropic destabilisation estimated from MD is robust to its underlying assumptions (see methods section page 30-31 (in magenta), main text page 5 from line 31, page 6/lines 1-8, and Figure S8K-Q, supplementary note 9 in the revised manuscript). Overall, we argue that our integrative analyses are state-of-the-art and have led to entirely novel insights that have been lacking in the field. This is the first paper rationalising coTF thermodynamics, common to multiple proteins (see Introduction), and we describe the first direct relationships between coTF thermodynamics and nascent polypeptide structure.

Below I list my concerns in decreasing order of relevance.

1) The effect of ribosome on protein stability would be biologically relevant if cotranslationally folded state gets "locked" upon exit from the ribosome. That actually might happen with proteins of very high molecular weight and complex topologies, especially for multimer and most importantly for proteins with disulfides in oxidizing environment. However, Ig fold proteins/ domains can fold and unfold through normal thermal fluctuations in the cytoplasm and their fate (degradation, aggregation etc.) is determined by their stability sans ribosome. Thus the effect of the ribosome on stability of such spontaneously folding proteins as studies here appears irrelevant.

Spontaneous (un)folding

The reviewer is correct that if the fate of a protein were solely governed by post-translational energetics, the significance of coTF would be diminished. However, this is not the case for most proteins. The average protein domain has a favourable thermodynamic stability of -5 to -15 kcal mol⁻¹ (folded relative to unfolded state), which translates to typical protein domains only populating approximately 1.0×10^{-9} – 0.02% of unfolded state at room temperature assuming a two-state equilibrium. Kinetically, many proteins do not unfold on a biologically meaningful timescale, rendering them resistant to proteolysis experiments (see ref.¹⁵ below for a review, also referenced in the manuscript). According to the dataset in the protein folding kinetics database¹⁶, up to 35% of protein domains (that are shorter than 300 amino acids and do not have disulfide bonds) unfold with a rate of less than 1/hour at room temperature. Thus, many single-domain protein effectively get "locked" in the native state upon release from the ribosome because unfolding is extremely disfavoured both thermodynamically and kinetically. We have added a sentence in the introduction to state that many

proteins do not unfold after translation on page 1 (lines 33-37) in the revised manuscript to stress the general importance of coTF in proteostasis.

Choice of model systems

That being said, the reviewer is correct in pointing out that Ig-like domains, like FLN5 and I27 studied here, have been shown (including by our group^{1,11,17,18}) to fold reversibly off the ribosome when studied at elevated temperatures, destabilised by mutations or in chemical denaturants. (Under native conditions, however, even these individual domains do not unfold in the absence of mechanical force through normal thermal fluctuations; their unfolding rates at room temperature are ~0.01/hour for FLN5¹¹ and ~0.31/hour for I27¹⁹.) Our manuscript aims to develop an understanding of how the ribosome influences protein folding (relative to *in vitro* folding). All proteins folding on the ribosome, regardless of size or topology, begin their conformational search from the unfolded state. We therefore investigated how the structures and dynamics of unfolded nascent polypeptides impact coTF energetics. Thus, we have chosen the two Ig-like model systems for the following three reasons:

- a) Both proteins fold reversibly and spontaneously in isolation in the presence of mild concentrations of denaturants and upon dilution from fully denaturing conditions^{11,17-20}. This property is essential to enable the kind of thermodynamic measurements and comparisons between the isolated and ribosome-bound nascent polypeptide that we have made. Proteins that do not fold reversibly in isolation cannot be used to study the temperature dependence of the equilibrium constant and extract quantitative thermodynamic parameters, because processes such as misfolding, aggregation or disulfide bond formation can result in irreversible loss of protein from equilibrium conditions.
- b) The structural studies of the unfolded nascent polypeptide required a system that is amenable to quantitative 2D ¹H, ¹⁵N-correlated NMR studies. Currently, no other experimental methods have been successful in obtaining high-resolution structural restraints for disordered RNCs. Due to our extensive experience with FLN5 and its NMR studies on and off the ribosome, this system presented a unique opportunity to study the impact of the ribosome on NC structure. Titin I27 and HRAS RNCs, for example, are not amenable to characterisation by 2D ¹H-¹⁵N NMR, with no observable NC resonances because even transient and weak (non-specific) interactions (a few %) with the ribosome surface are sufficient to broaden the signal beyond detection¹. These factors are however not as limiting for ¹⁹F NMR, a promising future NMR tool to study RNCs more generally, enabling us to generalise our findings from FLN5 to these other systems by using ¹⁹F NMR.
- c) We designed our study to allow us to first undertake a highly detailed residue-specific structural analysis of one RNC, before generalising our findings across protein folds/sequences using more sensitive ¹⁹F NMR measurements (Figure 4 in the manuscript). This reduces the importance of the model systems chosen (aside from the technical considerations outlined in points a-b above). By analysing the thermodynamic properties of FLN5, titin I27, and HRAS, we can conclude that our results do not predominantly depend on the sequence or fold of the protein. This is mentioned in the manuscript on page 11 (lines 28-29).

To clarify our choice of systems, we have added additional sentences for FLN5 (page 2, lines 24-26), I27 (page 11 and lines 14-15) and HRAS (page 12 lines 24-26) in the revised manuscript.

Protein refolding vs coTF

While the Ig-like domains we have studied here can refold successfully, we note there is both established and growing literature evidence that protein refolding (if unfolded in the first place) is significantly less efficient than *de novo* folding in the cell. We have added an additional sentence and references in the introduction on page 1 (lines 33-37) to clarify this and illustrate that, hence, understanding coTF is physiologically relevant in general. To further support the claims and importance of the conclusions from our study, we have investigated the refolding of HRAS which has not been extensively studied in contrast to the other two systems (FLN5 and titin I27). When we studied the refolding properties of HRAS off the ribosome, we found to our surprise that even this single-domain protein is unable to refold into a fully native and active conformation, including in eukaryotic cell lysate, and instead, only achieves this when folding on the ribosome during *de novo* synthesis. Our data also show that fully native state and protein activity begin to form on the ribosome, which directly connects the *de novo* folding pathway via the two coTF intermediates with the activity of the synthesised protein. These data have been added in the revision in Figure S12 and are described in the main text (page 12/line 21-page 13/line 6). Thus, understanding the general physical principles and mechanisms of early coTF events starting with the unfolded state and folding energetics is of general importance.

2) Even more so is the effect of mutations on stability. It is irrelevant whether ribosome "buffers" such effects as their biological effects are determined by stability of the variants in the cytoplasm without the ribosome.

We thank the reviewer for bringing this up and realise that the buffering effect requires further clarification. The reviewer's point is strongly related to their point 1 (and our response) above, in that the relevance of co- versus post-translational folding/stability is questioned. In addition to our new HRAS data (Figure S12 in the revised manuscript) mentioned in response to point 1 above, in the context of buffering we additionally argue that many mutants that we studied in our manuscript remain natively folded in isolation (Figure S13) but would be expected to fully unfold co-translationally in the absence of buffering because folded structure on the ribosome is less stable than the native state in isolation (see

response to point 4 below). Therefore, mutations can shift a protein from folding co-translationally to folding post-translationally and buffering promotes coTF for mutant proteins. In this way, buffering can help nascent proteins avoid potentially harmful consequences of mutation-induced unfolding on the ribosome, such as degradation/ubiquitination²¹ or post-translational misfolding (see e.g., HRAS, Figure S12). We have added additional text in the discussion paragraph of the revised manuscript page 16 (lines 9-18), where we discuss and clarify this.

Moreover, we would also like to stress that even temporary/transient buffering effects can have implications for protein function. As we summarise in the discussion of the manuscript (page 16, lines 18-21), chaperones have also been implicated in mutation buffering and shown to directly modulate protein evolutionary rates. Both chaperones and the ribosome do not remain associated with proteins indefinitely and interact/guide the nascent chain for limited periods of time during or after translation. These notable parallels between the ribosome and chaperones therefore suggest that mutation buffering by the ribosome is also likely to have biological implications for protein function and evolution.

Additionally, we have added new ¹⁹F NMR data showing that the mutation buffering effect correlates with the temperature dependence of coTF, linking mutation buffering to the general thermodynamic effects we describe in the manuscript. These data have been added to Figure 5D and are described on page 13 (lines 24-29).

3) The authors analyses of stability effects in the unfolded state appears to contradict basic principles of thermodynamics. They argue that the unfolded state on the Rs is entropically depleted and so that the corresponding increase in free energy cannot be compensated by energy of RS binding. But that means that binding (or more generally constraints imposed by Rs) goes uphill in free energy compared to the unconstrained unfodled state unbound to the Rs. In this scenario it is this state that will be observed as it is lower in free energy rather than unfolded fragment of the protein bound to the Rs. The fact that actually Rs-bound state is observed means that some estimates of free energies are off.

We have re-evaluated this extensively and respectfully disagree with the reviewer. We concluded that our free energy estimates and the corresponding models are thermodynamically reasonable, and we have added additional clarification in the revised manuscript as indicated below:

- a) We have previously shown (ref.¹) that ribosome binding (*interactions*) of the C-terminal NC to the ribosome surface is energetically favoured (i.e., $\Delta G_{\text{binding}} < 0$ since the bound population at FLN5+31 is $85 \pm 5\%$). Therefore, ribosome *interactions* with the unfolded nascent chain do not go uphill but rather downhill energetically (with respect to the non-interacting unfolded NC). We have updated the energy diagram in Figure 3D of the revised manuscript to clarify this point.
- b) The entropic contribution to energetics of RNCs is independent of the ribosome surface interaction contribution, as supported by our ¹⁹F NMR data on the FLN5 E6 variant (Figure S9I-N). Thus, the entire unfolded state population on the ribosome is entropically depleted/destabilised (regardless of ribosome interaction status, relative to the isolated unfolded state). We believe the updated energy diagram in Figure 3D also clarifies this.
- c) The statement that the “*increase in free energy cannot be compensated by energy of RS binding*” is correct. The increase in free energy we discuss in the manuscript is relative to the isolated (off the ribosome) protein, and not relative to the NC that is tethered to but not interacting with the ribosome surface. The energetic contributions to the modulation of the unfolded state (RNC relative to the isolated protein) are summarised in Figure 2F in the manuscript.
- d) The subpopulation of the unfolded state on the ribosome that binds to the surface ($85 \pm 5\%$ at FLN5+31) only involves the C-terminus of FLN5 and is in fast exchange (on the NMR timescale) with the free state, having a lifetime of $< 3\text{ms}^1$. These bound residues (730-750) are not observed by NMR due to extensive line broadening¹. We observe resonances for the RNC residues upstream of this C-terminal region, and the NMR-observable residues do not interact appreciably with the ribosome surface¹.

4) While the authors explore energetic consequences of unfolded state binding to the RS they omit from consideration the effect of Rs on the folded state of the nascent chain. This is a serious omission as the free energy balance of the transient state is determined by both effect of the Rs on both folded and unfolded states and intermediates, But as pointed out in (1) the free energy of the transient Rs bound state is of little biological significance.

We agree with the reviewer that the energetic analysis is incomplete without additional data on the folded RNC. We have added additional analyses of the folded state by NMR in the revised manuscript in Figure S9O-Q, described in the main text on page 8 (lines 2-10). The data show that the structure and/or environment of the folded domain is different on versus off the ribosome, thus supporting that the free energy of this state is different as an RNC. We stress that the notion that long-range forces, such as electrostatics, can destabilise natively folded NCs on the ribosome (relative to in isolation) is in agreement with earlier pulse proteolysis data²² and recent mass spectrometry experiments²³ in the literature for other nascent proteins. We mention and cite these additional sources in the first paragraph of the discussion (page 14/line 22 - page 15/line 2) in the revised manuscript.

The paper is clearly written, and statistical analyses appear adequate.

All in all while this work reports valuable data on structure of the ensemble of Rs bound nascent chain its conclusions and significance are not clear cut.

We thank the reviewer for their valuable feedback, but respectfully argue very strongly indeed that our manuscript adds significantly more in addition to the data on the structural ensembles of RNCs. Our manuscript addresses a longstanding gap in our understanding of coTF thermodynamic, a subject that has remained at a rudimentary stage in the field for the last two decades. Consequently, it was unknown why protein folding on the ribosome differed from isolated *in vitro* folding for so many proteins. Despite the well-established role of coTF in enabling production of native proteins which in many cases do not refold spontaneously, the underlying thermodynamic principles governing this phenomenon remained unclear (see Introduction paragraphs of the manuscript). This is in stark contrast to the extensive literature on protein folding *in vitro*, where thermodynamic parameters for many small proteins have been determined by biophysical methods, and many fundamental principles and driving forces of protein folding were already well known²⁴.

We hope the addition of new experimental data and computational analyses in the revised manuscript clarify our conclusions:

- a) The field primarily focused on the role of NC-ribosome surface interactions. We show for the first time that these ribosome interactions are not the dominant factor determining coTF energetics (Figures 3C/S9I-N, page 8/lines 12-22 and page 9/lines 17-30), which are instead mostly modulated due to an increased solvation of unfolded NCs relative to their isolated counterparts.
- b) Our updated manuscript now shows a clear correlation between the structural expansion of the structural ensemble of the unfolded states and the changes in thermodynamics on/off the ribosome (Figure 3A-B in the revision).
- c) The entropic destabilisation of unfolded NCs (computationally predicted in Figure 2F, experimentally validated in Figure 2H-I) is a novel finding that has not been reported to date.
- d) Our manuscript, for the first time, presents experimental measurements of the folding enthalpy and entropy on the ribosome (Figures 2-5), made possible by our recent development of ¹⁹F NMR applied to RNCs and improved sample preparation protocols developed over the last two decades to study coTF (ref.¹¹).
- e) We have shown that our findings are generalisable by extending the observation of stable coTF intermediates and thermodynamics to two more protein systems (titin I27 and HRAS, Figure 4), and we have quantitatively shown for several mutants that the ribosome buffers destabilising mutations.
- f) Our analyses rationalise, also for the first time, why coTF intermediates are stabilised on the ribosome and how the ribosome promotes folding via partially structured states. Our thermodynamic model thus reconciles and explains the results of decades-old (mainly biochemical, see Introduction and references therein) studies that found coTF proceeds via states not populated in isolation (see Discussion, page 15, lines 13-17), often resulting in greater folding yields than refolding in isolation.
- g) In the case of HRAS, we likewise show that the re-balancing of folding enthalpy/entropy and altered folding pathway via coTF intermediates on the ribosome is indeed required to yield in active, functional protein (Figure S12 in the revision).

References

- 1 Cassaignau, A. M. E. *et al.* Interactions between nascent proteins and the ribosome surface inhibit co-translational folding. *Nat Chem* **13**, 1214-1220, doi:10.1038/s41557-021-00796-x (2021).
- 2 Deckert, A. *et al.* Common sequence motifs of nascent chains engage the ribosome surface and trigger factor. *P Natl Acad Sci USA* **118**, doi:ARTN e2103015118 10.1073/pnas.2103015118 (2021).
- 3 Wang, X. *et al.* Probing the dynamic stalk region of the ribosome using solution NMR. *Sci Rep* **9**, 13528, doi:10.1038/s41598-019-49190-1 (2019).
- 4 Burridge, C. *et al.* Nascent chain dynamics and ribosome interactions within folded ribosome-nascent chain complexes observed by NMR spectroscopy. *Chem Sci* **12**, 13120-13126, doi:10.1039/d1sc04313g (2021).
- 5 Lin, S. T., Maiti, P. K. & Goddard, W. A., 3rd. Two-phase thermodynamic model for efficient and accurate absolute entropy of water from molecular dynamics simulations. *J Phys Chem B* **114**, 8191-8198, doi:10.1021/jp103120q (2010).
- 6 Hsu, S. T. *et al.* Structure and dynamics of a ribosome-bound nascent chain by NMR spectroscopy. *Proc Natl Acad Sci U S A* **104**, 16516-16521, doi:10.1073/pnas.0704664104 (2007).
- 7 Cabrita, L. D., Hsu, S. T. D., Launay, H., Dobson, C. M. & Christodoulou, J. Probing ribosome-nascent chain complexes produced *in vivo* by NMR spectroscopy. *P Natl Acad Sci USA* **106**, 22239-22244, doi:10.1073/pnas.0903750106 (2009).
- 8 Deckert, A. *et al.* Structural characterization of the interaction of alpha-synuclein nascent chains with the ribosomal surface and trigger factor. *Proc Natl Acad Sci U S A* **113**, 5012-5017, doi:10.1073/pnas.1519124113 (2016).

- 9 Cassaignau, A. M. *et al.* A strategy for co-translational folding studies of ribosome-bound nascent chain complexes using NMR spectroscopy. *Nat Protoc* **11**, 1492-1507, doi:10.1038/nprot.2016.101 (2016).
- 10 Cabrita, L. D. *et al.* A structural ensemble of a ribosome-nascent chain complex during cotranslational protein folding. *Nat Struct Mol Biol* **23**, 278-285, doi:10.1038/nsmb.3182 (2016).
- 11 Chan, S. H. S. *et al.* The ribosome stabilizes partially folded intermediates of a nascent multi-domain protein. *Nature Chemistry* **14**, 1165-1173, doi:10.1038/s41557-022-01004-0 (2022).
- 12 Ahn, M. *et al.* Modulating co-translational protein folding by rational design and ribosome engineering. *Nat Commun* **13**, doi:ARTN 4243
10.1038/s41467-022-31906-z (2022).
- 13 O'Brien, E. P., Christodoulou, J., Vendruscolo, M. & Dobson, C. M. New scenarios of protein folding can occur on the ribosome. *J Am Chem Soc* **133**, 513-526, doi:10.1021/ja107863z (2011).
- 14 Camilloni, C., De Simone, A., Vranken, W. F. & Vendruscolo, M. Determination of secondary structure populations in disordered states of proteins using nuclear magnetic resonance chemical shifts. *Biochemistry* **51**, 2224-2231, doi:10.1021/bi3001825 (2012).
- 15 Braselmann, E., Chaney, J. L. & Clark, P. L. Folding the proteome. *Trends Biochem Sci* **38**, 337-344, doi:10.1016/j.tibs.2013.05.001 (2013).
- 16 Manavalan, B., Kuwajima, K. & Lee, J. PFDB: A standardized protein folding database with temperature correction. *Sci Rep* **9**, 1588, doi:10.1038/s41598-018-36992-y (2019).
- 17 Hsu, S. T., Cabrita, L. D., Fucini, P., Dobson, C. M. & Christodoulou, J. Structure, dynamics and folding of an immunoglobulin domain of the gelation factor (ABP-120) from *Dictyostelium discoideum*. *J Mol Biol* **388**, 865-879, doi:10.1016/j.jmb.2009.02.063 (2009).
- 18 Waudby, C. A. *et al.* Systematic mapping of free energy landscapes of a growing filamin domain during biosynthesis. *Proc Natl Acad Sci U S A* **115**, 9744-9749, doi:10.1073/pnas.1716252115 (2018).
- 19 Fowler, S. B. & Clarke, J. Mapping the folding pathway of an immunoglobulin domain: structural detail from Phi value analysis and movement of the transition state. *Structure* **9**, 355-366, doi:10.1016/s0969-2126(01)00596-2 (2001).
- 20 Clarke, J., Cota, E., Fowler, S. B. & Hamill, S. J. Folding studies of immunoglobulin-like beta-sandwich proteins suggest that they share a common folding pathway. *Structure* **7**, 1145-1153, doi:10.1016/s0969-2126(99)80181-6 (1999).
- 21 Duttler, S., Pechmann, S. & Frydman, J. Principles of cotranslational ubiquitination and quality control at the ribosome. *Mol Cell* **50**, 379-393, doi:10.1016/j.molcel.2013.03.010 (2013).
- 22 Samelson, A. J., Jensen, M. K., Soto, R. A., Cate, J. H. & Marqusee, S. Quantitative determination of ribosome nascent chain stability. *Proc Natl Acad Sci U S A* **113**, 13402-13407, doi:10.1073/pnas.1610272113 (2016).
- 23 Tan, R. *et al.* Folding stabilities of ribosome-bound nascent polypeptides probed by mass spectrometry. *Proc Natl Acad Sci U S A* **120**, e2303167120, doi:10.1073/pnas.2303167120 (2023).
- 24 Dill, K. A. Dominant forces in protein folding. *Biochemistry* **29**, 7133-7155, doi:10.1021/bi00483a001 (1990).

Reviewer Reports on the First Revision:

Referees' comments:

Referee #1 (Remarks to the Author):

The revised manuscript by Streit and colleagues answers my previous questions. I found the original version of the manuscript strong and interesting, and with important (and to some extent provocative) biological implications. While some of the analyses remain imprecise for technical reasons, I believe that the authors have done everything one can reasonably expect given the difficulties of studying these systems.

I only have two minor remaining comments, changes to which are in my opinion purely optional

1.

The authors have added new experimental SAXS data, which they compare to the various simulations. They also estimate R_g directly from the SAXS data. First, the authors may note that automatic estimation of R_g from SAXS data on disordered proteins is tricky and that the chosen $q_{max} * R_g$ should possibly set lower than in folded proteins; see e.g. Borgia et al (<https://doi.org/10.1021/jacs.6b05917>) and Pesce et al (authors ref 142). The authors might explore using a different approach to estimate R_g for Fig. S5 panel H, for example an extended guinier analysis method (<https://doi.org/10.1016/j.jmb.2018.03.007>) or using the molecular form factor (<https://doi.org/10.1126/science.aan5774>).

Also, it is unclear why a99sb-disp and CA can only be reweighted a little bit whereas c36m+W can apparently be reweighted more. Is this due to inherent problems with these ensembles or just a choice of the balance between experiments and force field?

2.

The authors report uncertainties with >1 significant figures (up to 3 in 29.0 +/- 10.3) in places; I would suggest sticking to just 1 significant figure.

Comment on concerns by reviewer #3:

I understand the point of the reviewer that the thermodynamic effect identified in the paper might not have a biological consequence for these specific systems. I also, however, find the authors' response reasonable, namely that they selected these two systems (FLN5/I27) to look at proteins that can fold reversibly. And I also find the results on HRAS, while perhaps less developed, convincing and very interesting. I think the paper shows an important conceptual point (that the thermodynamics might be perturbed and buffered on the ribosome) and one specific example (HRAS) where there might be biological consequences. But I also think that the conceptual parts are sufficiently strong to suggest that this could be a wider spread phenomenon with an impact on many proteins. I don't know, but maybe this paper would spur many such studies.

Referee #2 (Remarks to the Author):

With the revised manuscript, the technical challenge of combining multiple techniques and developing nontrivial sample preparation to study the multi-component ribosomal nascent chain (RNC) complexes is better articulated. Accordingly, the revised discussion on pages 15-16 also helps the significance of understanding the RNC thermodynamics to reach a boarder audience beyond the co-translational folding field (related to my comment #1).

The new PRE data in Figure 3 and S10 address my comment #2 and convincingly support the entropic effect as the nascent chain elongates from the ribosomal exit vestibule.

I also appreciate the new SAXS data (Figure S5) that independently validate the modeled structural ensemble (my comment #3).

Therefore, the authors have addressed all my comments satisfactorily.

Minor point:

As the authors mentioned in the reply, the stability of the RNC complexes is limited (<1 d at 298 K). Given the relatively long data acquisition time for ^{19}F or ^{15}N NMR (several days), it may be beneficial for the authors to add how they ensure the sample quality during and after the NMR measurements in the experimental methods section.

Referee #3 (Remarks to the Author):

I reviewed the responses by the authors and did not find their counterarguments compelling. My main criticism is that as applied to their model systems the proteins can spontaneously fold and unfold in cytoplasm post-translationally making the thermodynamic effect of Rs on the protein irrelevant regardless of whether their thermodynamic estimate are right or wrong (and they are wrong in important aspects. Specifically, if one includes the enthalpic effect of binding to Rs in the "ENTROPIC" effect of Rs on the unfolded state then the balance of stability should include also the effect of Rs binding changing the conclusion as in the title. One CANNOT separate enthalpic and entropic effects while analyzing the effect on stability. That is possible only in a fully non-equilibrium system which a folded protein in cytoplasm is not). Whether their analysis can be extended to proteins that fold irreversibly remains to be seen and represents a pure speculation at this point.

Author Rebuttals to First Revision:

Response to reviewers (manuscript version 2)

The ribosome lowers the entropic penalty of protein folding, Streit *et al.*

We are thankful to the reviewers for their comments. Reviewers 1 and 2 seem entirely appreciative of our revised manuscript that considered all reviewer comments in detail and that have certainly improved the manuscript. Reviewer 3's new comments are addressed below but we firmly believe that it is unlikely that any changes/additions on our part would convince this reviewer. As we stated in our initial cover letter, 'Reviewer 3 also made some important points that enabled improvements to the manuscript, but we also felt that this reviewer overlooked the biological significance of our novel findings and methodologies and have thus made those clearer in our response and revision'. As we also stated in our initial cover letter, 'no other group is currently able to undertake the kind of advanced integrative analysis of nascent chain structure and folding thermodynamics that we performed in this work.'

Referee #1

The revised manuscript by Streit and colleagues answers my previous questions. I found the original version of the manuscript strong and interesting, and with important (and to some extent provocative) biological implications. While some of the analyses remain imprecise for technical reasons, I believe that the authors have done everything one can reasonably expect given the difficulties of studying these systems.

We thank the reviewer for their positive feedback.

I only have two minor remaining comments, changes to which are in my opinion purely optional

1.

The authors have added new experimental SAXS data, which they compare to the various simulations. They also estimate R_g directly from the SAXS data. First, the authors may note that automatic estimation of R_g from SAXS data on disordered proteins is tricky and that the chosen $q_{max} \cdot R_g$ should possibly set lower than in folded proteins; see e.g. Borgia *et al* (<https://doi.org/10.1021/jacs.6b05917>) and Pesce *et al* (authors ref 142). The authors might explore using a different approach to estimate R_g for Fig. S5 panel H, for example an extended guinier analysis method (<https://doi.org/10.1016/j.jmb.2018.03.007>) or using the molecular form factor (<https://doi.org/10.1126/science.aan5774>).

We agree that estimating R_g values from SAXS of disordered proteins is 'tricky', and this is the reason we have also compared our MD simulations directly against the full scattering profiles (Extended Data Fig. in the revised version). This comparison of MD ensembles with SAXS data shows the same trend as from the inferred R_g values: the C36m+W ensemble agrees best with the SAXS data. We welcome the additional suggestion to determine R_g with an alternative approach (e.g., the molecular form factor analysis), which yields a R_g of $34.8 \pm 0.1 \text{ \AA}$, in line with the value obtained by Guinier analysis ($34.4 \pm 0.6 \text{ \AA}$).

Molecular form factor analysis of the SAXS data (output from the webserver provided by Riback *et al.*, *Science*, 2007).

For completeness, we have added the MFF-derived value to Extended Data Fig. 4H.

Also, it is unclear why a99sb-disp and CA can only be reweighted a little bit whereas c36m+W can apparently be reweighted more. Is this due to inherent problems with these ensembles or just a choice of the balance between experiments and force field?

This indeed arises from a combination of the balance between experiment and MD (by nature of the L-curve analysis, Extended Data Fig. 3I), but our additional comparisons with NMR chemical shifts, RDCs and SAXS also indicate that C36m+W is more accurate for our system (Extended Data Fig. 4). We have briefly clarified this in supplementary note 5 (final paragraph of this SI note) in the SI.

2.

The authors report uncertainties with >1 significant figures (up to 3 in 29.0 +/- 10.3) in places; I would suggest sticking to just 1 significant figure.

We have changed the reported numbers from the computational analyses to 1 significant figure in the manuscript.

Comment on concerns by reviewer #3:

I understand the point of the reviewer that the thermodynamic effect identified in the paper might not have a biological consequence for these specific systems. I also, however, find the authors' response reasonable, namely that they selected these two systems (FLN5/I27) to look at proteins that can fold reversibly. And I also find the results on HRAS, while perhaps less developed, convincing and very interesting. I think the paper shows an important conceptual point (that the thermodynamics might be perturbed and buffered on the ribosome) and one specific example (HRAS) where there might be biological consequences. But I also think that the conceptual parts are sufficiently strong to suggest that this could be a wider spread phenomenon with an impact on many proteins. I don't know, but maybe this paper would spur many such studies.

We thank the reviewer for their assessment and agree.

Referee #2

With the revised manuscript, the technical challenge of combining multiple techniques and developing nontrivial sample preparation to study the multi-component ribosomal nascent chain (RNC) complexes is better articulated. Accordingly, the revised discussion on pages 15-16 also helps the significance of understanding the RNC thermodynamics to reach a boarder audience beyond the co-translational folding field (related to my comment #1). The new PRE data in Figure 3 and S10 address my comment #2 and convincingly support the entropic effect as the nascent chain elongates from the ribosomal exit vestibule.

I also appreciate the new SAXS data (Figure S5) that independently validate the modeled structural ensemble (my comment #3).

Therefore, the authors have addressed all my comments satisfactorily.

We thank the reviewer for their positive comments

Minor point:

As the authors mentioned in the reply, the stability of the RNC complexes is limited (<1 d at 298 K). Given the relatively long data acquisition time for 19F or 15N NMR (several days), it may be beneficial for the authors to add how they ensure the sample quality during and after the NMR measurements in the experimental methods section.

Absolutely! The inherent limited stability of ribosomal samples necessitates continual monitoring of NMR samples to ensure that the observed signals derive exclusively from intact ribosome-bound complexes rather than released or degraded species. Indeed, given their disparate dynamics, even a small population (a few percent) of released/degraded material will result in large changes in NMR spectra of otherwise intact RNCs.

Our rigorous approach to assess sample integrity, developed over several years is embedded in the experimental design. NMR experiments are recorded and continuously interleaved with a series of 1D 1H/19F spectra and 1H,15N/19F diffusion measurements (Extended Data Fig. 2H). These provide the most sensitive means to assess changes in the sample, and when alterations in signal intensities or linewidths (i.e., transverse relaxation rates), chemical shifts, or translational diffusion measurements of the nascent chain are observed, data acquisition is halted. Only data corresponding to intact RNCs are summed together and subjected to a final round of analysis. Where signal-to-noise remains low, data sets from multiple samples are compared to ensure identical spectra, before summation together into a single NMR spectrum. Biochemical assays provide an orthogonal means to assess nascent chain attachment to the ribosome. Identical samples incubated in parallel with NMR samples are subjected to SDS-PAGE (under low pH conditions, e.g., <https://www.nature.com/articles/nprot.2016.101>) analysed and detected by nascent chain-specific antibodies. Ribosome-bound species migrate with an addition ~17-kDa band-shift relative to released nascent chains due to the presence of the tRNA covalently linked to the nascent chain. Combined with time-resolved NMR measurements, these analyses confirm that the reported NMR resonances originate exclusively from intact RNCs.

We have clarified this in the Methods section (last paragraph of the NMR section).

Referee #3

Remarks to the Author: I reviewed the responses by the authors and did not find their counterarguments compelling. My main criticism is that as applied to their model systems the proteins can spontaneously fold and unfold in cytoplasm post-translationally making the thermodynamic effect of Rs on the protein irrelevant regardless of whether their

thermodynamic estimate are right or wrong (and they are wrong in important aspects. Specifically, if one includes the enthalpic effect of binding to Rs in the "ENTROPIC" effect of Rs on the unfolded state then the balance of stability should include also the effect of Rs binding changing the conclusion as in the title. One CANNOT separate enthalpic and entropic effects while analyzing the effect on stability. That is possible only in a fully non-equilibrium system which a folded protein in cytoplasm is not). Whether their analysis can be extended to proteins that fold irreversibly remains to be seen and represents a pure speculation at this point.

In response to reviewer 3's initial review comments, we described substantial additional experiments and clarifications carried out over the past several months, (pages 10-14 of response) and in the revised manuscript (previous version, pages 1-2,8,11,12-16 & Figures 3D,S9O-Q,S12). These changes dishearteningly appear not to have been engaged at all. Over 100 ribosomal samples were designed and investigated by NMR and despite such an extensive range of experiments it appears unappreciated for any of their stark and novel conclusions (i.e., the PRE NMR data clearly revealing the expansion and the thermodynamic results from ¹⁹F-NMR analyses).

Reviewer 3 re-iterates their previously made argument that the proteins we study "can spontaneously fold and unfold in the cytoplasm post-translationally making the thermodynamic effect of Rs on the protein irrelevant". This is true for FLN5/127, which we have chosen specifically for this reason (see page 2/lines 24-26 and page11/lines 14-15 in the previous version of the revised manuscript) to enable the first enthalpy/entropy comparisons on/off the ribosome that require reversible folding. **Therefore, we have additionally added a statement in the discussion to provide a more balanced assessment of these effects.** However, the reviewer also seemingly ignored both our revised Introduction (page 1, '[many proteins] fail to refold spontaneously') referencing literature that this is not true for many other proteins, and the comprehensive series of experiments we added in the revised manuscript to specifically address this point, where we show that the HRAS protein does not refold spontaneously to an active conformation post-translationally, but only cotranslationally (Extended Data Fig. 9). Moreover, we have generalised our analyses using different protein systems. Extending our analyses to proteins that fold irreversibly is therefore surely not able to be dismissed as "a pure speculation at this point".

We disagree with reviewer 3's statement that our thermodynamic estimates are "wrong", strongly contrasting the assessments by reviewers 1 and 2. Reviewer 3's explanation of why our thermodynamic estimates are "wrong" appear to refer to the contribution of enthalpy of the nascent chain binding to the ribosome surface. We stress that we have not ignored the contribution of ribosome binding; in fact, our group published the first quantitative studies investigating such interactions, and given these effects, a substantial portion of the original and revised manuscript is dedicated to understanding and quantifying these interactions relative to the entropic effects. Our experimental measurements of ribosome binding highlight that the entropic effects are substantially greater in magnitude than, and thus outcompete, ribosome binding-enthalpy contributions (and indeed other thermodynamic effects, Fig. 2F, Extended Data Fig. 7-8). This point is further addressed in our revised manuscript, where we demonstrate longer RNCs are subjected to persistent entropic effects despite significantly diminished ribosome interactions (Fig. 3). We have summarised in the discussion: "The entropic destabilisation observed in the unfolded NC relative to the isolated protein outcompetes the enthalpic stabilisation provided by electrostatic ribosome interactions and increased solvation (Figure 2F)." The response from reviewer 3 highlights the prevailing view of the coTF field that ribosome interactions dominate the thermodynamics of coTF. Our work is thus progressing an understanding of coTF beyond nascent chain-interactions, supported by direct experiments investigating the temperature dependence of folding on and off the ribosome (Fig. 2-5), and reconciles the paradoxical formation of stable coTF intermediates on the ribosome which is common to many proteins as stated in the manuscript's introduction and discussion.